# SA-PEF: Step-Ahead Partial Error Feedback for Efficient Federated Learning

## Abstract

Biased gradient compression with error feedback (EF) reduces communication in federated learning (FL), but under non-IID data, the residual error can decay slowly, causing gradient mismatch and stalled progress in the early rounds. We propose step-ahead partial error feedback (SA-PEF), which integrates step-ahead (SA) correction with partial error feedback (PEF). SA-PEF recovers EF when the step-ahead coefficient $\alpha = 0$ and step-ahead EF (SAEF) when $\alpha = 1$. For non-convex objectives and $\delta$-contractive compressors, we establish a second-moment bound and a residual recursion that guarantee convergence to stationarity under heterogeneous data and partial client participation. The resulting rates match standard non-convex Fed-SGD guarantees up to constant factors, achieving $O((\eta\,\eta_0 TR)^{-1})$ convergence to a variance/heterogeneity floor with a fixed inner step size. Our analysis reveals a step-ahead-controlled residual contraction $\rho_r$ that explains the observed acceleration in the early training phase. To balance SAEF's rapid warm-up with EF's long-term stability, we select $\alpha$ near its theory-predicted optimum. Experiments across diverse architectures and datasets show that SA-PEF consistently reaches target accuracy faster than EF.

## 1 Introduction

Modern large-scale machine learning increasingly relies on distributed computation, where both data and compute are spread across many devices. Federated learning (FL) enables model training in this setting without centralizing raw data, enhancing privacy and scalability under heterogeneous client distributions (McMahan et al., 2017; Kairouz et al., 2021). In FL, a potentially vast population of clients collaborates to train a shared model $w \in \mathbb{R}^d$ by solving

$$w^\star \in \arg\min_{w \in \mathbb{R}^d} f(w) := \frac{1}{K} \sum_{k=1}^{K} f_k(w), \qquad f_k(w) := \mathbb{E}_{z \sim \mathcal{D}_k}\big[\ell(w; z)\big], \tag{1}$$

where $\mathcal{D}_k$ is the (potentially heterogeneous) data distribution at client $k$, $\ell(\cdot)$ is a sample loss (often nonconvex), and $K$ is the number of clients. In each synchronous FL round, the server broadcasts the current global model to a subset of clients. These clients perform several steps of stochastic gradient descent (SGD) on their local data and return updates to the server, which aggregates them to form the next global iterate (Huang et al., 2022; Wang & Ji, 2022; Li et al., 2024).

Although FL leverages rich distributed data, it faces two key challenges. The first challenge is communication bottlenecks. Model updates are typically high-dimensional, with millions or even billions of parameters, which makes uplink bandwidth a major constraint (Reisizadeh et al., 2020; Kim et al., 2024; Islamov et al., 2025). This has spurred extensive work on communication-efficient algorithms, including quantization (Seide et al., 2014; Alistarh et al., 2017), sparsification (Stich et al., 2018), and biased compression with error feedback (Beznosikov et al., 2023; Bao et al., 2025). The second challenge is statistical heterogeneity. When client data are non-IID, multiple local updates can cause client models to drift toward minimizing their own objectives, slowing or even destabilizing global convergence (Karimireddy et al., 2020; Li & Li, 2023).

To reduce communication, many methods compress client-server messages using quantization (Alistarh et al., 2017), sparsification (Lin et al., 2018), or sketching (Rothchild et al., 2020). Compressors may be unbiased (e.g., Rand-$k$ (Wangni et al., 2018; Stich et al., 2018)) or biased (e.g.,

signSGD (Bernstein et al., 2019), Top-$k$ (Lin et al., 2018)), with the latter often delivering better accuracy-communication trade-offs at a given bit budget (Beznosikov et al., 2023). However, naive use of biased compression can introduce a persistent bias, leading to slow or even divergent training (Beznosikov et al., 2023; Li & Li, 2023). Error feedback (EF) addresses this problem by storing past compression errors and injecting them into the next update before compression (Seide et al., 2014). This mechanism cancels the compressor bias and restores convergence guarantees comparable to those of uncompressed SGD, assuming standard smoothness and appropriately chosen stepsizes (Karimireddy et al., 2019; Bao et al., 2025).

EF in federated settings faces two important limitations. First, under highly non-IID data, the residual can align with client-specific gradient directions, inducing cross-client gradient mismatch and slowing early progress (Hsu et al., 2019b; Karimireddy et al., 2020). Second, EF retains residual mass until fully transmitted. Once the residual norm is small, communication rounds may be wasted transmitting stale, low-magnitude coordinates rather than fresh gradient signal (Li & Li, 2023).

Step-ahead EF (SAEF) (Xu et al., 2021) mitigates the first issue by *previewing* the residual: before local SGD, each client shifts its model by the current error and optimizes from that offset. This strategy often yields a strong warm-up, as the residual is injected in full every round. However, in FL regimes with non-IID data, multiple local steps, aggressive compression, and partial participation, the full step-ahead variant exhibits late-stage plateaus and larger gradient-mismatch spikes compared to EF. Moreover, prior analysis (Xu et al., 2021) has largely focused on classical distributed optimization, leaving open whether one can *systematically* combine SAEF's fast initial progress with the long-term stability of EF in federated settings with local steps and data heterogeneity.

**Our approach: SA-PEF.** We propose *step-ahead partial error feedback (SA-PEF)*, by introducing a tunable coefficient $\alpha_r \in [0, 1]$ and shifting only a *fraction* of the residual ($w_{r+\frac{1}{2}} = w_r - \alpha_r e_r$) while carrying the remainder through standard EF. This partial shift provides several benefits:

- *Early acceleration with reduced noise.* A moderate (or decaying) $\alpha_r$ removes most early-round mismatch, while the injected noise automatically diminishes as $\|e_r\|$ shrinks.
- *Tighter theoretical recursions.* We establish a residual contraction $\rho_r = (1 - \frac{1}{\delta})[2(1 - \alpha_r)^2 + 24\,\alpha_r^2(\eta_r LT)^2]$, which, for small $s_r = \eta_r LT$, is strictly smaller than the EF value $2(1 - \frac{1}{\delta})$. The local-drift constants also improve with $\alpha_r$.
- *Graceful interpolation.* SA-PEF reduces to EF when $\alpha_r = 0$ (maximal stability) and to SAEF when $\alpha_r = 1$ (maximal jump-start), allowing practitioners to adapt to different data and communication regimes.

**Contributions.**

- *Algorithm.* We introduce SA-PEF, a lightweight drop-in variant of Local-SGD with biased compression and a tunable step-ahead coefficient $\alpha_r$, compatible with any $\delta$-contractive compressor.
- *Theory.* We provide a convergence analysis of SA-PEF. Our results include new inequalities for drift, second moments, and residual recursion, which recover EF constants at $\alpha_r = 0$ and quantify how step-ahead alters drift and residual memory. From these, we derive nonconvex stationarity guarantees of order $O\big((\eta\,\eta_0 TR)^{-1}\big)$, with compression dependence appearing only through $(1 - 1/\delta)$ and $\rho_{\max}$, in line with prior compressed-FL work.

## 2 RELATED WORK

**Error feedback and compressed optimization.** Error feedback (EF) was first introduced as a practical heuristic for 1-bit SGD (Seide et al., 2014) and later formalized as a *memory* mechanism in sparsified or biased SGD (Stich et al., 2018; Karimireddy et al., 2019). By accumulating the compression residual and injecting it into subsequent updates, EF restores descent directions and admits convergence guarantees for broad classes of *contractive (possibly biased)* compressors. These results include linear rates in the strongly convex setting and standard stationary-point guarantees in the nonconvex case (Gorbunov et al., 2020; Beznosikov et al., 2023). More recently, EF21 (Richtárik et al., 2021) and its extensions (Fatkhullin et al., 2025) provide a modern error-feedback framework for compressing full gradients (or gradient differences) at a shared iterate in synchronized data-parallel training with $T=1$, achieving clean contraction guarantees and removing the error floor. However, these works assume no local steps and no client drift. A complementary line of work

replaces residual memory with control variates (global gradient estimators), as in DIANA and MA-RINA (Mishchenko et al., 2019; Gorbunov et al., 2021), which reduce or remove compressor bias without maintaining a full residual vector. *EControl* (Gao et al., 2024) regulates the strength of the feedback signal and fuses residual and estimator updates into a single compressed message, providing fast convergence under arbitrary contractive compressors and heterogeneous data.

**Local updates in FL.** Local or periodic averaging (a.k.a. Local-SGD) reduces communication rounds by performing $T > 1$ local steps between synchronizations (Stich, 2019). While effective in homogeneous settings, non-IID data induces *client drift*, where model trajectories diverge across clients, degrading both convergence speed and final accuracy. Several approaches mitigate drift while retaining the communication savings of local updates. *Proximal regularization* (Fed-Prox) stabilizes local objectives by penalizing deviation from the current global model (Li et al., 2020). *Control variates* (SCAFFOLD) estimate and correct the client-specific gradient bias caused by heterogeneity, yielding tighter convergence with multiple local steps (Karimireddy et al., 2020). *Dynamic regularization* (FedDyn) further aligns local stationary points with the global objective via a round-wise correction term, improving robustness on highly non-IID data (Acar et al., 2021).

**Compression with local updates.** Combining local updates with message compression compounds communication savings but also amplifies distortions from both local drift and compression error. FedPAQ (Reisizadeh et al., 2020) performs $T > 1$ local steps and transmits quantized model deltas at synchronization points, exposing explicit trade-offs among the local period, step-sizes, and quantization accuracy. QSparse-Local-SGD (Basu et al., 2019) extends this to contractive compressors, transmitting Top-$k$ updates after $T$ local steps. While achieving significant traffic reduction, it also reveals that aggressive sparsity can destabilize convergence. CSER (Xie et al., 2020) mitigates this with *error reset*, which immediately injects the residual back into the local model to restore stability under high compression. In the federated local-SGD setting with partial participation and biased compression, Fed-EF (Li & Li, 2023) provides a first nonconvex analysis of classical EF and serves as the EF-style backbone that SA-PEF builds upon. On the control-variate side, Scaffnew/ProxSkip (Mishchenko et al., 2022) is a more recent local-training method in the SCAFFOLD family, using probabilistic local updates to achieve theoretical acceleration. However, it still relies on full-precision exchanges and no inherent compression. CompressedScaffnew (Condat et al., 2022) extends this mechanism with quantization, TAMUNA (Condat et al., 2023) further handles partial client participation, and LoCoDL (Condat et al., 2025) generalizes the analysis to arbitrary unbiased compressors. These methods primarily establish accelerated convergence in (strongly) convex settings. In parallel, SCALLION/SCAFCOM (Huang et al., 2024) combines SCAFFOLD-style control variates with compression and, in SCAFCOM, local momentum to handle heterogeneity and partial participation, at the cost of additional per-client state.

**Step-ahead error feedback.** SAEF (Xu et al., 2021) addresses *gradient mismatch*, i.e., the discrepancy between the model used for gradient computation and the model actually updated when delayed residuals are applied. SAEF performs a *preview* shift of the model using the residual before local SGD and augments this with occasional *error averaging* across workers. Although this reduces mismatch and accelerates early progress, the analysis is developed for classical distributed settings with single-step synchronization and *bounded-gradient* assumptions, and does not cover federated regimes with multiple local steps, non-IID data, or biased compressors. Moreover, error averaging requires extra coordination and communication, which is often impractical in cross-device FL.

Despite progress, it remains unclear how to (i) control gradient mismatch in federated settings with local steps and biased compression *without* extra communication, or (ii) combine step-ahead correction with EF to balance early acceleration and long-term stability. Our work closes this gap by introducing SA-PEF, which performs a controlled step-ahead shift with partial residual retention on top of Fed-EF and provides a contraction-based analysis yielding nonconvex guarantees under heterogeneous data.

## 3 PROPOSED ALGORITHM

In EF with *biased* compression (e.g., Top-$k$), each client $k$ maintains a residual $e_r^{(k)}$ of *unsent* coordinates. Although the stochastic gradient is *computed* at the received global model $w_r$, the effective

---

**Algorithm 1:** Step-Ahead Partial Error-Feedback (SA-PEF) for Efficient FL

---

**Input:** total communication rounds $R$, client number $K$, stepsizes $\{\eta_r\}_{r=0}^{R-1}$, step-ahead schedule $\{\alpha_r\}_{r=0}^{R-1}$ with $0 \leq \alpha_r \leq 1$, compressor $\mathcal{C}(\cdot)$, initial model $w_0$, local step number $T$

**1 foreach** *client $k = 1, \ldots, K$ in parallel* **do**

**2** $\quad w_0^{(k)} \leftarrow w_0; \quad e_0^{(k)} \leftarrow 0$

**3 for** $r \leftarrow 0$ **to** $R - 1$ **do**

**4** $\quad$ **foreach** *client $k = 1, \ldots, K$ in parallel* **do**

$\quad\quad$ /* Step-ahead start                                                     */

**5** $\quad\quad w_{r+\frac{1}{2},0}^{(k)} \leftarrow w_r^{(k)} - \alpha_r\, e_r^{(k)}$

$\quad\quad$ /* Local SGD with $T$ steps                                             */

**6** $\quad\quad$ **for** $t \leftarrow 0$ **to** $T - 1$ **do**

**7** $\quad\quad\quad w_{r+\frac{1}{2},t+1}^{(k)} \leftarrow w_{r+\frac{1}{2},t}^{(k)} - \eta_r\, \nabla f\big(w_{r+\frac{1}{2},t}^{(k)}; \zeta_{r,t}^{(k)}\big)$

$\quad\quad$ /* accumulated local update                                            */

**8** $\quad\quad g_r^{(k)} \leftarrow w_{r+\frac{1}{2},0}^{(k)} - w_{r+\frac{1}{2},T}^{(k)}$

**9** $\quad\quad u_{r+1}^{(k)} \leftarrow (1 - \alpha_r)\, e_r^{(k)} + g_r^{(k)}$

**10** $\quad\quad e_{r+1}^{(k)} \leftarrow u_{r+1}^{(k)} - \mathcal{C}\big(u_{r+1}^{(k)}\big)$

**11** $\quad\quad$ send $\mathcal{C}\big(u_{r+1}^{(k)}\big)$ to server

$\quad$ /* Server-side aggregation                                                 */

**12** $\quad u_{r+1} \leftarrow \frac{1}{K} \sum_{k=1}^{K} \mathcal{C}\big(u_{r+1}^{(k)}\big)$

$\quad$ /* apply averaged update                                                   */

**13** $\quad w_{r+1} \leftarrow w_r - \eta\, u_{r+1}$

**14** $\quad$ broadcast $w_{r+1}$ to all clients

**15** $\quad$ **foreach** *client $k$* **do**

**16** $\quad\quad w_{r+1}^{(k)} \leftarrow w_{r+1}$

---

update is applied at a de-errored point $\tilde{w}_r := w_r - \delta_r$, where $\delta_r$ denotes the EF carry (often close to the mean residual $\bar{e}_r := \frac{1}{K}\sum_k e_r^{(k)}$ under biased sparsification). Thus, the gradient used for the step is *stale* with respect to the point being updated, an effect akin to a *staleness-of-one* delay in asynchronous SGD where gradients are evaluated at one iterate but applied to another (Lian et al., 2015). Formally, this mismatch is captured by $g_k(\tilde{w}_r; \zeta) - g_k(w_r; \zeta)$, where $g_k := \nabla f_k$, and under $L$-smoothness the local displacement error satisfies

$$\varepsilon_r^{\text{loc}}(0) := \frac{1}{K} \sum_{k=1}^{K} \mathbb{E}_\zeta \big\| g_k(\tilde{w}_r; \zeta) - g_k(w_r; \zeta)\big\|^2 \leq L^2 \|\delta_r\|^2 + 4\sigma^2.$$

**Algorithm overview.** SA-PEF (Algorithm 1) augments Local-SGD with two complementary ideas: (i) a *step-ahead preview* of each client's EF residual, and (ii) a *partial carry-over* of that residual through standard EF. A per-round weight $\alpha_r \in [0, 1]$ determines how much of the residual is previewed versus retained. In effect, SA-PEF *previews a fraction of the residual and remembers the rest*, achieving fast early progress like SAEF while preserving the long-term stability of EF. Each client $k$ maintains a local model $w_r^{(k)}$ and a residual $e_r^{(k)}$, both initialized at round $r = 0$.

At each communication round $r = 0, \ldots, R - 1$, SA-PEF implements the following steps:

1. *Step-ahead preview.* Before local training, client $k$ shifts its model by a fraction $\alpha_r$ of its residual:

$$w_{r+\frac{1}{2},0}^{(k)} = w_r^{(k)} - \alpha_r\, e_r^{(k)}.$$

This moves gradient evaluation closer to where EF actually applies the update, improving alignment of the next direction with $-\nabla f(w_r)$.

2. *Local SGD.* Starting from $w^{(k)}_{r+\frac{1}{2},0}$, client $k$ performs $T$ local SGD steps with stepsize $\eta_r$:

$$w^{(k)}_{r+\frac{1}{2},t+1} \;=\; w^{(k)}_{r+\frac{1}{2},t} - \eta_r \nabla f\big(w^{(k)}_{r+\frac{1}{2},t};\zeta^{(k)}_{r,t}\big), \quad t = 0,\ldots,T-1.$$

The accumulated local update is: $g^{(k)}_r \;=\; w^{(k)}_{r+\frac{1}{2},0} - w^{(k)}_{r+\frac{1}{2},T}$.

3. *Partial EF composition.* SA-PEF blends the remaining residual with the new local update:

$$u^{(k)}_{r+1} \;=\; (1-\alpha_r)\,e^{(k)}_r + g^{(k)}_r.$$

The compressed message and residual update are then $\tilde{u}^{(k)}_{r+1} = \mathcal{C}\big(u^{(k)}_{r+1}\big)$, $e^{(k)}_{r+1} = u^{(k)}_{r+1} - \tilde{u}^{(k)}_{r+1}$. Only $\tilde{u}^{(k)}_{r+1}$ is transmitted to the server and $e^{(k)}_{r+1}$ is retained locally.

4. *Server aggregation.* The server averages compressed updates and applies them:

$$u_{r+1} = \frac{1}{K}\sum_{k=1}^{K}\tilde{u}^{(k)}_{r+1}, \qquad w_{r+1} = w_r - \eta\,u_{r+1}.$$

The new global model $w_{r+1}$ is broadcast to all clients for the next round.

**Relation to EF and SAEF.** The step-ahead coefficient $\alpha_r$ allows SA-PEF to smoothly interpolate between prior methods:

- $\alpha_r = 0$: reduces to Fed-EF/classical EF in the federated local-SGD setting (Li & Li, 2023).
- $\alpha_r = 1$: reduces to a full step-ahead variant analogous to SAEF (Xu et al., 2021).
- $0 < \alpha_r < 1$: partial preview, fast early progress with reduced late-stage noise.

A first-order view (descent lemma) gives $g_k(w_r - \alpha_r e^{(k)}_r) \approx g_k(w_r) - \alpha_r \nabla^2 f_k(w_r)\,e^{(k)}_r$, so $\mathbb{E}_r[\bar{g}_r(\alpha_r)] \approx \nabla f(w_r) - \alpha_r \bar{H}_r \bar{e}_r$ with $\bar{H}_r := \frac{1}{K}\sum_k \nabla^2 f_k(w_r)$. Thus preview provides a *linear* alignment gain via $-\alpha_r \bar{H}_r \bar{e}_r$. On the other hand, the *local-displacement* mismatch between $\nabla f_k(w_r)$ and $\nabla f_k(w_r - \alpha_r e^{(k)}_r)$ grows *quadratically* with $\alpha_r$ under $L$-smoothness. Hence, SA-PEF chooses an *interior* $\alpha_r \in (0,1)$ to balance alignment benefits against the smoothness-driven cost, while preserving EF memory through $(1-\alpha_r)e^{(k)}_r$.

**Residual contraction.** Our analysis yields a per-round residual contraction

$$\rho_r = \Big(1 - \tfrac{1}{\delta}\Big)\Big(2(1-\alpha_r)^2 + 24\,\alpha_r^2(\eta_r LT)^2\Big),$$

which is strictly smaller than EF's value $2(1-1/\delta)$ when $s_r = \eta_r LT$ is small, explaining SA-PEF's faster early-phase convergence while retaining EF-style long-term stability.

## 4 CONVERGENCE ANALYSIS

**Definition 1** (Compression Operator). *A (possibly randomized) mapping $\mathcal{C} : \mathbb{R}^d \to \mathbb{R}^d$ is called a compression operator if there exists $\delta > 0$ such that, for all $w \in \mathbb{R}^d$,*

$$\mathbb{E}\big[\|\mathcal{C}(w) - w\|_2^2\big] \;\leq\; \Big(1 - \tfrac{1}{\delta}\Big)\|w\|_2^2, \tag{4}$$

*where the expectation is taken over the internal randomness of $\mathcal{C}$.*

This class includes many commonly used compressors, such as Top-$k$, scaled Rand-$k$, and quantization operators. The parameter $\delta$ captures the contraction strength, with larger $\delta$ corresponding to weaker contraction (i.e., more aggressive compression).

Our objective is to establish nonconvex convergence guarantees by upper-bounding the average squared gradient norm: $\frac{1}{R}\sum_{r=0}^{R-1} \mathbb{E}\|\nabla f(w_r)\|^2$, where expectations are taken over mini-batch sampling, client selection, and compression randomness.

We adopt the standard assumption set used in Alistarh et al. (2018); Li & Li (2023); Beznosikov et al. (2023) and follow the notation from Alg. 1. Fix a communication round $r$ and let $\{\mathcal{F}_{r,t}\}_{t\geq 0}$

denote the natural filtration generated by all randomness up to the beginning of local step $t$ of round $r$. At local step $t \in \{0, \dots, T-1\}$ on client $k$, the stochastic gradient takes the form $g_{r,t}^{(k)} = \nabla f_k\left(w_{r+\frac{1}{2},t}^{(k)}\right) + \xi_{r,t}^{(k)}$, where $\xi_{r,t}^{(k)}$ captures stochastic noise.

**Assumption 1** (Smoothness). *Each local objective $f_k$ is differentiable with L-Lipschitz gradient:* $\left\| \nabla f_k(y) - \nabla f_k(x) \right\| \leq L \| y - x \|, \forall x, y \in \mathbb{R}^d.$

**Assumption 2** (Stochastic gradients). *For every client $k$ and step $t$, we have (i) $\mathbb{E}[\xi_{r,t}^{(k)} | \mathcal{F}_{r,t}] = 0$ (unbiasedness), (ii) $\mathbb{E}[\|\xi_{r,t}^{(k)}\|^2 | \mathcal{F}_{r,t}] \leq \sigma^2$ (bounded variance), (iii) given $\mathcal{F}_{r,0}$, the family $\{\xi_{r,t}^{(k)} : k = 1, \dots, K, \ t = 0, \dots, T-1\}$ is conditionally independent.*

**Assumption 3** (Gradient dissimilarity). *There exist constants $\beta^2 \geq 1$ and $\nu^2 \geq 0$ such that,*

$$\frac{1}{K} \sum_{k=1}^{K} \left\| \nabla f_k(x) \right\|^2 \ \leq \ \beta^2 \left\| \nabla f(x) \right\|^2 \ + \ \nu^2, \qquad \forall x \in \mathbb{R}^d$$

We now state a convergence guarantee for SA-PEF in our federated setting. The full proof is provided in Appendix A.1.

**Theorem 1** (Stationary-point bound with constant inner-loop step). *Assume 1–3 and let the compressor satisfy Definition 1 with parameter $\delta \geq 1$. Run SA-PEF for $R \geq 1$ rounds with a constant inner-loop stepsize $\eta_r \equiv \eta_0$, and set $s_0 := \eta_0 L T \leq \frac{1}{8}$. Suppose further that $18\,\beta^2 s_0^2 \leq \frac{1}{8}$ and $\eta \leq \frac{1}{256\,\beta^2 L \eta_0 T}$. Define the maximal residual contraction*

$$\rho_{\max} := \sup_r \left(1 - \tfrac{1}{\delta}\right)\left(2(1-\alpha_r)^2 + 24\,\alpha_r^2 s_0^2\right) < 1,$$

*and the effective error constant*

$$\Theta := \frac{16}{\eta} \times \frac{\mathcal{E}_{\max}}{1 - \rho_{\max}} \left(1 - \tfrac{1}{\delta}\right)\beta^2 \left(8\,\eta_0 T + 288\, L^2 \eta_0^3 T^3\right),$$

*where $\mathcal{E}_{\max} := \sup_r \mathcal{E}_r$ is the maximum residual–error coefficient across rounds. If $\Theta \leq \frac{1}{2}$, then with $f^\star := \inf_x f(x)$ and initial residuals $e_0^{(k)} \equiv 0$, we have*

$$\frac{1}{R} \sum_{r=0}^{R-1} \mathbb{E}\|\nabla f(w_r)\|^2 \ \leq \ \frac{32\,(f(w_0) - f^\star)}{\eta\,\eta_0 T\,R} \ + \ \left(1 - \tfrac{1}{\delta}\right)\left[C_\sigma\,\eta_0^2 L^2 T\,\sigma^2 + C_\nu\,\eta_0^2 L^2 T^2\,\nu^2\right]$$

$$+ \ \frac{128\,L\,\eta_0}{K}\,\sigma^2,$$

*where*

$$C_\sigma = \frac{32}{\eta}\left[6\,\eta\,\eta_0^2 L^2 T + 96\,L^3 \eta\,\eta_0^3 T^2\right] + \frac{32}{\eta} \times \frac{\mathcal{E}_{\max}}{1 - \rho_{\max}}\left[4\,\eta_0 + 96\,L^2 \eta_0^3 T^2\right],$$

$$C_\nu = \frac{32}{\eta}\left[84\,\eta\,\eta_0^2 L^2 T^2 + 1344\,L^3 \eta\,\eta_0^3 T^3\right] + \frac{32}{\eta} \times \frac{\mathcal{E}_{\max}}{1 - \rho_{\max}}\left[8\,\eta_0 T + 1344\,L^2 \eta_0^3 T^3\right].$$

**Discussion.** Our result matches the standard nonconvex picture for *biased* compression. With a constant inner stepsize $\eta_0$, the optimization error decreases at rate $O\left((\eta\,\eta_0 T R)^{-1}\right)$, while an $R$-independent floor remains due to residual drift. As in prior EF analyses, only the *mini-batch variance* benefits from a $1/K$ reduction, where the residual-induced floor does not. Data heterogeneity contributes additively with a $T^2$ multiplier in the floor, while the stochastic variance floor carries a $T$ multiplier (both scaled by $\eta_0^2 L^2$). The effect of compressor appears only via the usual bias factor $(1 - 1/\delta)$ and the residual contraction $\rho_{\max} < 1$, which depends on $\alpha_r$ and $s_r = \eta_r L T$. This matches the qualitative dependence reported in earlier analyses of compressed FL (Li & Li, 2023; Karimireddy et al., 2020).

**Remark 1** (Partial participation (PP)). *Let $p = m/K \in (0, 1]$ be the participation rate, and assume constant client stepsize $\eta_r = \eta_0$ with $T$ local steps per round. Then, the averaged stationarity bound under partial participation is*

$$\frac{1}{R} \sum_{r=0}^{R-1} \mathbb{E}\|\nabla f(w_r)\|^2 \leq \mathcal{O}\left(\frac{f(w_0) - f_\star}{\eta\,p\,\eta_0 T\,R}\right) + \mathcal{O}\left(\frac{L\,\eta_0}{\eta\,p\,m}\,\sigma^2\right) + \mathcal{O}\left(\frac{1}{\eta\,p}\left(1 - \tfrac{1}{\delta}\right)\eta_0^2 L^2\left(T\,\sigma^2 + T^2\nu^2\right)\right).$$

*Thus, partial participation effectively reduces the horizon to $R_{\text{eff}} = pR$: the optimization term slows by a factor $1/p$. The pure mini-batch variance averages down as $1/m$, whereas the compression/EF floors depend on $(1 - 1/\delta)$ and the local-work parameter $s = \eta_0 LT$, and do not benefit from $1/m$ averaging.*

**Comparison to EF under PP.** With a diminishing stepsize chosen so that $\sum_{r=0}^{R-1} \eta_r T = \Theta(\sqrt{R})$, the optimization term scales as $\mathcal{O}\big(1/(p\sqrt{R})\big)$, or equivalently as $\mathcal{O}\big(\sqrt{K/m}/\sqrt{R}\big)$, which recovers the $\sqrt{K/m}$ slow-down of EF under partial participation (Li & Li, 2023, Theorem 4.10). Our variance terms likewise exhibit a $1/m$ reduction for mini-batch noise, while the compressor- and heterogeneity-induced floors remain of order $(1 - 1/\delta)$, in agreement with prior analyses.

**Why step-ahead helps (and how much).** Step-ahead modifies the residual contraction factor to

$$\rho_r = (1-\tfrac{1}{\delta})\Big(2(1-\alpha_r)^2 + 24\,\alpha_r^2(\eta_r LT)^2\Big) = (1-\tfrac{1}{\delta})\Big(2 - 4\alpha_r + (2+24s_r^2)\alpha_r^2\Big), \quad s_r := \eta_r LT \leq \tfrac{1}{8}.$$

This quadratic is minimized at $\alpha_r^\star = \frac{1}{1+12s_r^2} \in (0.84, 1]$, hence a *moderate-to-large* step-ahead (close to 1 when $s_r$ is small) achieves the strongest contraction. Relative to EF ($\alpha_r = 0$), the contraction gain is $\rho_r - \rho_{\text{EF}} = (1 - \tfrac{1}{\delta})[-4\alpha_r + (2 + 24s_r^2)\alpha_r^2] < 0$ for $\alpha_r \in (0, \frac{1}{1+12s_r^2})$, with minimum value $\rho_{\min} = 2(1 - \tfrac{1}{\delta})(1 - \frac{1}{1+12s_r^2}) = \rho_{\text{EF}}(1 - \frac{1}{1+12s_r^2})$. For $\alpha_r > \alpha_r^\star$, $\rho_r$ increases (since $\alpha_r^\star < 1$), though the descent coupling in $(\eta, \eta_r)$ is unchanged. The trade-off is that larger $\alpha_r$ tightens the requirement on $s_r = \eta_r LT$ (via the $\alpha_r^2 s_r^2$ term) or necessitates milder compression (larger $\delta$) to maintain $\rho_r < 1$. Overall, step-ahead reduces the contraction factor while leaving the leading optimization rate and the qualitative variance/heterogeneity terms unchanged.

*Practical takeaway.* When $s_r = \eta_r LT$ is small and $\alpha_r$ is chosen near $\alpha_r^\star$, the residual contraction factor is strictly smaller than in EF (cf. Prop. 1), resulting in smaller constants in the residual–induced error terms. This leads to a steeper *initial* decrease in the objective and gradient norm under the same stepsizes, which in turn yields faster convergence within a fixed communication budget. In regimes with high compression or strong data heterogeneity, SA-PEF can therefore outperform both standard EF and uncompressed Local-SGD. To place SA-PEF in context, we provide a brief comparison of Fed-EF, SAEF, CSER, SCAFCOM, and SA-PEF in Table 1, highlighting that SA-PEF operates in the same FL regime as Fed-EF but achieves strictly better residual contraction under biased compression, while SCAFCOM relaxes heterogeneity assumptions at the cost of additional control-variates state.

## 5 EXPERIMENTS

### 5.1 EXPERIMENTAL SETUP

We evaluate SA-PEF on three image classification benchmarks of increasing difficulty and scale. We use CIFAR-10 (Krizhevsky et al., 2009) with ResNet-9 (Page, 2024), CIFAR-100 Krizhevsky et al. (2009) with ResNet-18 He et al. (2016), and Tiny-ImageNet (Le & Yang, 2015) with ResNet-34 (He et al., 2016), trained with cross-entropy loss. We apply standard preprocessing: per-dataset mean/std normalization. We create $K = 100$ clients and adopt *partial participation* with rate $p \in \{0.1, 0.5, 1.0\}$ where, in each round $r$, the server samples $m = \lfloor pK \rfloor$ clients uniformly without replacement. To induce client data heterogeneity, we apply *Dirichlet* label partitioning with concentration parameter $\gamma \in \{0.1, 0.5, 1.0\}$, where smaller $\gamma$ indicates stronger non-IID (Hsu et al., 2019a). Each client's local dataset remains fixed across rounds. Each selected client performs $T = 5$ local SGD steps per round. Training runs for $R = 200$ communication rounds. Unless otherwise stated, the local mini-batch size is 64, momentum is 0.9, and weight decay is $5 \times 10^{-4}$ on CIFAR and $1 \times 10^{-4}$ on Tiny-ImageNet. We use Top-$k$ sparsification with sparsity level $k/d \in \{0.01, 0.05, 0.1\}$. Clients transmit both indices and values of selected entries. We compare SA-PEF with uncompressed LocalSGD, EF (Li & Li, 2023), SAEF (Xu et al., 2021), and CSER (Xie et al., 2020). All methods use the same client sampling, optimizer, learning-rate schedules, and total communication budget (rounds and bits). We report Top-1 accuracy versus rounds and communicated bits, rounds-to-target accuracy, and final accuracy at a fixed communication budget. We repeat all experiments with five random seeds and report mean values in the plots. For CIFAR-10 and CIFAR-100, we additionally report the final test accuracy as mean $\pm$ standard deviation over these five runs in Table 2.

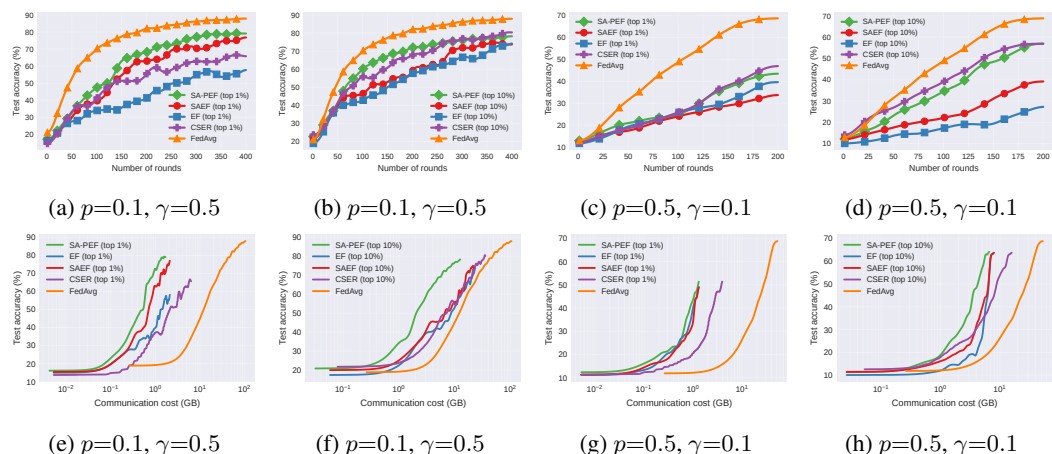

(a) $p$=0.1, $\gamma$=0.5 (b) $p$=0.1, $\gamma$=0.5 (c) $p$=0.5, $\gamma$=0.1 (d) $p$=0.5, $\gamma$=0.1

(e) $p$=0.1, $\gamma$=0.5 (f) $p$=0.1, $\gamma$=0.5 (g) $p$=0.5, $\gamma$=0.1 (h) $p$=0.5, $\gamma$=0.1

Figure 1: Test accuracy vs number of rounds (row 1) and communicated GB (row 2) on the CIFAR-10 dataset using ResNet-9.

Due to space constraints, we present results under high compression, with comprehensive results across all settings in the Appendix.

## 5.2 Empirical results

Figure 1 compares SA-PEF with FedAvg (dense) and compressed baselines (EF, SAEF, CSER) on CIFAR-10 with ResNet-9 under two participation rates ($p \in \{0.1, 0.5\}$) and two compression budgets (Top-1%, Top-10%). In the accuracy versus rounds plots (top row), SA-PEF generally reaches a given accuracy in fewer rounds than EF and SAEF, with the largest margin in the harder regime ($\gamma$=0.1, Top-1%). SAEF often shows an initial jump but tends to plateau, while SA-PEF continues to improve. CSER typically lags early and only catches up later. In the accuracy versus communication plots (bottom row), SA-PEF's curves are left-shifted: for the same test accuracy it requires less uplink communication (in Gigabyte) than EF or SAEF, whereas FedAvg attains high accuracy only at orders of magnitude higher cost. Raising participation to $p$=0.5 benefits all approaches and narrows round-wise gaps, but SA-PEF remains the most communication-efficient across schemes.[1]

Figure 2 shows results on CIFAR-100 with ResNet-18. Despite the increased task difficulty, the same qualitative trends persist: SA-PEF tends to dominate early rounds and delivers higher accuracy per unit of communication across most regimes, SAEF often plateaus early, and CSER improves mainly at larger communication budgets.[2] The gains are most pronounced under aggressive compression (Top-1%) and low participation (e.g., $p$=0.1). Overall, these results suggest that combining pre-view with partial error feedback provides faster early progress and superior accuracy-communication trade-offs across architectures and datasets.

**Discussion:** Overall, our results position SA-PEF as a lightweight but effective upgrade of classical EF in FL. Compared to EF and its step-ahead variant SAEF, SA-PEF converges consistently faster and offers better accuracy-communication trade-offs under practical settings. Relative to CSER, which periodically resets residuals to control mismatch, SA-PEF achieves comparable or better robustness without introducing any additional reset-period hyperparameter, and it avoids CSER's high *peak* communication cost when compressed or full residuals are transmitted at reset rounds. Since practical systems must provision for peak, rather than average, bandwidth and latency, this makes SA-PEF more attractive as a drop-in component in resource-constrained deploy-

---

[1]Under low participation (e.g., 1-10%), effective batch sizes shrink and both drift and compression noise increase, hence participation-aware hyperparameters (e.g., learning rate, local steps $T$, or $\alpha_r$) may need tuning. Here, we fix hyperparameters across methods for fairness, which can reduce the observable advantage of SA-PEF in extreme low-participation regimes.

[2]As in CIFAR-10, under low partial participation (1–10%), the differences between methods may be less visible without participation-aware tuning of hyperparameters.

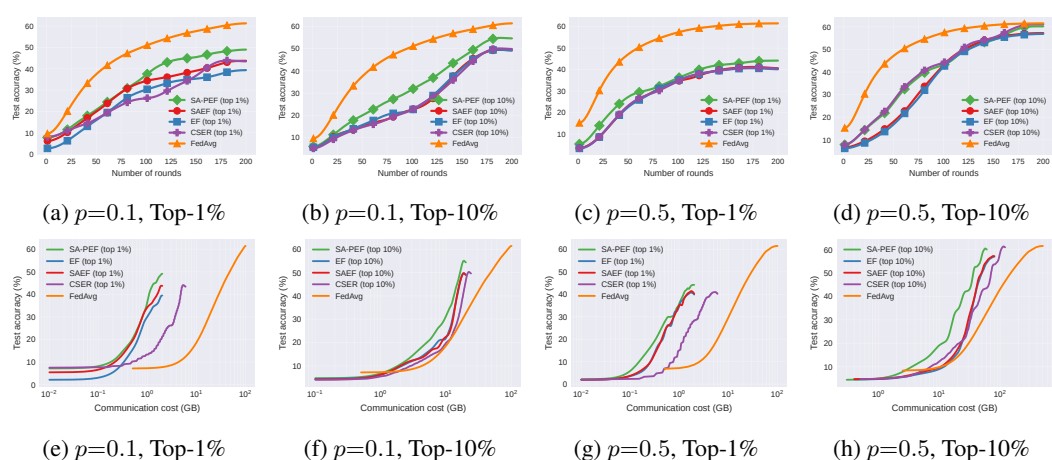

(a) $p$=0.1, Top-1%  (b) $p$=0.1, Top-10%  (c) $p$=0.5, Top-1%  (d) $p$=0.5, Top-10%

(e) $p$=0.1, Top-1%  (f) $p$=0.1, Top-10%  (g) $p$=0.5, Top-1%  (h) $p$=0.5, Top-10%

Figure 2: Test accuracy vs number of rounds (row 1) and communicated GB (row 2) on the CIFAR-100 dataset using ResNet-18 and $\gamma$=0.1.

ments. Control-variate methods such as SCAFFOLD and SCAFCOM target a complementary axis, mitigating client drift and heterogeneity via additional per-client state, whereas SA-PEF focuses on reducing compression-induced residual mismatch within the EF family. In this sense, SA-PEF is best viewed as a drop-in improvement for EF-style compressed FL (and a natural building block for future combinations with control variates), providing significant gains in challenging regimes with high compression and strong heterogeneity.

## 5.3 SENSITIVITY ANALYSIS OF STEP-AHEAD COEFFICIENT $\alpha$

To assess the robustness of SA-PEF to the choice of step-ahead coefficient, we sweep $\alpha_r$ between zero and one with increment of $0.1$ on CIFAR-10 with ResNet-9 and CIFAR-100 with ResNet-18 under non-IID Dirichlet partitioning ($\gamma = 0.1$), Top-1% sparsification, $T = 5$ local steps, and $p = 0.1$ participation. Figure 3 reports test accuracy versus rounds for EF ($\alpha_r = 0$), SAEF ($\alpha_r = 1$), and SA-PEF with intermediate $\alpha_r$ values. Three regimes emerge: (i) *Small $\alpha$* ($\alpha_r \leq 0.3$) behaves similarly to EF, with noticeably slower convergence and lower final accuracy. (ii) *Intermediate $\alpha$* ($\alpha_r \in [0.6, 0.9]$) produces nearly identical curves, yielding the fastest convergence and highest final accuracy. This interval includes the default $\alpha_r = 0.85$ used in our main experiments. (iii) *Full step-ahead* ($\alpha_r = 1.0$, SAEF) accelerates early rounds but plateaus slightly below the best SA-PEF setting in the later phases. Overall, SA-PEF is robust across a broad high-$\alpha$ region, while performance is significantly affected only at the extremes: $\alpha_r \approx 0$ (reducing to EF) or $\alpha_r = 1$ (SAEF). This supports treating $\alpha$ as a momentum-like parameter and use a single default value (e.g., $\alpha_r \approx 0.8$-$0.9$) across tasks without heavy tuning.

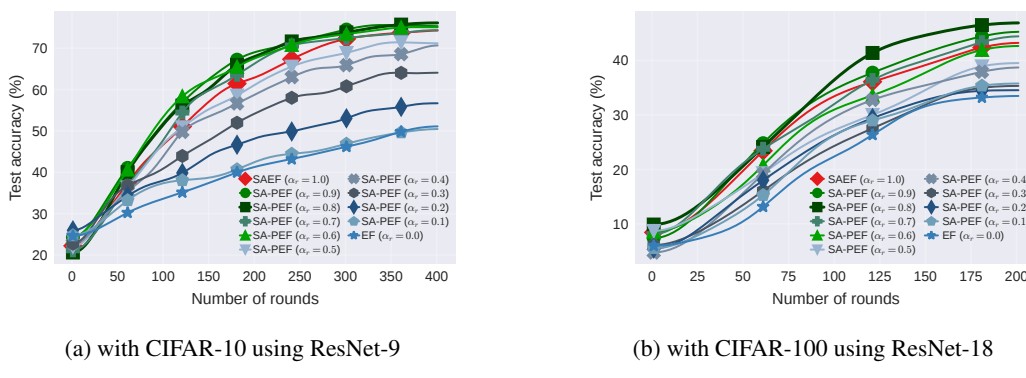

(a) with CIFAR-10 using ResNet-9  (b) with CIFAR-100 using ResNet-18

Figure 3: Sensitivity analysis of step-ahead coefficient $\alpha$.

## 5.4 GRADIENT MISMATCH

Let $w \in \mathbb{R}^d$ stack all trainable parameters, and let $f(\cdot; w)$ be the network in *evaluation* mode (dropout disabled, BatchNorm with frozen statistics). For a mini-batch $S = \{(x_i, y_i)\}_{i=1}^b$ drawn from a held-out loader, define the batch loss as $\mathcal{L}_S(w) = \frac{1}{b} \sum_{(x,y) \in S} \ell(f(x; w), y)$. At round $r$, for client $k$ and $\alpha \in [0, 1]$, we consider two evaluation points $w_r$ and $w_r^{(\alpha,k)} = w_r - \alpha \, e_r^{(k)}$. Using the same mini-batch $S$ for both, we compute the associated gradients as $g_r = \nabla_w \mathcal{L}_S(w_r)$ and $g_r^{(\alpha,k)} = \nabla_w \mathcal{L}_S(w_r^{(\alpha,k)})$, and define the squared gradient-mismatch as $\hat{\varepsilon}_r^{(k)}(\alpha) = \left\| g_r - g_r^{(\alpha,k)} \right\|_2^2$ and its client average as $\hat{\varepsilon}_r(\alpha) = \frac{1}{K} \sum_{k=1}^K \hat{\varepsilon}_r^{(k)}(\alpha)$. In particular, we switch the model to `eval` mode, clear any stale gradients, reuse the same $S$, compute $g_r$ and $g_r^{(\alpha,k)}$ via first-order autodiff, and then restore the model to training mode. We do not alter any optimizer state or BatchNorm buffer.

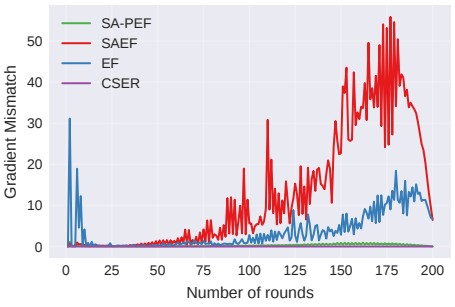

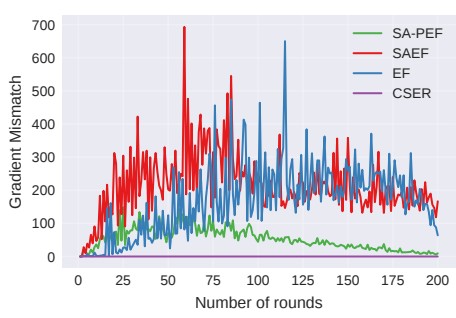

(a) with CIFAR-10 using ResNet-9                (b) with CIFAR-100 using ResNet-18

Figure 4: Gradient mismatch for different algorithms.

Figure 4 shows the gradient-mismatch probe across training rounds for all methods. SA-PEF keeps the mismatch essentially flat and near zero throughout training: previewing only a fraction of the residual shifts the evaluation point closer to where EF applies the update, while the retained $(1-\alpha)e_r$ further prevents residual build-up. EF exhibits a steady late-phase rise, consistent with its one-step staleness effect as displacement accumulates over rounds. SAEF produces the largest spikes and highest overall mismatch, as with full preview ($\alpha=1$), the evaluation occurs at $w_r - e_r$. Large or heterogeneous residuals spike mismatch, causing late-stage plateaus. CSER stays near zero, reflecting its error-reset/averaging mechanism that suppresses persistent residual drift. Overall, the results confirm that combining partial preview with partial EF controls mismatch throughout training, unlike plain EF or SAEF.

## 6 CONCLUSION

We presented SA-PEF, which combines a step-ahead preview with partial error feedback to accelerate early-round progress in FL while preserving the long-run stability of EF under biased compression. Our theoretical analysis covers non-convex objectives, local SGD, partial participation, and $\delta$-contractive compressors, and establishes convergence to stationarity with rates matching non-convex FedSGD up to constants. To our knowledge, this is the first analysis of SAEF in a federated setting that simultaneously accounts for local updates, non-IID data, partial participation, and biased compression. Empirically, across datasets, models, and compressors, SA-PEF consistently reaches target accuracy in fewer rounds than EF and avoids the late-stage plateaus observed with full step-ahead. Looking ahead, a promising direction is developing *adaptive schedules for* $\alpha$, guided by online estimates of noise, residual norms, or client drift, and analyzing the feedback between mismatch reduction and residual dynamics under partial participation. Another direction is integrating SA-PEF with *adaptive optimizers and momentum*, including a preconditioned drift analysis to control momentum-residual interactions. Finally, a natural direction for future work is to bring ideas from EF21 into the federated local-SGD regime we study. We expect these extensions to further improve early-phase alignment while retaining stability at scale, advancing communication-efficient FL under realistic constraints.

**Ethics statement.** This work studies communication-efficient federated optimization using public, non-sensitive benchmark datasets (CIFAR-10/100 and Tiny-ImageNet). No personal, identifiable, or protected-class attributes are used, and no user-generated private data was accessed. All datasets are widely used for research and were obtained under their respective licenses; we cite the original sources. Our algorithms are intended to *reduce* communication and potentially lower the energy costs of federated training.

**Reproducibility statement.** We provide a zip file with anonymized code and configuration files. The package includes: (i) exact configs for all experiments; (ii) scripts to download datasets and to create federated partitions (IID and Dirichlet with $\alpha \in \{0.1, 0.5\}$); (iii) model definitions (ResNet-9/18/34), compressor implementations (Top-$k$ with $k/d \in \{1\%, 10\%\}$, error-feedback variants), and our SA-PEF scheduler. Unless otherwise stated, experiments use client participation $q \in \{0.1, 0.5\}$, local epochs $T \in \{1, \ldots, 5\}$, batch size $B = 64$, optimizer SGD (momentum 0.9), weight decay $5 \times 10^{-4}$, and base step size $\eta_0 = 0.1$ with cosine or step decay. Communication is measured as uplink GB, including indices and values for sparse updates. Hardware: runs were executed on A100/A5000/H200 GPUs.

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

## A  PROOF OF CONVERGENCE RESULTS

### A.1  PROOF OF THEOREM 1

*Proof.* Subtracting the *average residual* from the communicated model converts a biased, compressed update into something that resembles vanilla SGD.

We have:

1. **Step-ahead shift** $w_{r+\frac{1}{2}}^{(k)} = w_r^{(k)} - \alpha_r e_r^{(k)}$.

2. **Residual update & compression** $e_{r+1}^{(k)} = u_{r+1}^{(k)} - C(u_{r+1}^{(k)})$ with
   $u_{r+1}^{(k)} = (1 - \alpha_r)e_r^{(k)} + g_r^{(k)}$.

Let

$$\bar{e}_r := \frac{1}{K}\sum_{k=1}^{K} e_r^{(k)}, \quad \bar{u}_{r+1} := (1-\alpha_r)\bar{e}_r + \bar{g}_r, \quad \bar{C}_{r+1} := \frac{1}{K}\sum_{k} C(u_{r+1}^{(k)}),$$

so the server update is $w_{r+1} = w_r - \eta\,\bar{C}_{r+1}$.

Define a virtual iterate:

$$\begin{aligned}
x_r &= w_r - \eta\,\bar{e}_r \\
x_{r+1} &= w_{r+1} - \eta\,\bar{e}_{r+1} \\
&= (w_r - \eta\,\bar{C}_{r+1}) - \eta\,(\bar{u}_{r+1} - \bar{C}_{r+1}) \\
&= w_r - \eta\,\bar{u}_{r+1} \\
&= (x_r + \eta\,\bar{e}_r) - \eta\,[(1-\alpha_r)\bar{e}_r + \bar{g}_r] \\
&= x_r + \eta\,\alpha_r\bar{e}_r - \eta\,\bar{g}_r.
\end{aligned}$$

$$f(x_{r+1}) \leq f(x_r) + \langle \nabla f(x_r),\, x_{r+1} - x_r \rangle + \frac{L}{2}\left\| x_{r+1} - x_r \right\|^2. \tag{2}$$

Because $x_{r+1} - x_r = \eta(\alpha_r\bar{e}_r - \bar{g}_r)$, taking the expectation over all randomness inside round $r$ yields

$$\mathbb{E}[f(x_{r+1})] \leq \mathbb{E}[f(x_r)] + \eta\,\mathbb{E}[\langle \nabla f(x_r),\, \alpha_r\bar{e}_r - \bar{g}_r \rangle] + \frac{\eta^2 L}{2}\,\mathbb{E}[\|\alpha_r\bar{e}_r - \bar{g}_r\|^2]. \tag{3}$$

Isolating the drift of the objective between two consecutive rounds and decomposing the inner product gives

$$\begin{aligned}
\mathbb{E}[f(x_{r+1})] - \mathbb{E}[f(x_r)] &\leq \eta\,\mathbb{E}[\langle \nabla f(x_r),\, \alpha_r\bar{e}_r - \bar{g}_r \rangle] + \frac{\eta^2 L}{2}\,\mathbb{E}[\|\alpha_r\bar{e}_r - \bar{g}_r\|^2] \\
&= \underbrace{\eta\,\mathbb{E}[\langle \nabla f(w_r),\, \alpha_r\bar{e}_r - \bar{g}_r \rangle]}_{T1} + \underbrace{\frac{\eta^2 L}{2}\,\mathbb{E}[\|\alpha_r\bar{e}_r - \bar{g}_r\|^2]}_{T2} \\
&\quad + \underbrace{\eta\,\mathbb{E}[\langle \nabla f(x_r) - \nabla f(w_r),\, \alpha_r\bar{e}_r - \bar{g}_r \rangle]}_{T3}. \tag{4}
\end{aligned}$$

**Bounding the first inner-product term**   Recall

$$T_1 = \eta\,\mathbb{E}_r[\langle \nabla f(w_r),\, \alpha_r\bar{e}_r - \bar{g}_r \rangle] = \eta\,\alpha_r\,\langle \nabla f(w_r), \bar{e}_r \rangle - \eta\,\mathbb{E}_r[\langle \nabla f(w_r), \bar{g}_r \rangle],$$

where $\bar{g}_r = \frac{\eta_r}{K}\sum_{k=1}^{K}\sum_{t=0}^{T-1}(\nabla f_k(w_{r+\frac{1}{2},t}^{(k)}) + \xi_{k,t})$ and $\mathbb{E}_r[\xi_{k,t}] = 0$.

Define $\bar{\nabla}_t := \frac{1}{K}\sum_{k=1}^{K}\nabla f_k(w_{r+\frac{1}{2},t}^{(k)})$ and $\Delta_t = \frac{1}{K}\sum_{k=1}^{K}(\nabla f_k(w_{r+\frac{1}{2},t}^{(k)}) - \nabla f(w_r))$

Write $\bar{\nabla}_t = \nabla f(w_r) + \Delta_t$ and use $L$-smoothness to get $\mathbb{E}_r \|\Delta_t\|^2 \le L^2 \times \frac{1}{K} \sum_k \mathbb{E}_r \|w^{(k)}_{r+\frac{1}{2},t} - w_r\|^2$.
Young's inequality gives, for any $\lambda_1 > 0$,

$$\mathbb{E}_r \langle \nabla f(w_r), \bar{g}_r \rangle \ge \eta_r T \Big(1 - \frac{\lambda_1}{2}\Big) \|\nabla f(w_r)\|^2 - \frac{\eta_r L^2}{2\lambda_1} S, \quad S := \frac{1}{K} \sum_{k=1}^{K} \sum_{t=0}^{T-1} \mathbb{E}_r \|w^{(k)}_{r+\frac{1}{2},t} - w_r\|^2.$$

By Lemma 2 (summed over $t$),

$$S \le 12\,\eta_r^2 T^2 \sigma^2 + 168\,\eta_r^2 T^3 \nu^2 + 36\,\eta_r^2 T^3 \beta^2 \|\nabla f(w_r)\|^2 + 3T\alpha_r^2 \bar{E}_r.$$

For the step-ahead piece, Young's inequality yields, for any $\lambda_2 > 0$,

$$\eta\,\alpha_r \langle \nabla f(w_r), \bar{e}_r \rangle \le \frac{\eta \lambda_2}{2} \|\nabla f(w_r)\|^2 + \frac{\eta \alpha_r^2}{2\lambda_2} \bar{E}_r.$$

Combining,

$$T_1 \le \eta\Big(\frac{\lambda_2}{2} - \eta_r T\Big(1 - \frac{\lambda_1}{2}\Big)\Big)\|\nabla f(w_r)\|^2 + \frac{\eta \alpha_r^2}{2\lambda_2} \bar{E}_r + \frac{\eta \eta_r L^2}{2\lambda_1} S.$$

Choose $\lambda_1 = 1$ and $\lambda_2 = \eta_r T/2$; then

$$T_1 \le -\frac{\eta\,\eta_r T}{4} \|\nabla f(w_r)\|^2 + \frac{\eta \alpha_r^2}{\eta_r T} \bar{E}_r + \frac{\eta \eta_r L^2}{2} S,$$

and substituting the bound on $S$ gives

$$T_1 \le \eta\,\eta_r \Big[-\frac{T}{4} + 18\,\eta_r^2 L^2 \beta^2 T^3\Big] \|\nabla f(w_r)\|^2$$
$$+ \eta \alpha_r^2 \Big[\frac{1}{\eta_r T} + \frac{3}{2}\eta_r L^2 T\Big] \bar{E}_r + 6\,\eta\,\eta_r^3 L^2 T^2\,\sigma^2 + 84\,\eta\,\eta_r^3 L^2 T^3\,\nu^2. \tag{5}$$

**Bounding the quadratic (smoothness) term**

Recall the second contribution in the descent inequality

$$T_2 \;=\; \frac{L\eta^2}{2} \mathbb{E}_r\big[\|\alpha_r \bar{e}_r - \bar{g}_r\|^2\big]. \tag{8}$$

**Bounding the gradient–mismatch term**

$$T_3 := \eta\,\mathbb{E}_r\big[\langle \nabla f(x_r) - \nabla f(w_r),\,\alpha_r \bar{e}_r - \bar{g}_r\rangle\big].$$

Since $x_r = w_r - \eta\bar{e}_r$, we have $\|x_r - w_r\| = \eta\|\bar{e}_r\|$, and by $L$-smoothness,

$$\|\nabla f(x_r) - \nabla f(w_r)\| \le L\|x_r - w_r\| = L\eta\|\bar{e}_r\|.$$

Apply Young's inequality (any $\lambda > 0$) with $a = \nabla f(x_r) - \nabla f(w_r)$ and $b = \alpha_r \bar{e}_r - \bar{g}_r$:

$$\big|\langle \nabla f(x_r) - \nabla f(w_r),\,\alpha_r \bar{e}_r - \bar{g}_r\rangle\big| \le \frac{\lambda}{2} L^2 \eta^2 \|\bar{e}_r\|^2 + \frac{1}{2\lambda} \|\alpha_r \bar{e}_r - \bar{g}_r\|^2.$$

Taking $\mathbb{E}_r[\cdot]$, using $\|\bar{e}_r\|^2 \le \bar{E}_r$, and multiplying by $\eta$ yields

$$|T_3| \;\le\; \frac{\lambda L^2 \eta^3}{2} \bar{E}_r \;+\; \frac{\eta}{2\lambda} \mathbb{E}_r\big[\|\alpha_r \bar{e}_r - \bar{g}_r\|^2\big].$$

Choosing $\lambda = \frac{1}{L\eta}$ balances the two terms and $T_3 \le |T_3|$ gives

$$T_3 \le \frac{L\eta^2}{2} \bar{E}_r + \frac{L\eta^2}{2} \mathbb{E}_r\big[\|\alpha_r \bar{e}_r - \bar{g}_r\|^2\big]. \tag{6}$$

Put the pieces together:

$$\mathbb{E}_r[f(x_{r+1}) - f(x_r)] \leq \eta\,\eta_r\Big[-\frac{T}{4} + 18\,\eta_r^2 L^2\beta^2 T^3\Big]\,\|\nabla f(w_r)\|^2$$

$$+ \eta\,\alpha_r^2\Big[\frac{1}{\eta_r T} + \frac{3}{2}\eta_r L^2 T\Big]\bar{E}_r + 6\,\eta\,\eta_r^3 L^2 T^2\,\sigma^2 + 84\,\eta\,\eta_r^3 L^2 T^3\,\nu^2$$

$$+ L\eta^2\,\mathbb{E}_r\big[\,\|\alpha_r\bar{e}_r - \bar{g}_r\|^2\,\big] + \frac{L\eta^2}{2}\,\bar{E}_r \tag{7}$$

Using Lemma 3,

$$\mathbb{E}_r\big[\|\alpha_r\bar{e}_r - \bar{g}_r\|^2\big] \leq 2\alpha_r^2\bar{E}_r + 8\eta_r^2 T^2\|\nabla f(w_r)\|^2\big(1 + 36L^2\eta_r^2 T^2\beta^2\big)$$

$$+ \frac{4\eta_r^2 T}{K}\,\sigma^2 + 96L^2\eta_r^4 T^3\big(\sigma^2 + 14T\nu^2\big) + 24\alpha_r^2\eta_r^2 L^2 T^2\bar{E}_r.$$

$$\mathbb{E}_r[f(x_{r+1}) - f(x_r)] \leq \Big[\eta\,\eta_r\Big(-\tfrac{T}{4} + 18\,\eta_r^2 L^2\beta^2 T^3\Big) + L\eta^2\Big(8\eta_r^2 T^2 + 288\,L^2\eta_r^4 T^4\beta^2\Big)\Big]\|\nabla f(w_r)\|^2$$

$$+ \Big[\eta\,\alpha_r^2\Big(\tfrac{1}{\eta_r T} + \tfrac{3}{2}\eta_r L^2 T\Big) + L\eta^2\Big(2\alpha_r^2 + 24\,\alpha_r^2\eta_r^2 L^2 T^2\Big) + \tfrac{L\eta^2}{2}\Big]\bar{E}_r,$$

$$+ \Big[6\,\eta\,\eta_r^3 L^2 T^2 + \tfrac{4L\eta^2\eta_r^2 T}{K} + 96\,L^3\eta^2\eta_r^4 T^3\Big]\sigma^2$$

$$+ \Big[84\,\eta\,\eta_r^3 L^2 T^3 + 1344\,L^3\eta^2\eta_r^4 T^4\Big]\nu^2 \tag{8}$$

**Telescoping.** Summing over $r = 0, \ldots, R-1$ and taking total expectation,

$$\mathbb{E}[f(x_R) - f(x_0)] \leq \sum_{r=0}^{R-1}\Big\{A_r\,\mathbb{E}\|\nabla f(w_r)\|^2 + E_r\,\mathbb{E}[\bar{E}_r] + V_r\,\sigma^2 + H_r\,\nu^2\Big\}, \tag{9}$$

where

$$A_r := \eta\,\eta_r\Big(-\tfrac{T}{4} + 18\,\eta_r^2 L^2\beta^2 T^3\Big) + L\eta^2\Big(8\eta_r^2 T^2 + 288\,L^2\eta_r^4 T^4\beta^2\Big),$$

$$E_r := \eta\,\alpha_r^2\Big(\tfrac{1}{\eta_r T} + \tfrac{3}{2}\eta_r L^2 T\Big) + L\eta^2\Big(2\alpha_r^2 + 24\,\alpha_r^2\eta_r^2 L^2 T^2\Big) + \tfrac{L\eta^2}{2},$$

$$V_r := 6\,\eta\,\eta_r^3 L^2 T^2 + \tfrac{4L\eta^2\eta_r^2 T}{K} + 96\,L^3\eta^2\eta_r^4 T^3,$$

$$H_r := 84\,\eta\,\eta_r^3 L^2 T^3 + 1344\,L^3\eta^2\eta_r^4 T^4.$$

We now derive the main convergence guarantee. The analysis begins with the telescoped recursion from equation 9, which provides an upper bound on the function value progress, $\mathbb{E}[f(x_R) - f(x_0)]$. The key step is to ensure that the coefficient $A_r$ of the squared gradient norm is negative, which guarantees descent. The following lemma establishes sufficient conditions for this.

**Lemma 1** (Sufficient Conditions for Descent)**.** *Let $s_r := \eta_r LT$. Assume that for all rounds $r$, the parameters satisfy $s_r \leq 1/8$ and the following two conditions:*

$$18\,\beta^2\,s_r^2 \leq \frac{1}{8}, \tag{10}$$

$$\eta \leq \frac{1}{256\,\beta^2\,L\,\eta_r T}, \tag{11}$$

*where it is assumed that $\beta^2 \geq 1$. Then, the coefficient $A_r$ is bounded as*

$$A_r \leq -\frac{1}{16}\,\eta\,\eta_r T.$$

*Proof.* The coefficient $A_r$ is defined as

$$A_r = -\frac{\eta\eta_r T}{4} + 18\eta\eta_r^3 L^2\beta^2 T^3 + L\eta^2(8\eta_r^2 T^2 + 288L^2\eta_r^4 T^4\beta^2).$$

We proceed by bounding the two positive terms separately.

First, we bound the term $18\eta\eta_r^3 L^2\beta^2 T^3$. By applying the definition $s_r = \eta_r LT$ and condition equation 10, we obtain

$$18\eta\eta_r^3 L^2\beta^2 T^3 = (\eta\eta_r T) \times (18\beta^2\eta_r^2 L^2 T^2)$$
$$= (\eta\eta_r T) \times (18\beta^2 s_r^2) \leq \frac{1}{8}\eta\eta_r T.$$

Next, we bound the term $L\eta^2(8\eta_r^2 T^2 + 288L^2\eta_r^4 T^4\beta^2)$. Condition equation 10 implies $36\beta^2 s_r^2 \leq 1/4$. This allows us to simplify the expression within parentheses:

$$8\eta_r^2 T^2 + 288L^2\eta_r^4 T^4\beta^2 = 8\eta_r^2 T^2\left(1 + 36\beta^2 L^2\eta_r^2 T^2\right)$$
$$= 8\eta_r^2 T^2(1 + 36\beta^2 s_r^2)$$
$$\leq 8\eta_r^2 T^2\left(1 + \frac{1}{4}\right) = 10\eta_r^2 T^2.$$

Using this intermediate result and condition equation 11, we bound the full term. The assumption $\beta^2 \geq 1$ ensures that $256\beta^2 \geq 160$.

$$L\eta^2(8\eta_r^2 T^2 + 288L^2\eta_r^4 T^4\beta^2) \leq L\eta^2(10\eta_r^2 T^2)$$
$$= (10L\eta\eta_r^2 T^2) \times \eta$$
$$\leq 10L\eta\eta_r^2 T^2 \times \left(\frac{1}{256\beta^2 L\eta_r T}\right)$$
$$= \frac{10}{256\beta^2}\eta\eta_r T \leq \frac{10}{160}\eta\eta_r T = \frac{1}{16}\eta\eta_r T.$$

Finally, substituting these bounds back into the expression for $A_r$ yields the desired result:

$$A_r \leq -\frac{\eta\eta_r T}{4} + \frac{1}{8}\eta\eta_r T + \frac{1}{16}\eta\eta_r T$$
$$= \left(-\frac{4}{16} + \frac{2}{16} + \frac{1}{16}\right)\eta\eta_r T$$
$$= -\frac{1}{16}\eta\eta_r T.$$

This completes the proof. $\qquad\square$

### A.1.1 GENERAL CONVERGENCE RATE FOR DECAYING STEP-SIZES

Applying the bound on $A_r$ from Lemma 1 to the main inequality equation 9 yields:

$$\mathbb{E}[f(x_R) - f(x_0)] \leq \sum_{r=0}^{R-1}\left(-\frac{\eta\eta_r T}{16}\mathbb{E}\|\nabla f(w_r)\|^2 + E_r\,\mathbb{E}[\bar{E}_r] + V_r\,\sigma^2 + H_r\,\nu^2\right).$$

Rearranging the terms to isolate the sum of squared gradients and assuming the function is bounded below by $f_\star := \inf_x f(x)$, we obtain a bound on the weighted sum:

$$\sum_{r=0}^{R-1}\eta_r T\,\mathbb{E}\|\nabla f(w_r)\|^2 \leq \frac{16}{\eta}(f(x_0) - f_\star) + \frac{16}{\eta}\sum_{r=0}^{R-1}\mathbb{E}[E_r\,\bar{E}_r + V_r\,\sigma^2 + H_r\,\nu^2]. \qquad (12)$$

The main challenge is to bound the term involving the residual energy, $\sum_r E_r\mathbb{E}[\bar{E}_r]$. We first decouple the coefficient using the supremum $\mathcal{E}_{\max} := \sup_r E_r$. Then, we apply the bound on the

sum of residuals from Lemma 4, which is derived by unrolling the per-round recursion for $\bar{E}_r$:

$$\frac{16}{\eta} \sum_{r=0}^{R-1} E_r \mathbb{E}[\bar{E}_r] \le \frac{16\mathcal{E}_{\max}}{\eta} \sum_{r=0}^{R-1} \mathbb{E}[\bar{E}_r]$$

$$\le \frac{16\mathcal{E}_{\max}}{\eta(1-\rho_{\max})} \bar{E}_0 + \frac{16\mathcal{E}_{\max}(1-1/\delta)}{\eta(1-\rho_{\max})} \sum_{r=0}^{R-1} \mathbb{E}[B_r^{(\nabla)} + B_r^{(\nu,\sigma)}]$$

$$\stackrel{(a)}{\le} \frac{16\mathcal{E}_{\max}(1-1/\delta)}{\eta(1-\rho_{\max})} \sum_{r=0}^{R-1} \mathbb{E}[B_r^{(\nabla)} + B_r^{(\nu,\sigma)}]$$

where (a) uses $\bar{E}_0 = 0$ since $e_0^{(k)} \equiv 0$.

Substituting the residual recursion (Lemma 4) into equation 12 introduces forcing terms. Since $B_r^{(\nabla)}$ is proportional to $\|\nabla f(w_r)\|^2$, it produces a contribution of the form $\Theta \sum_{r=0}^{R-1} \eta_r T \, \mathbb{E}\|\nabla f(w_r)\|^2$, where we *define*

$$\Theta := \frac{16}{\eta} \times \frac{\mathcal{E}_{\max}}{1-\rho_{\max}} \left(1 - \frac{1}{\delta}\right) \beta^2 \sup_{0 \le r < R} \frac{8\,\eta_r^2 T^2 + 288\,L^2\eta_r^4 T^4}{\eta_r T}$$

$$= \frac{16}{\eta} \times \frac{\mathcal{E}_{\max}}{1-\rho_{\max}} \left(1 - \frac{1}{\delta}\right) \beta^2 \sup_r \left(8\,\eta_r T + 288\,L^2\eta_r^3 T^3\right).$$

With this notation, the weighted inequality becomes

$$(1-\Theta) \sum_{r=0}^{R-1} \eta_r T \, \mathbb{E}\|\nabla f(w_r)\|^2$$

$$\le \frac{16}{\eta} \left[ (f(x_0) - f_\star) + \frac{\mathcal{E}_{\max}}{1-\rho_{\max}} \left(1 - \frac{1}{\delta}\right) \sum_{r=0}^{R-1} \mathbb{E}B_r^{(\nu,\sigma)} + \sum_{r=0}^{R-1} V_r\,\sigma^2 + \sum_{r=0}^{R-1} H_r\,\nu^2 \right].$$

$$(13)$$

We now assume a small-steps/compression regime that ensures $\Theta \le \frac{1}{2}$. We can therefore move this term to the left-hand side and absorb it, which tightens the overall bound at the cost of multiplying the remaining terms on the right-hand side by a factor of 2.

To obtain a bound on the conventional average, compare it to the weighted sum with weights $q_r := \eta_r T > 0$. Let $q_{\min} := \min_{0 \le r < R} q_r$, $S_R := \sum_{r=0}^{R-1} q_r$, and $\phi_R := \dfrac{S_R}{R\,q_{\min}} \ge 1$. Then, for nonnegative terms,

$$\frac{1}{R} \sum_{r=0}^{R-1} \mathbb{E}\|\nabla f(w_r)\|^2 \le \frac{S_R}{R\,w_{\min}} \times \frac{1}{S_R} \sum_{r=0}^{R-1} w_r \, \mathbb{E}\|\nabla f(w_r)\|^2 \qquad (14)$$

$$= \phi_R \times \frac{1}{S_R} \sum_{r=0}^{R-1} \eta_r T \, \mathbb{E}\|\nabla f(w_r)\|^2. \qquad (15)$$

Combining this with the absorbed bound and noting $x_0 = w_0$ since $e_0^{(k)} \equiv 0$, we obtain

$$\frac{1}{R} \sum_{r=0}^{R-1} \mathbb{E}\|\nabla f(w_r)\|^2 \le \phi_R \frac{32}{\eta\,S_R} \left[ (f(w_0) - f_\star) + \sum_{r=0}^{R-1} V_r\,\sigma^2 + \sum_{r=0}^{R-1} H_r\,\nu^2 \right.$$

$$+ \frac{\mathcal{E}_{\max}}{1-\rho_{\max}} \left(1 - \frac{1}{\delta}\right) \sum_{r=0}^{R-1} \left(4\,\eta_r^2 T + 96\,L^2\eta_r^4 T^3\right)\sigma^2$$

$$\left. + \frac{\mathcal{E}_{\max}}{1-\rho_{\max}} \left(1 - \frac{1}{\delta}\right) \sum_{r=0}^{R-1} \left(8\,\eta_r^2 T^2 + 1344\,L^2\eta_r^4 T^4\right)\nu^2 \right].$$

### A.1.2 Convergence Rate for a Constant Step-Size

In the simpler setting where the inner learning rate is constant, $\eta_r \equiv \eta_0$ for all $r$, the sum of weights is $S_R = R\eta_0 T$. Under the descent and absorption conditions, we obtain

$$\frac{1}{R}\sum_{r=0}^{R-1} \mathbb{E}\|\nabla f(w_r)\|^2 \leq \frac{32\,(f(w_0)-f^\star)}{\eta\,\eta_0 T\,R} + \left(1-\tfrac{1}{\delta}\right)\left[C_\sigma\,\eta_0^2 L^2 T\,\sigma^2 + C_\nu\,\eta_0^2 L^2 T^2\,\nu^2\right]$$
$$+ \frac{128\,L\,\eta_0}{K}\,\sigma^2$$

with

$$C_\sigma = \frac{32}{\eta}\left[6\,\eta\,\eta_0^2 L^2 T + 96\,L^3\eta\,\eta_0^3 T^2\right] + \frac{32}{\eta}\times\frac{\mathcal{E}_{\max}}{1-\rho_{\max}}\left[4\,\eta_0 + 96\,L^2\eta_0^3 T^2\right],$$
$$C_\nu = \frac{32}{\eta}\left[84\,\eta\,\eta_0^2 L^2 T^2 + 1344\,L^3\eta\,\eta_0^3 T^3\right] + \frac{32}{\eta}\times\frac{\mathcal{E}_{\max}}{1-\rho_{\max}}\left[8\,\eta_0 T + 1344\,L^2\eta_0^3 T^3\right].$$

which proves the theorem. $\qquad\square$

### A.2 Lemmas

**Lemma 2** (Local-model drift under SA-PEF). *Fix a communication round $r$ and an inner-loop horizon $T \geq 1$. Assume the step-size condition $0 < \eta_r \leq \frac{1}{8LT}$ and Assumptions 1–3.*

*Then, for every local step $t \in \{0, \ldots, T-1\}$,*

$$\frac{1}{K}\sum_{k=1}^{K}\mathbb{E}_r\left[\|w_{r+\frac{1}{2},t}^{(k)} - w_r\|^2\right] \leq 12\,\eta_r^2 T\,\sigma^2 + 168\,\eta_r^2 T^2\,\nu^2 + 36\,\eta_r^2 T^2\,\beta^2\,\|\nabla f(w_r)\|^2$$

$$+ 3\,\alpha_r^2\times\frac{1}{K}\sum_{k=1}^{K}\mathbb{E}_r\|e_r^{(k)}\|^2. \tag{16}$$

*Proof.* All expectations are conditional on the randomness inside round $r$.

Set $u_{k,t} := w_{r+\frac{1}{2},t}^{(k)} - w_r$ $(u_{k,0} = -\alpha_r e_r^{(k)})$. Client $k$ updates by

$$u_{k,t+1} = u_{k,t} - \eta_r\left(\nabla f_k(w_{r+\frac{1}{2},t}^{(k)}) + \xi_{k,t}\right), \qquad \mathbb{E}_r[\xi_{k,t}] = 0,\ \mathbb{E}_r\|\xi_{k,t}\|^2 \leq \sigma^2.$$

Using $\|a-b\|^2 = \|a\|^2 - 2\langle a, b\rangle + \|b\|^2$, we get

$$\mathbb{E}_r\|u_{k,t+1}\|^2 = \mathbb{E}_r\|u_{k,t}\|^2 - 2\eta_r\,\mathbb{E}_r\langle u_{k,t}, \nabla f_k(w_{r+\frac{1}{2},t}^{(k)})\rangle + \eta_r^2\mathbb{E}_r\|\nabla f_k(w_{r+\frac{1}{2},t}^{(k)})\|^2 + \eta_r^2\sigma^2.$$

Write $\nabla f_k(w_{r+\frac{1}{2},t}^{(k)}) = \nabla f_k(w_r) + \Delta_{k,t}$ with $\|\Delta_{k,t}\| \leq L\|u_{k,t}\|$ ($L$-smoothness), so

$$-2\eta_r\langle u_{k,t}, \Delta_{k,t}\rangle \leq 2\eta_r L\|u_{k,t}\|^2,$$

and using Young's inequality $2ab \leq \gamma a^2 + \frac{1}{\gamma}b^2$ with $\gamma = \frac{1}{4T}$,

$$-2\eta_r\langle u_{k,t}, \nabla f_k(w_r)\rangle \leq \tfrac{1}{4T}\|u_{k,t}\|^2 + 4\eta_r^2 T\|\nabla f_k(w_r)\|^2.$$

Since $\eta_r \leq 1/(8LT)$, then $2\eta_r L \leq 1/(4T)$, hence

$$-2\eta_r\,\mathbb{E}_r\langle u_{k,t}, \nabla f_k(w_{r+\frac{1}{2},t}^{(k)})\rangle \leq \tfrac{1}{2T}\,\mathbb{E}_r\|u_{k,t}\|^2 + 4\eta_r^2 T\|\nabla f_k(w_r)\|^2.$$

Also,

$$\mathbb{E}_r\|\nabla f_k(w_{r+\frac{1}{2},t}^{(k)})\|^2 \leq 2\|\nabla f_k(w_r)\|^2 + 2L^2\mathbb{E}_r\|u_{k,t}\|^2.$$

Combining gives:

$$\mathbb{E}_r \|u_{k,t+1}\|^2 \le \left(1 + \tfrac{1}{2T} + 2\eta_r^2 L^2\right) \mathbb{E}_r \|u_{k,t}\|^2 + (2 + 4T)\eta_r^2 \|\nabla f_k(w_r)\|^2 + \eta_r^2 \sigma^2.$$

Let $A := 1 + \tfrac{1}{2T} + 2\eta_r^2 L^2 \le 1 + \tfrac{1}{T} \le e^{1/T}$. Define $\bar{D}_t := \tfrac{1}{K} \sum_k \mathbb{E}_r \|u_{k,t}\|^2$. Using Assumption 3 we get

$$\bar{D}_{t+1} \le A\,\bar{D}_t + (2 + 4T)\eta_r^2 \left(\beta^2 \|\nabla f(w_r)\|^2 + \nu^2\right) + \eta_r^2 \sigma^2 \;=:\; A\bar{D}_t + B.$$

Since $A \le e^{1/T}$, for $t \le T$ we have $A^t \le e \le 3$ and

$$\frac{A^t - 1}{A - 1} \le \frac{e - 1}{A - 1} \le 2eT \le 6T,$$

because $A - 1 \ge \tfrac{1}{2T}$. Therefore,

$$\bar{D}_t \le A^t \bar{D}_0 + \frac{A^t - 1}{A - 1} B \le 3\bar{D}_0 + 6T\eta_r^2 \sigma^2 + 6T(2 + 4T)\eta_r^2 \left(\beta^2 \|\nabla f(w_r)\|^2 + \nu^2\right).$$

Finally, with $\bar{D}_0 = \alpha_r^2 \tfrac{1}{K} \sum_k \mathbb{E}_r \|e_r^{(k)}\|^2$, we loosen constants to the displayed form in the lemma:

$$\bar{D}_t \le 12\eta_r^2 T \left(\sigma^2 + 6T\nu^2 + \beta^2 \|\nabla f(w_r)\|^2\right) + 24\eta_r^2 T^2 \left(\beta^2 \|\nabla f(w_r)\|^2 + 4\nu^2\right) + 3\alpha_r^2 \tfrac{1}{K} \sum_k \mathbb{E}_r \|e_r^{(k)}\|^2.$$

Combine the $T$ and $T^2$ terms using $12T + 24T^2 \le 36T^2$ and compute $12T \times 6T + 24T^2 \times 4 = 168T^2$, we get

$$\bar{D}_t \le 12\,\eta_r^2 T\,\sigma^2 + 168\,\eta_r^2 T^2\,\nu^2 + 36\,\eta_r^2 T^2\,\beta^2\,\|\nabla f(w_r)\|^2 + 3\,\alpha_r^2 \times \frac{1}{K} \sum_{k=1}^{K} \mathbb{E}_r \|e_r^{(k)}\|^2.$$

$\square$

**Lemma 3** (Second moment of the shifted average update). *Let $K$ be the number of clients, $T \ge 1$ the number of local steps, $\eta_r \in (0, \tfrac{1}{8LT}]$ the local stepsize in round $r$, and $\alpha_r \in [0, 1]$ the step-ahead parameter. Define*

$$\bar{e}_r = \frac{1}{K} \sum_{k=1}^{K} e_r^{(k)}, \qquad g_r^{(k)} = \eta_r \sum_{t=0}^{T-1} \left(\nabla f_k(w_{r+\frac{1}{2},t}^{(k)}) + \xi_{k,t}\right), \qquad \bar{g}_r = \frac{1}{K} \sum_{k=1}^{K} g_r^{(k)}.$$

*Under Assumptions 1–3,*

$$\mathbb{E}_r\left[\|\alpha_r \bar{e}_r - \bar{g}_r\|^2\right] \le 2\alpha_r^2 \bar{E}_r + 8\eta_r^2 T^2 \|\nabla f(w_r)\|^2 \left(1 + 36L^2 \eta_r^2 T^2 \beta^2\right)$$
$$+ \frac{4\eta_r^2 T \sigma^2}{K} + 96L^2 \eta_r^4 T^3 \left(\sigma^2 + 14T\nu^2\right) + 24\alpha_r^2 \eta_r^2 L^2 T^2 \bar{E}_r. \tag{17}$$

*Here $\bar{E}_r = \dfrac{1}{K} \sum_{k=1}^{K} \|e_r^{(k)}\|^2$ is the average residual energy at the beginning of round $r$, and $w_r$ is the global model before local computation.*

*Proof.* Using Young's inequality, for any $a, b \in \mathbb{R}^d$, $\|a - b\|^2 \le 2\|a\|^2 + 2\|b\|^2$. With $a = \alpha_r \bar{e}_r$ (deterministic given $\mathcal{F}_r$) and $b = \bar{g}_r$,

$$\|\alpha_r \bar{e}_r - \bar{g}_r\|^2 \le 2\alpha_r^2 \|\bar{e}_r\|^2 + 2\|\bar{g}_r\|^2. \tag{18}$$

Since $\|\bar{e}_r\|^2 \le \bar{E}_r$ (Jensen), the first term in equation 18 gives $2\alpha_r^2 \bar{E}_r$ after expectation.

Bounding the second moment of $\bar{g}_r$ and the gradient–noise cross term. Write $\bar{g}_r = \eta_r \sum_{t=0}^{T-1} (\bar{\nabla}_t + \bar{\xi}_t)$ with $\bar{\nabla}_t = \tfrac{1}{K} \sum_k \nabla f_k(w_{r+\frac{1}{2},t}^{(k)})$ and $\bar{\xi}_t = \tfrac{1}{K} \sum_k \xi_{k,t}$. By Young's inequality,

$$\mathbb{E}_r \|\bar{g}_r\|^2 \;\le\; 2\eta_r^2 \, \mathbb{E}_r \left\| \sum_{t=0}^{T-1} \bar{\nabla}_t \right\|^2 + 2\eta_r^2 \, \mathbb{E}_r \left\| \sum_{t=0}^{T-1} \bar{\xi}_t \right\|^2.$$

Using independence across $(k, t)$ and Assumption 2, $\mathbb{E}_r \| \sum_t \bar{\xi}_t \|^2 = \sum_t \mathbb{E}_r \| \bar{\xi}_t \|^2 \le T \sigma^2 / K$.

Bounding the summed local gradients. Decompose $\nabla f_k(w_{r+\frac{1}{2},t}^{(k)}) = \nabla f_k(w_r) + \Delta_{k,t}$ with $\| \Delta_{k,t} \| \le L \| w_{r+\frac{1}{2},t}^{(k)} - w_r \|$ (Assumption 1). Then

$$\frac{1}{K^2} \mathbb{E}_r \left\| \sum_{k,t} \nabla f_k(w_{r+\frac{1}{2},t}^{(k)}) \right\|^2 \le 2T^2 \| \nabla f(w_r) \|^2 + \frac{2L^2 T}{K} \sum_{k,t} \mathbb{E}_r \| w_{r+\frac{1}{2},t}^{(k)} - w_r \|^2.$$

Insert the local-model drift (Lemma 2) and sum over $t$. For each $t \in \{0, \ldots, T-1\}$, Lemma 2 yields

$$\frac{1}{K} \sum_k \mathbb{E}_r \| w_{r+\frac{1}{2},t}^{(k)} - w_r \|^2 \le 12 \eta_r^2 T \sigma^2 + 168 \eta_r^2 T^2 \nu^2 + 36 \eta_r^2 T^2 \beta^2 \| \nabla f(w_r) \|^2 + 3\alpha_r^2 \bar{E}_r.$$

Summing over $t = 0, \ldots, T-1$ gives

$$\frac{1}{K} \sum_{k,t} \mathbb{E}_r \| w_{r+\frac{1}{2},t}^{(k)} - w_r \|^2 \le 12 \eta_r^2 T^2 \sigma^2 + 168 \eta_r^2 T^3 \nu^2 + 36 \eta_r^2 T^3 \beta^2 \| \nabla f(w_r) \|^2 + 3T\alpha_r^2 \bar{E}_r.$$

Assemble,

$$\mathbb{E}_r \| \bar{g}_r \|^2 \le 2\eta_r^2 \left[ 2T^2 \| \nabla f(w_r) \|^2 + \frac{2L^2 T}{K} \sum_{k,t} \mathbb{E}_r \| w_{r+\frac{1}{2},t}^{(k)} - w_r \|^2 \right] + 2\eta_r^2 \times \frac{T\sigma^2}{K}$$

$$\le 4\eta_r^2 T^2 \| \nabla f(w_r) \|^2 \left( 1 + 36 L^2 \eta_r^2 T^2 \beta^2 \right) + \frac{2\eta_r^2 T \sigma^2}{K} + 48 L^2 \eta_r^4 T^3 \left( \sigma^2 + 14 T \nu^2 \right)$$

$$+ 12 \alpha_r^2 \eta_r^2 L^2 T^2 \bar{E}_r.$$

Finally, apply the factor 2 from equation 18 and add $2\alpha_r^2 \bar{E}_r$. This is precisely the assertion of the lemma. $\qquad \square$

**Lemma 4** (Residual recursion under a $\delta$–contractive compressor)**.** *Fix a round $r$ and define $s_r := \eta_r LT \le \frac{1}{8}$. Let Assumption 1–3 hold and the compressor satisfies Definition 1 with parameter $\delta \ge 1$. Let*

$$\bar{E}_r := \frac{1}{K} \sum_{k=1}^K \| e_r^{(k)} \|^2, \qquad g_r^{(k)} := \eta_r \sum_{t=0}^{T-1} \left( \nabla f_k(w_{r+\frac{1}{2},t}^{(k)}) + \xi_{k,t} \right).$$

*Then, with all expectations conditional on the randomness inside round $r$, the averaged residual energy obeys*

$$\bar{E}_{r+1} \le \rho_r \bar{E}_r + \left( 1 - \frac{1}{\delta} \right) \left[ B_r^{(\nabla)} + B_r^{(\nu,\sigma)} \right], \tag{19}$$

*where*

$$\rho_r = \left( 1 - \frac{1}{\delta} \right) \left( 2(1 - \alpha_r)^2 + 24 \alpha_r^2 s_r^2 \right) = \left( 1 - \frac{1}{\delta} \right) \left( 2 - 4\alpha_r + (2 + 24 s_r^2)\alpha_r^2 \right),$$

*and*

$$B_r^{(\nabla)} := 8 \eta_r^2 T^2 \beta^2 \| \nabla f(w_r) \|^2 + 288 L^2 \eta_r^4 T^4 \beta^2 \| \nabla f(w_r) \|^2,$$

$$B_r^{(\nu,\sigma)} := 8 \eta_r^2 T^2 \nu^2 + 4 \eta_r^2 T \sigma^2 + 96 L^2 \eta_r^4 T^3 \sigma^2 + 1344 L^2 \eta_r^4 T^4 \nu^2.$$

*Proof.* Write $u_{r+1}^{(k)} = (1 - \alpha_r) e_r^{(k)} + g_r^{(k)}$ and $e_{r+1}^{(k)} = u_{r+1}^{(k)} - C(u_{r+1}^{(k)})$. By Definition 1,

$$\mathbb{E}_r \| e_{r+1}^{(k)} \|^2 \le \left( 1 - \frac{1}{\delta} \right) \mathbb{E}_r \| u_{r+1}^{(k)} \|^2.$$

Average over $k$ and use $\| a + b \|^2 \le 2\|a\|^2 + 2\|b\|^2$:

$$\frac{1}{K} \sum_k \mathbb{E}_r \| u_{r+1}^{(k)} \|^2 \le 2(1 - \alpha_r)^2 \bar{E}_r + 2 \times \frac{1}{K} \sum_k \mathbb{E}_r \| g_r^{(k)} \|^2.$$

Next,

$$\mathbb{E}_r \|g_r^{(k)}\|^2 \leq 2\eta_r^2 \, \mathbb{E}_r \Big\| \sum_t \nabla f_k(w_{r+\frac{1}{2},t}^{(k)}) \Big\|^2 + 2\eta_r^2 \, T\sigma^2,$$

and by $L$-smoothness, $\nabla f_k(w_{r+\frac{1}{2},t}^{(k)}) = \nabla f_k(w_r) + \Delta_{k,t}$ with $\|\Delta_{k,t}\| \leq L\|w_{r+\frac{1}{2},t}^{(k)} - w_r\|$. Thus

$$\mathbb{E}_r \Big\| \sum_t \nabla f_k(w_{r+\frac{1}{2},t}^{(k)}) \Big\|^2 \leq 2T^2 \|\nabla f_k(w_r)\|^2 + 2L^2 T \sum_t \mathbb{E}_r \|w_{r+\frac{1}{2},t}^{(k)} - w_r\|^2.$$

Averaging over $k$ and using the dissimilarity condition yields

$$\frac{1}{K} \sum_k \mathbb{E}_r \|g_r^{(k)}\|^2 \leq 4\eta_r^2 T^2 \big(\beta^2 \|\nabla f(w_r)\|^2 + \nu^2\big) + 4\eta_r^2 L^2 T \, S + 2\eta_r^2 T\sigma^2,$$

where $S := \frac{1}{K} \sum_k \sum_{t=0}^{T-1} \mathbb{E}_r \|w_{r+\frac{1}{2},t}^{(k)} - w_r\|^2$. By Lemma 2 (summed over $t$),

$$S \leq 12 \, \eta_r^2 T^2 \sigma^2 + 168 \, \eta_r^2 T^3 \nu^2 + 36 \, \eta_r^2 T^3 \beta^2 \|\nabla f(w_r)\|^2 + 3T\alpha_r^2 \bar{E}_r.$$

Substituting and noting $L^2 \eta_r^2 T^2 = s_r^2$ gives

$$\frac{1}{K} \sum_k \mathbb{E}_r \|g_r^{(k)}\|^2 \leq \tfrac{1}{2}\big(B_r^{(\nabla)} + B_r^{(\nu,\sigma)}\big) + 12 \, \alpha_r^2 \eta_r^2 L^2 T^2 \, \bar{E}_r.$$

Insert this into the previous split to obtain

$$\frac{1}{K} \sum_k \mathbb{E}_r \|u_{r+1}^{(k)}\|^2 \leq \Big(2(1-\alpha_r)^2 + 24 \, \alpha_r^2 \eta_r^2 L^2 T^2\Big)\bar{E}_r + B_r^{(\nabla)} + B_r^{(\nu,\sigma)}.$$

Finally multiply by $(1 - \frac{1}{\delta})$ to conclude equation 19. $\qquad\square$

**Proposition 1** (Residual contraction vs. EF). *Under the conditions of Lemma 4, let $\rho_{\mathrm{EF}} := 2\big(1 - \frac{1}{\delta}\big)$ (the $\alpha_r = 0$ baseline). Then*

$$\rho_r = \Big(1 - \tfrac{1}{\delta}\Big)\Big(2 - 4\alpha_r + (2 + 24s_r^2)\alpha_r^2\Big), \qquad s_r = \eta_r LT \leq \tfrac{1}{8},$$

*and:*

1. *Strict improvement region. For any $\alpha_r \in \big(0, \frac{1}{1+12s_r^2}\big)$,*

$$\rho_r - \rho_{\mathrm{EF}} = \Big(1 - \tfrac{1}{\delta}\Big)\Big[-4\alpha_r + (2 + 24s_r^2)\alpha_r^2\Big] < 0,$$

*and at the boundary $\alpha_r = \frac{2}{1+12s_r^2}$ the difference equals $0$.*

2. *Optimal step-ahead. The minimiser over $\alpha_r \in [0,1]$ is $\alpha_r^\star = \frac{1}{1+12s_r^2} \in (0.84, 1]$ and the minimum value is*

$$\rho_{\min} = \Big(1 - \tfrac{1}{\delta}\Big)\Big[2 - \tfrac{2}{1+12s_r^2}\Big] = \rho_{\mathrm{EF}}\Big(1 - \tfrac{1}{1+12s_r^2}\Big).$$

*In particular, as $s_r \to 0$, $\rho_{\min} \to 0$ and $\rho_{\min}/\rho_{\mathrm{EF}} \to 0$.*

*Proof.* Direct algebra from the quadratic form of $\rho_r$. For (i), the sign of $-4\alpha_r + (2 + 24s_r^2)\alpha_r^2 = \alpha_r\big((2 + 24s_r^2)\alpha_r - 4\big)$ is negative when $\alpha_r < 4/(2 + 24s_r^2) = 1/(1 + 12s_r^2)$, and zero at equality. For (ii), minimize $q(\alpha) = 2 - 4\alpha + (2 + 24s_r^2)\alpha^2$ to get $\alpha_r^\star = 2/(2 + 24s_r^2) = 1/(1 + 12s_r^2)$ and $q(\alpha_r^\star) = 2 - 2/(1 + 12s_r^2)$; multiply by $(1 - \frac{1}{\delta})$. $\qquad\square$

### A.3 PARTIAL PARTICIPATION: ANALYSIS AND RATES

At round $r$, let $\mathcal{M}_r \subseteq [K]$ be sampled uniformly without replacement, $|\mathcal{M}_r| = m$, and denote $p := m/K \in (0, 1]$. Write $I_r^{(k)} = \mathbf{1}\{k \in \mathcal{M}_r\}$. Active clients run the same inner loop as in full participation,

$$g_r^{(k)} = \begin{cases} \eta_r \sum_{t=0}^{T-1} \big(\nabla f_k(w_{r+\frac{1}{2},t}^{(k)}) + \xi_{k,t}\big), & I_r^{(k)} = 1, \\ 0, & I_r^{(k)} = 0, \end{cases} \qquad e_{r+1}^{(k)} = \begin{cases} u_{r+1}^{(k)} - C(u_{r+1}^{(k)}), & I_r^{(k)} = 1, \\ e_r^{(k)}, & I_r^{(k)} = 0, \end{cases}$$

with $u_{r+1}^{(k)} = (1 - \alpha_r)e_r^{(k)} + g_r^{(k)}$ for active clients. The server update is

$$w_{r+1} = w_r - \eta\,\bar{C}_{r+1}, \qquad \bar{C}_{r+1} := \frac{1}{m} \sum_{k \in \mathcal{M}_r} C\big(u_{r+1}^{(k)}\big).$$

Expectations $\mathbb{E}_r[\cdot]$ are over local randomness and the draw of $\mathcal{M}_r$. Assumptions 1–3 and Definition 1 (with $\delta \geq 1$) hold.

**Active/global averages; virtual iterate.** Define

$$\tilde{e}_r := \frac{1}{K} \sum_{k=1}^{K} e_r^{(k)}, \qquad \bar{e}_r := \frac{1}{m} \sum_{k \in \mathcal{M}_r} e_r^{(k)}, \qquad \bar{g}_r := \frac{1}{m} \sum_{k \in \mathcal{M}_r} g_r^{(k)}.$$

Note $\mathbb{E}_{\mathcal{M}_r}[\bar{e}_r \mid \{e_r^{(k)}\}] = \tilde{e}_r$. Let $x_r := w_r - \eta\,\tilde{e}_r$.

**Lemma 5** (PP virtual-iterate identity). *With the definitions above,*

$$x_{r+1} - x_r = \eta\Big[ p\big(\alpha_r \bar{e}_r - \bar{g}_r\big) \;-\; (1 - p)\,\bar{C}_{r+1}\Big]. \tag{20}$$

**Lemma 6** (PP compression second moment). *Let $c_\delta := 2\big(2 - \frac{1}{\delta}\big)$. Then, conditionally on $\mathcal{M}_r$,*

$$\mathbb{E}_r \|\bar{C}_{r+1}\|^2 \;\leq\; \frac{1}{m} \sum_{k \in \mathcal{M}_r} \mathbb{E}_r \|C(u_{r+1}^{(k)})\|^2 \;\leq\; \frac{c_\delta}{m} \sum_{k \in \mathcal{M}_r} \mathbb{E}_r \|u_{r+1}^{(k)}\|^2.$$

**Lemma 7** (One-round descent under PP). *Let $\Delta_r^{\mathrm{act}} := \alpha_r \bar{e}_r - \bar{g}_r$. Under the same alignment and stepsize coupling as in full participation,*

$$\text{(C1)}\ \ \eta_r^2 L^2 \beta^2 (6T^2 + 12T^3) \leq \frac{T}{8}, \qquad \text{(C2)}\ \ \eta \leq \frac{1}{256\,\beta^2\,L\,\eta_r T},$$

*there exists a universal $c > 0$ such that*

$$\mathbb{E}_r\big[f(x_{r+1}) - f(x_r)\big] \;\leq\; -c\,p\,\eta\,\eta_r T\,\|\nabla f(w_r)\|^2 \;+\; \mathbb{E}_r\big[\widehat{\mathcal{E}}_r\,\tilde{E}_r\big] \;+\; \widehat{\mathcal{V}}_r\,\sigma^2 \;+\; \widehat{\mathcal{H}}_r\,\nu^2, \tag{21}$$

*where $\tilde{E}_r := \frac{1}{K} \sum_{k=1}^{K} \|e_r^{(k)}\|^2$ and $\widehat{\mathcal{E}}_r, \widehat{\mathcal{V}}_r, \widehat{\mathcal{H}}_r$ equal the full-participation coefficients scaled by $p$ and augmented by $(1 - p)$–terms originating from $\bar{C}_{r+1}$ via Lemma 6.*

**Lemma 8** (Residual recursion under PP). *Let*

$$\rho_r^{\mathrm{PP}} := (1 - p) \;+\; p\Big(1 - \frac{1}{\delta}\Big)\Big(2(1 - \alpha_r)^2 + 24\,\alpha_r^2(\eta_r LT)^2\Big).$$

*Then*

$$\mathbb{E}_r \tilde{E}_{r+1} \;\leq\; \rho_r^{\mathrm{PP}}\,\tilde{E}_r \;+\; p\Big(1 - \frac{1}{\delta}\Big)\Big[B_r^{(\nabla)} + B_r^{(\nu,\sigma)}\Big], \tag{22}$$

*with $B_r^{(\nabla)}$ and $B_r^{(\nu,\sigma)}$ as in full participation.*

**Telescoping, absorption, and final bounds (PP).** Let $S_R := \sum_{r=0}^{R-1} \eta_r T$ and $S_R^{\mathrm{PP}} := \sum_{r=0}^{R-1} p\,\eta_r T = p\,S_R$. Summing equation 21 over $r$ and using $\mathcal{C}_r^{\mathrm{PP}} \leq -c\,p\,\eta\,\eta_r T$,

$$\sum_{r=0}^{R-1} p\,\eta_r T\,\mathbb{E}\|\nabla f(w_r)\|^2 \;\leq\; \frac{16}{\eta}\big(f(x_0) - \mathbb{E}f(x_R)\big) + \frac{16}{\eta} \sum_{r=0}^{R-1} \mathbb{E}\big[\widehat{\mathcal{E}}_r\,\tilde{E}_r + \widehat{\mathcal{V}}_r\,\sigma^2 + \widehat{\mathcal{H}}_r\,\nu^2\big].$$

If $f$ is bounded below by $f_\star$, then $f(x_R) \geq f_\star$. Summing equation 22 and assuming $\rho_{\max}^{\text{PP}} := \sup_r \rho_r^{\text{PP}} < 1$,

$$\sum_{r=0}^{R-1} \mathbb{E}\tilde{E}_r \leq \frac{1}{1 - \rho_{\max}^{\text{PP}}} \tilde{E}_0 + \frac{p}{1 - \rho_{\max}^{\text{PP}}} \left(1 - \tfrac{1}{\delta}\right) \sum_{r=0}^{R-1} \mathbb{E}\big[B_r^{(\nabla)} + B_r^{(\nu,\sigma)}\big].$$

Plugging this into the previous display yields a gradient-forcing term proportional to

$$\frac{16}{\eta} \times \frac{\widehat{\mathcal{E}}_{\max}}{1 - \rho_{\max}^{\text{PP}}} \left(1 - \tfrac{1}{\delta}\right) \times \underbrace{p \sum_r B_r^{(\nabla)}}_{\leq \beta^2 d_{\max} \, p \sum_r \eta_r T \, \mathbb{E}\|\nabla f(w_r)\|^2},$$

where $d_{\max} := \sup_r \big(8\,\eta_r T + 288\, L^2 \eta_r^3 T^3\big)$. Defining

$$\Theta_{\text{PP}} := \frac{16}{\eta} \times \frac{\widehat{\mathcal{E}}_{\max}}{1 - \rho_{\max}^{\text{PP}}} \left(1 - \tfrac{1}{\delta}\right) \beta^2 d_{\max},$$

we obtain the absorption inequality (the factor $p$ cancels on both sides):

$$\big(1 - \Theta_{\text{PP}}\big) \sum_{r=0}^{R-1} p\,\eta_r T \, \mathbb{E}\|\nabla f(w_r)\|^2 \leq \frac{16}{\eta} \Big[(f(x_0) - f_\star) + \sum_{r=0}^{R-1} \widehat{\mathcal{V}}_r \, \sigma^2 + \widehat{\mathcal{H}}_r \, \nu^2\Big].$$

Assuming $\Theta_{\text{PP}} \leq \frac{1}{2}$, we conclude

$$\sum_{r=0}^{R-1} p\,\eta_r T \, \mathbb{E}\|\nabla f(w_r)\|^2 \leq \frac{32}{\eta} \Big[(f(x_0) - f_\star) + \sum_{r=0}^{R-1} \widehat{\mathcal{V}}_r \, \sigma^2 + \widehat{\mathcal{H}}_r \, \nu^2\Big].$$

Dividing by $S_R^{\text{PP}} = p\,\eta_0 TR$ and with $\eta_r \equiv \eta_0$ yields the averaged bounds below.

$$\begin{aligned} \frac{1}{R} \sum_{r=0}^{R-1} \mathbb{E}\|\nabla f(w_r)\|^2 &\leq \frac{32}{\eta\,p\,\eta_0 TR}\,(f(x_0) - f_\star) + \frac{128\,L\,\eta_0}{\eta\,p\,m}\,\sigma^2 \\ &\quad + \frac{32}{\eta\,p}\left(1 - \tfrac{1}{\delta}\right)\Big[C_\sigma\,\eta_0^2 L^2 T\,\sigma^2 + C_\nu\,\eta_0^2 L^2 T^2\,\nu^2\Big]. \end{aligned} \tag{23}$$

So, the optimization term scales as $O\big((p\,\eta\,\eta_0 TR)^{-1}\big)$ (a per-round slow-down by $1/p = K/m$). The *pure mini-batch* variance enjoys a $1/m$ reduction: its contribution scales as $\frac{128\,L\,\eta_0}{m}\,\sigma^2$, while the residual-induced variance/heterogeneity floors scale as $\left(1 - \tfrac{1}{\delta}\right)\big[C_\sigma\,\eta_0^2 L^2 T\,\sigma^2 + C_\nu\,\eta_0^2 L^2 T^2 \nu^2\big]$.

**Stalling vs. step-ahead.** With $\alpha_r = 0$ (no step-ahead), the multiplicative factor becomes $\rho_r^{\text{PP}} = (1 - p) + 2p(1 - \tfrac{1}{\delta}) = 1 + p(1 - \tfrac{2}{\delta})$, which can be $\geq 1$ under aggressive compression and small $p$, explaining the slowdown in cross-device regimes. For moderate $\alpha_r$ (e.g., $\alpha_r \approx \alpha_r^\star$ from the full-participation analysis), $\rho_r^{\text{PP}}$ strictly decreases, improving the decay of $\tilde{E}_r$ each time a client participates and restoring faster early progress.

**Constants and feasibility.** The constants $C_\sigma$, $C_\nu$ and $\Theta$ in Theorem 1 collect the contributions of stochastic-gradient variance, data heterogeneity, and compression–induced residual drift. Inspecting their explicit formulas, we see that they depend on the algorithmic hyperparameters only through the effective local stepsize

$$s_0 = \eta_0 LT,$$

the compression bias factor $(1 - 1/\delta)$, and the residual–contraction term $1/(1 - \rho_{\max})$. In particular, $\rho_{\max}$ itself is an increasing function of $s_0$ and $(1 - 1/\delta)$, so $C_\sigma$, $C_\nu$ and $\Theta$ are *monotone nondecreasing* in $s_0$ and $(1 - 1/\delta)$: larger local work or more aggressive compression lead to larger constants and thus a higher residual-driven floor.

In the partial-participation extension (Remark 1), the corresponding constant $\Theta_{\text{PP}}$ inherits the same monotone dependence on $s_0$ and $(1 - 1/\delta)$ and, in addition, scales inversely with the participation

Table 1: Comparison of compressed algorithms for FL. **SA-PEF** bridges the gap between Fed-EF (stable but slower under aggressive compression) and SAEF (faster warm-up but fragile in heterogeneous FL), achieving improved residual contraction without the extra state and complexity of control-variate methods such as SCAFCOM.

| Algorithm | Assumptions (beyond $L$-smoothness) | Mechanism & State | Convergence / Behaviour (Nonconvex) |
|---|---|---|---|
| Fed-EF (Li & Li, 2023) | Bounded variance; local SGD with PP | Biased $\delta$-contractive; **Stateless** (one residual/client) | Standard FL nonconvex rate with $1/p$ slowdown under partial participation; residual contraction factor $\rho_{\mathrm{EF}}$ can be relatively weak under aggressive compression, leading to slower progress and earlier stalling. |
| SAEF (Xu et al., 2021) | Bounded gradient; centralized, synchronous setting; no PP analysis | Full step-ahead ($\alpha = 1$); **Stateless** (one residual/client) | Analyzed in the classical distributed setting (no local steps, no client sampling); reduces EF's gradient mismatch there. In our FL experiments with heterogeneous data, full step-ahead tends to produce larger gradient mismatch and late-stage plateaus compared to EF/SA-PEF (Sec. 4). |
| CSER (Xie et al., 2020) | Bounded variance; local SGD (typically full participation); | Error reset (periodic dense communication of residuals); **Stateless** | Controls residual drift via periodic resets, yielding a nonconvex local-SGD rate with an $R$-independent floor. However, resets require sending full residuals, inducing **high peak bandwidth** at reset rounds and analyses usually assume full participation. |
| SCAFCOM (Huang et al., 2024) | **Arbitrary heterogeneity**; bounded variance; local SGD with PP | Control variates + momentum; **Stateful** (extra state $c, c_i$ per client) | Achieves a nonconvex FL guarantees under arbitrary non-IID data, with improved dependence on heterogeneity, at the cost of **higher system complexity** (maintaining control-variates state). |
| **SA-PEF (ours)** | Gradient dissimilarity $(\beta, \nu)$; local SGD with PP | Partial step-ahead $(0 < \alpha < 1)$; **Stateless** (one residual/client) | Operates in the same FL regime and under the same assumptions as Fed-EF, with the same leading-order nonconvex rate, but with a **strictly improved** residual contraction factor $\rho_{\max} < \rho_{\mathrm{EF}}$ under biased compression, which lowers the error floor and balances warm-up speed with long-term stability. |

rate $p = m/K$ (i.e., it increases as $p$ decreases). Thus, the feasibility conditions $\rho_{\max}^{\mathrm{PP}} < 1$ and $\Theta_{\mathrm{PP}} \leq \frac{1}{2}$ can be interpreted as requiring a standard "small" effective local stepsize $s_0$, moderate compression, and not-too-extreme partial participation. For the default hyperparameters used in our experiments (e.g., $T = 5$, Top-$k$ compression, and $p \in \{0.1, 0.2, 1.0\}$), we numerically evaluate these constants and confirm that $\rho_{\max}^{\mathrm{PP}} < 1$ in all regimes. Moreover, in a mildly compressed setting (e.g., $\delta = 1.005$ with the same stepsizes), the corresponding $\Theta_{\mathrm{PP}}$ lies well below $\frac{1}{2}$, illustrating that the condition $\Theta_{\mathrm{PP}} \leq \frac{1}{2}$ is a conservative sufficient condition rather than a tight practical tuning rule for the aggressively compressed Top-$k$ regimes we study.

### A.4 COMPARISON OF EF-TYPE COMPRESSED FL METHODS.

For completeness, In Table 1, we summarizes the main assumptions, mechanisms, and qualitative nonconvex behavior of several closely related algorithms: Fed-EF, SAEF, CSER, SCAFCOM, and SA-PEF. The goal is not to restate full theorems, but to highlight the regimes they target. Fed-EF and SA-PEF share the same lightweight, stateless EF architecture and standard FL assumptions (local steps, partial participation, biased contractive compressors), with SA-PEF improving the residual-contraction constant under compression. CSER and SAEF focus on centralized/local-SGD settings without partial participation, while SCAFCOM achieves stronger robustness to arbitrary heterogeneity by adding SCAFFOLD-style control variates and momentum, at the cost of increasing per-client state and communication.

**Remark 2** (Relation to EF21). *EF21 and its extensions (Richtárik et al., 2021; Fatkhullin et al., 2025) obtain stronger guarantees (no error floor) under a different regime: synchronized data-parallel training with $T=1$, full-gradient (or gradient-difference) compression at a shared iterate, and no local steps. In this setting, EF21 is strictly preferable to classical EF. Our analysis targets a complementary regime, federated local-SGD with $T > 1$ local steps, partial participation, and biased contractive compressors, where the compressed object is the accumulated local update and client drift plays a central role. Extending EF21-style arguments (or designing EF21-style step-ahead variants) to this local-SGD, partial-participation, biased-compression setting is non-trivial and remains an interesting direction for future work.*

# B IMPLEMENTATION DETAILS AND ADDITIONAL EXPERIMENTS

## B.1 SETUP AND PARAMETER TUNING

This appendix details datasets, federated partitioning, compressors, the hyperparameter search protocol, and fairness controls used across all methods.

**Datasets, models, and preprocessing.** CIFAR-10 (ResNet-9), CIFAR-100 (ResNet-18), and Tiny-ImageNet ($64\times64$; ResNet-34) trained with cross-entropy loss. Preprocessing follows standard practice: per-dataset mean/std normalization; CIFAR uses random crop (4-pixel padding) and horizontal flip; Tiny-ImageNet uses random resized crop to 64 and horizontal flip. Unless stated otherwise, batch size is 64, momentum is 0.9, weight decay is $5\times10^{-4}$ on CIFAR and $10^{-4}$ on Tiny-ImageNet.

**Federated partitioning, participation, and local computation.** We create $K{=}100$ clients and apply Dirichlet label partitioning with $\gamma \in \{0.1, 1.0\}$ (smaller $\gamma \Rightarrow$ stronger non-IID). Each round samples $m = \lfloor pK \rfloor$ clients uniformly without replacement with $p \in \{0.1, 0.5, 1.0\}$. Participating clients run $T$ local SGD steps at stepsize $\eta_r$; default $T{=}5$. We train for $R{=}200$ rounds. Unless stated, server stepsize is $\eta{=}1.0$. We conducted all federated learning simulations using the FLOWER framework (Beutel et al., 2020).

**Compressors and communication accounting.** We use Top-$k$ sparsification with $k/d \in \{0.01, 0.05, 0.10\}$; each selected entry communicates its *index* and *value*. As a consequence, Top-$k$ satisfies Definition 1 with $\delta = d/k$; we record the standard bound below.

**Lemma 9** (Top-$k$ contraction; (Stich et al., 2018; Beznosikov et al., 2023)). *Let $C = \mathrm{Top}_k : \mathbb{R}^d \to \mathbb{R}^d$ keep the $k$ largest absolute-value coordinates of $x$ (ties broken arbitrarily), zeroing the rest. Then for all $x \in \mathbb{R}^d$,*

$$\|x - C(x)\|_2^2 \;\leq\; \left(1 - \frac{k}{d}\right) \|x\|_2^2, \qquad \frac{k}{d} \|x\|_2^2 \;\leq\; \|C(x)\|_2^2 \;=\; \langle C(x), x \rangle \;\leq\; \|x\|_2^2.$$

*In particular, $C$ is $\delta$-contractive with $\delta = d/k$ in the sense of Definition 1. The constants are tight when all $|x_i|$ are equal.*

Each selected entry transmits its *index* and *value*. *Raw uplink bits* per participating client per round are $k\left(\lceil \log_2 d \rceil + b_{\mathrm{val}}\right)$ with $b_{\mathrm{val}}{=}32$ for FP32 values. Unless stated, reported *cumulative communication* aggregates *uplink only* across participating clients (downlink is identical across compressed methods and omitted for fairness); FedAvg's downlink/uplink are both dense FP32.

**Hyperparameter search protocol.** We adopt a small, method-agnostic grid tuned on a held-out validation split. Unless noted, we select hyperparameters by best *validation top-1* at a *fixed communication budget* (bits) within $R$ rounds; ties are broken by higher accuracy at earlier checkpoints. We reuse the *same* grid across participation rates ($q$).

**Search spaces (shared across methods).**

- **Client LR $\eta_r$:**

| | |
|---|---|
| CIFAR-10: | $\{0.001, 0.05, 0.1, 0.2, 1.0, 10\}$ |
| CIFAR-100: | $\{0.001, 0.05, 0.1, 1.0, 10\}$ |
| Tiny-ImageNet: | $\{0.001, 0.02, 0.05, 1.0\}$ |
| | (cosine decay with 3-5 epoch warm-up, minimum $\eta_r = 0.005$) |

- **Server LR $\eta$:** $\{0.5, 1.0\}$.

- **Weight decay:** $\{5\times10^{-4}, 10^{-4}\}$.

- **Local steps $T$:** $\{1, 5, 10\}$.

- **Compressor level $k/d$:** $\{0.01, 0.05, 0.10\}$.

**SA-PEF-specific.** Constant-$\alpha$ ablations use $\alpha \in \{0.0, 0.1, 0.2, 0.3, 0.4, 0.5, 0.6, 0.7, 0.8, 0.9, 1.0\}$; unless stated, $\alpha$ is fixed across rounds.

Table 2: Final test accuracy (mean $\pm$ std. over five independent runs) for CIFAR-10 and CIFAR-100 under two participation/heterogeneity regimes. The hyperparameters used in all algorithms are $R = 200$, $T = 5$, and $\eta_l = 0.1$.

| Dataset | Model | Algorithm | Final test accuracy (%) | | | |
| --- | --- | --- | --- | --- | --- | --- |
| | | | $p = 0.1,\ \gamma = 0.5$ | | $p = 0.5,\ \gamma = 0.1$ | |
| | | | top-1 | top-10 | top-1 | top-10 |
| CIFAR-10 | ResNet-9 | FedAvg | $88.5 \pm 1.6$ | | $69.5 \pm 1.5$ | |
| | | EF | $57.0 \pm 2.0$ | $72.7 \pm 2.3$ | $40.3 \pm 3.7$ | $47.2 \pm 4.1$ |
| | | SAEF | $74.2 \pm 3.9$ | $75.5 \pm 2.9$ | $39.5 \pm 4.6$ | $48.5 \pm 3.3$ |
| | | CSER | $68.6 \pm 2.8$ | $80.5 \pm 1.2$ | $49.5 \pm 4.0$ | $67.2 \pm 2.6$ |
| | | SA-PEF | $80.5 \pm 2.6$ | $82.7 \pm 1.6$ | $47.5 \pm 3.6$ | $68.5 \pm 1.9$ |
| CIFAR-100 | ResNet-18 | FedAvg | $62.5 \pm 1.6$ | | $61.9 \pm 0.6$ | |
| | | EF | $40.4 \pm 2.4$ | $49.5 \pm 1.6$ | $41.5 \pm 1.6$ | $57.5 \pm 1.9$ |
| | | SAEF | $46.1 \pm 1.4$ | $50.5 \pm 2.0$ | $44.5 \pm 3.6$ | $57.5 \pm 3.9$ |
| | | CSER | $46.2 \pm 0.4$ | $51.5 \pm 1.9$ | $42.5 \pm 2.4$ | $60.7 \pm 1.0$ |
| | | SA-PEF | $49.6 \pm 1.8$ | $54.5 \pm 2.6$ | $48.5 \pm 2.0$ | $60.6 \pm 2.8$ |

**Fairness controls and evaluation.** (i) The *same* grid is used across methods; (ii) the best setting is selected at a matched bit budget; (iii) client sampling seeds are shared across methods; (iv) evaluation uses the server model in `eval` mode with identical preprocessing. We report both *accuracy vs. rounds* and *accuracy vs. bits*; the latter is our primary metric under communication constraints. Experiments ran on NVIDIA A100/A5000/H200 GPUs; hardware does not affect communication accounting.

### B.2 ADDITIONAL EXPERIMENTS

**Multi-seed stability.** In Table 2, we report final test accuracy as mean $\pm$ standard deviation over five independent runs with different random seeds. Across seeds, the relative ranking of methods is consistent, with SA-PEF retaining its advantage in high-compression, low-participation regimes.

Figures 5–8 report extra convergence curves for SA-PEF and the baselines on **CIFAR-10**, **CIFAR-100**, and **Tiny-ImageNet**. For each dataset we sweep (i) *participation* $q \in \{1.0,\ 0.1\}$, (ii) *compression budget* (Top-1% and Top-10% under full participation; Top-5% and Top-1% under $q=0.1$), and (iii) *data heterogeneity* via Dirichlet partitions. The plots include both *accuracy vs. rounds* and *accuracy vs. communicated GB*. Under full participation, SA-PEF consistently reaches a given accuracy in fewer rounds than EF/CSER and tracks SAEF without late-stage plateaus. Under partial participation with aggressive compression (Top-5%, Top-1%), the gaps naturally narrow but the qualitative trend persists, illustrating that the main conclusions are robust across datasets, architectures, and federation settings.

**IID control experiments.** To isolate the effect of data heterogeneity, we also evaluate under IID partitions in figure 9. We report *accuracy vs. rounds* and *accuracy vs. GB* for: (i) partial participation ($q=0.5$) with Top-5% and Top-10%. Across datasets, SA-PEF matches or exceeds EF/CSER in early rounds under the same communication budget.

**Extreme partial participation and local work.** To further stress-test our methods, we also consider more demanding FL regimes with very low participation and larger local work. In particular, we run experiments with participation $p = 0.05$ and $T = 10$ local SGD steps per round, comparing EF, CSER, and SA-PEF under the same compression level. As shown in Fig. 10, SA-PEF consistently attains higher accuracy than EF and CSER at a fixed communication budget, indicating that its advantages persist even under extreme partial participation and increased local work.

**Wall-clock efficiency.** To quantify implementation overhead, we report test accuracy versus wall-clock time on CIFAR-10/ResNet-9 under a fixed hardware setup (six NVIDIA RTX A5000 GPUs across 6 nodes) in the Figure 11. SA-PEF reaches a given accuracy level substantially earlier than

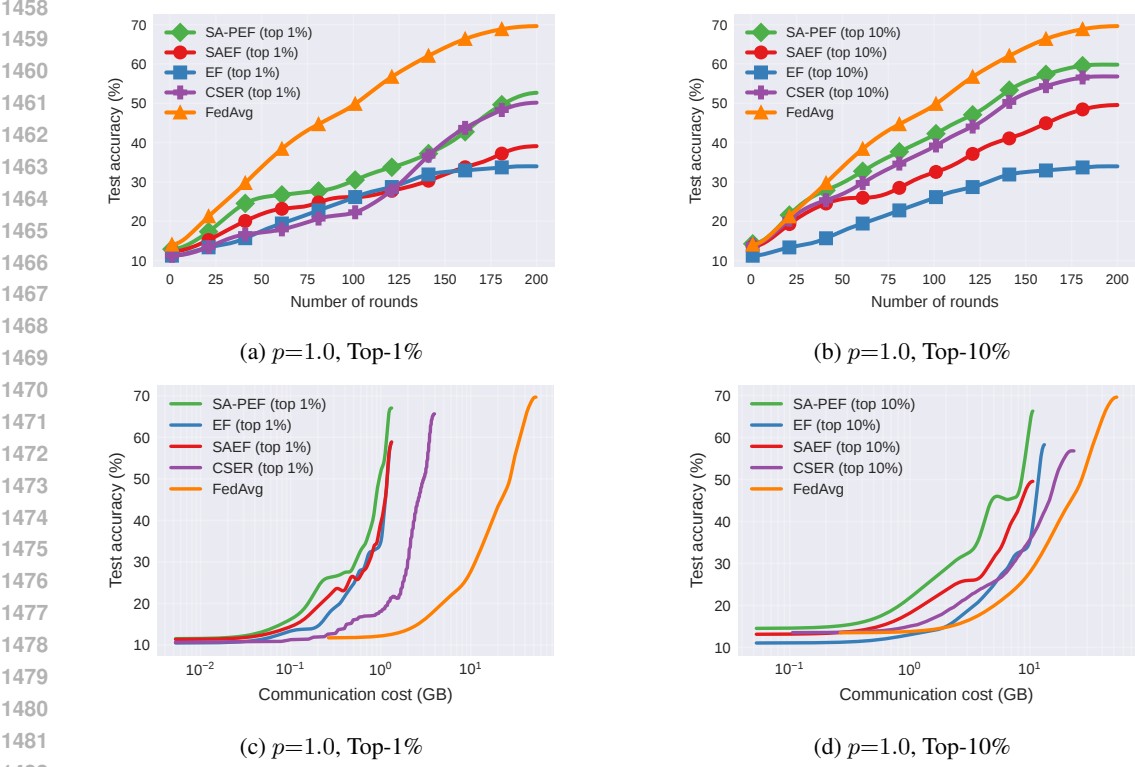

Figure 5: Test accuracy vs number of rounds (row 1) and communicated GB (row 2) on the CIFAR-10 dataset using ResNet-9 and $\gamma$=0.1.

EF, SAEF, CSER, and FedAvg, confirming that the small extra vector operations it introduces incur negligible runtime cost while improving time-to-accuracy.

**Scaled-sign compressor.** Besides Top-$k$, we also consider a scaled-sign compressor $C(x) = \frac{\|x\|_1}{d} \text{sign}(x)$. This is the group-scaled sign compressor of Li & Li (2023), specialized to a single block ($M = 1$), and Proposition C.1 in that work shows that it is contractive in the sense of our Definition 1 for a suitable $\delta > 0$, so it falls within our theoretical framework. Hence, we report preliminary CIFAR-10/ResNet-9 results with this scaled-sign compressor in Figure 12, where SA-PEF again achieves higher accuracy than EF, SAEF, and CSER at a fixed communication budget.

**Effect of momentum.** To isolate the role of momentum, we repeat our CIFAR-10/100 experiments using SGD *without* momentum, keeping all other hyperparameters and compression settings fixed, and provide the results in Figure 13. The test-accuracy trajectories and accuracy-communication curves show that SA-PEF consistently matches or outperforms EF, SAEF, and CSER, and remains close to FedAvg in terms of accuracy per communicated GB. This suggests that our conclusions are essentially momentum-agnostic: momentum slightly reshapes the trajectories but does not drive the gains of SA-PEF over EF-style baselines.

**Comparison with SCAFCOM.** To further assess SA-PEF, we follow the MNIST setup of Huang et al. (2024): a 2-layer fully-connected network is trained on MNIST distributed across $N = 200$ clients in a highly heterogeneous regime (each client holds data from at most two classes), with partial participation $p = 0.1$ and 10 local steps. We apply aggressive Top-1% compression to EF-style methods. As shown in Figure 14, SA-PEF closely tracks SCAFCOM and both substantially outperform standard EF and FedAvg, while uncompressed SCAFFOLD lies in between. This indicates that, under the same communication budget, SA-PEF can match the robustness of SCAFCOM's control-variate-plus-momentum design while retaining the simpler EF architecture (one residual per client, no additional drift-correction state).

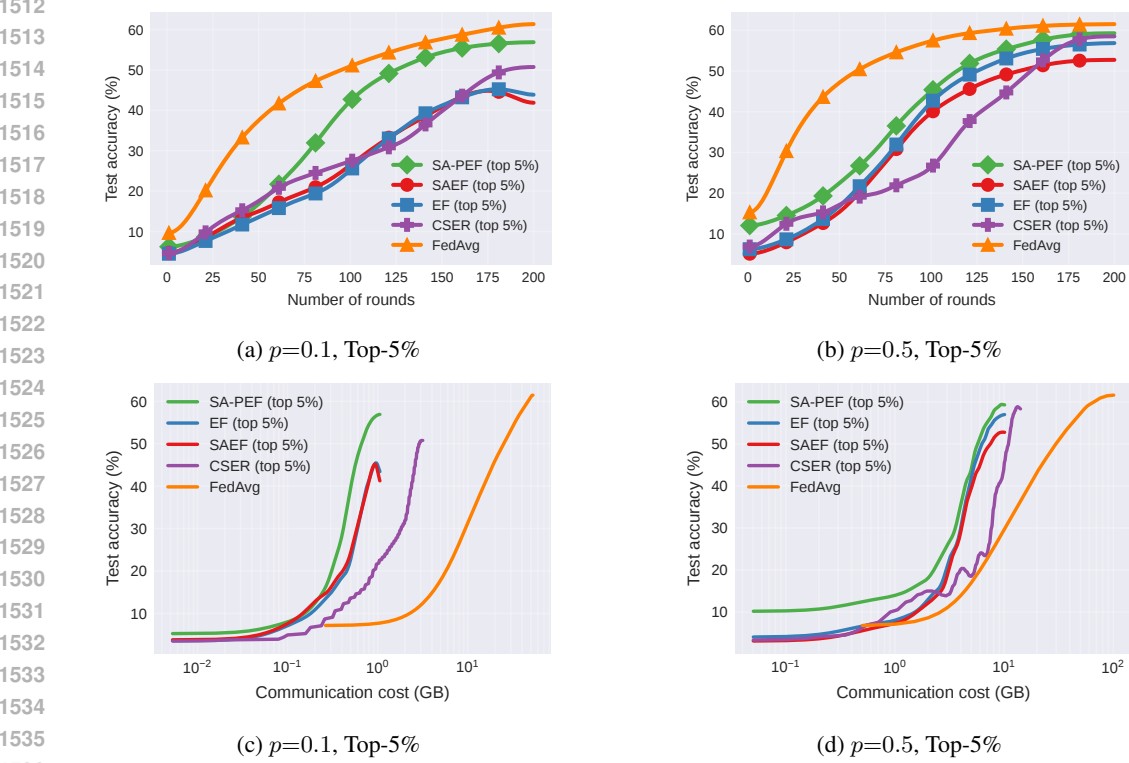

Figure 6: Test accuracy vs number of rounds (row 1) and communicated GB (row 2) on the CIFAR-100 dataset using ResNet-18 and $\gamma=0.1$.

| Algorithm | $\eta$ | $\eta_r$ |
|---|---|---|
| SA-PEF | 1 | $10^{-3}$ |
| SAEF | 1 | $10^{-3}$ |
| EF | 1 | $10^{-3}$ |
| SCAFCOM | 3 | $10^{-1}$ |
| FedAvg | 1 | $10^{-1}$ |

Table 3: Optimal global ($\eta$) and local ($\eta_r$) learning rate combinations.

To better align with our main experimental setup, we additionally report a preliminary CIFAR-10/ResNet-9 experiment under aggressive compression in the Figure 15. We follow the local mini-batch step formulation of Huang et al. (2024) (rather than local epochs) for a fair comparison. We use $K = 100$ clients and $R = 200$ communication rounds; at each round, the server samples 10 clients, and each selected client performs 20 local mini-batch SGD steps on a ResNet-9 model. For all EF-style methods (EF, SAEF, SA-PEF, and SCAFCOM) we apply Top-1% and Top-10% sparsification to the uplink updates, while FedAvg communicates dense updates. SCAFCOM uses control-variate and momentum coefficients $\alpha_{\mathrm{sc}} = 0.1$ and $\beta_{\mathrm{sc}} = 0.2$, selected via a small grid search. SA-PEF again behaves competitively with SCAFCOM while clearly improving over EF, SAEF, and FedAvg under the same communication budget. All curves are averaged over five independent runs with different random seeds. The learning-rate combinations that yield the highest test accuracy are listed in Table 3.

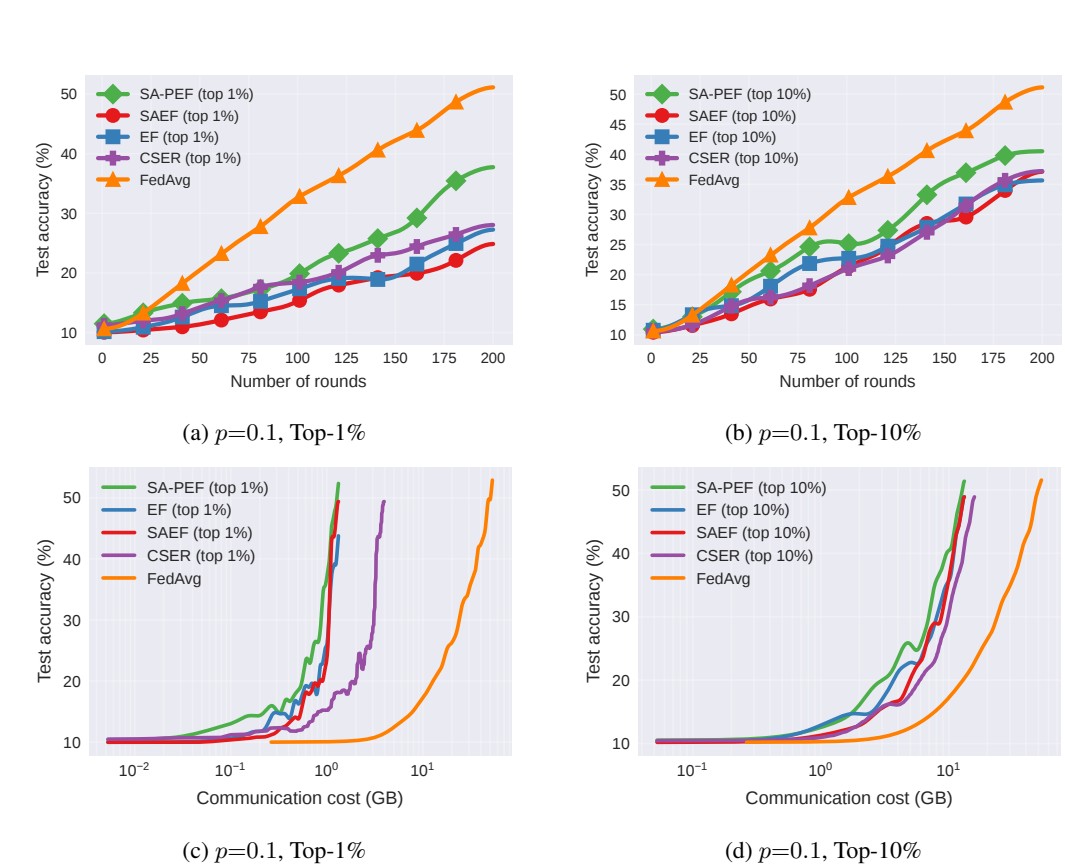

(a) $p$=0.1, Top-1%

(b) $p$=0.1, Top-10%

(c) $p$=0.1, Top-1%

(d) $p$=0.1, Top-10%

Figure 7: Test accuracy vs number of rounds (row 1) and communicated GB (row 2) on the CIFAR-10 dataset using ResNet-9 and $\gamma$=0.1.

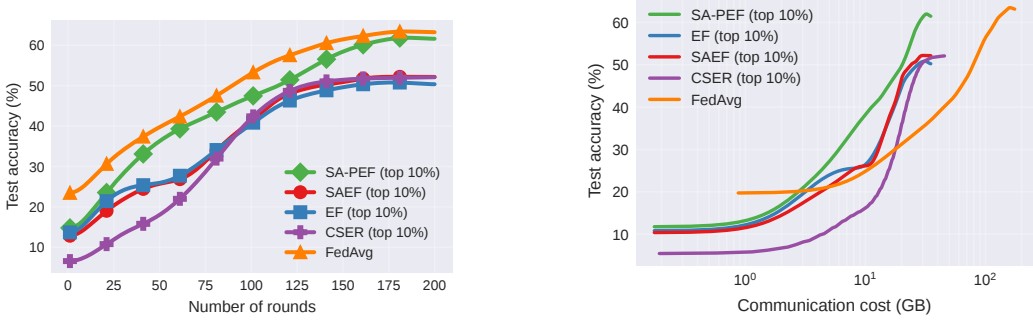

Figure 8: Test accuracy vs number of rounds (left) and communicated GB (right) on the Tiny-ImageNet dataset using ResNet-34 with $\gamma$=0.5, $p$=0.1.

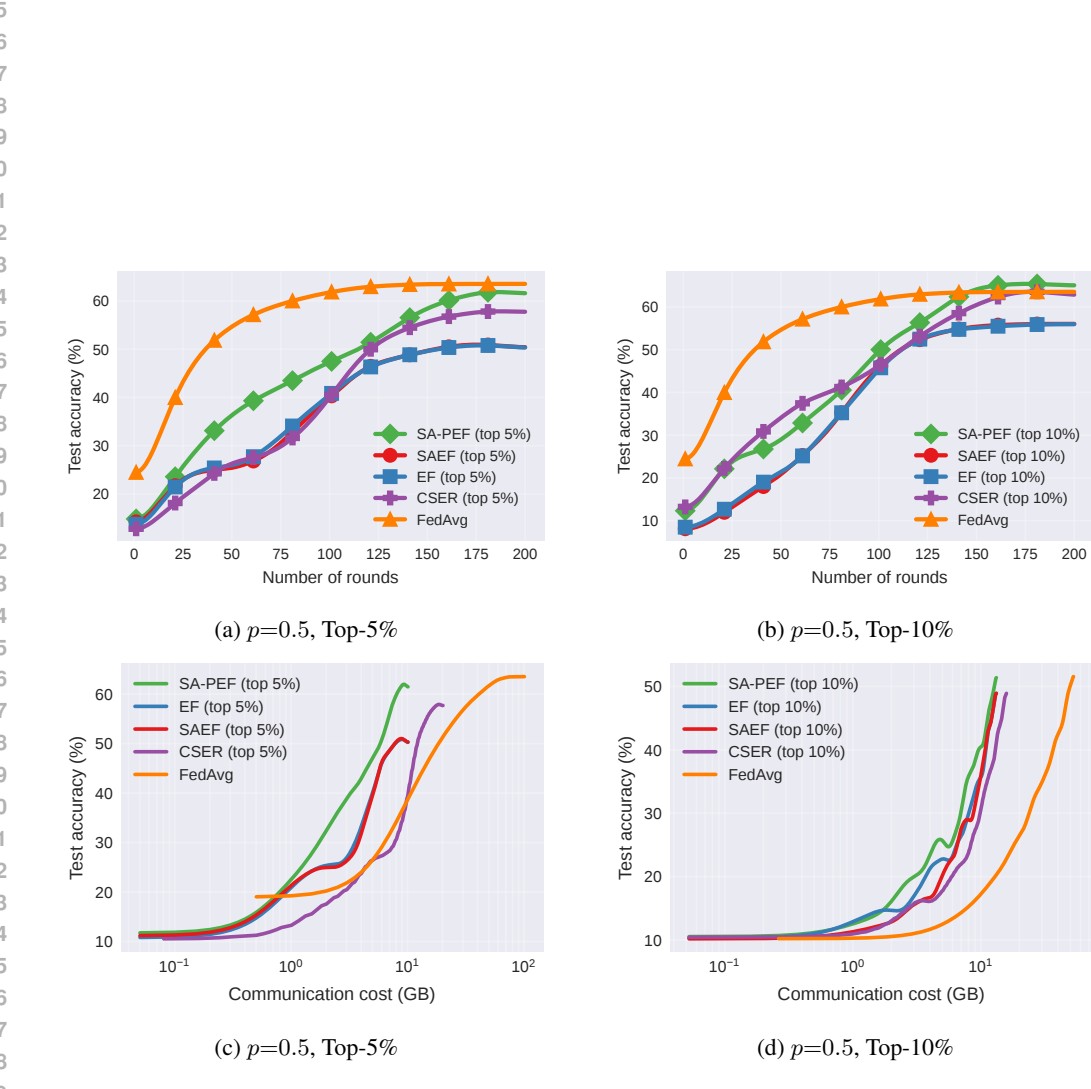

(a) $p$=0.5, Top-5%

(b) $p$=0.5, Top-10%

(c) $p$=0.5, Top-5%

(d) $p$=0.5, Top-10%

Figure 9: Test accuracy vs number of rounds (row 1) and communicated GB (row 2) on the CIFAR-100 dataset using ResNet-18.

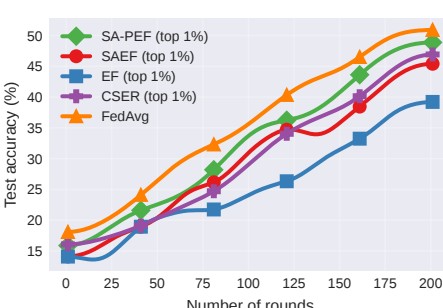 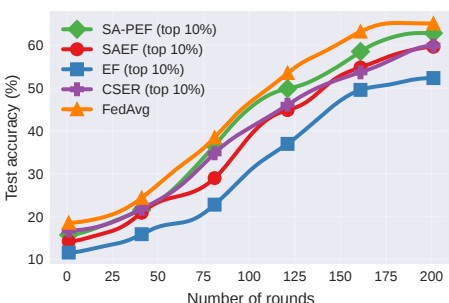

Figure 10: Test accuracy vs. number of rounds on CIFAR-10 with ResNet-9 under Top-1% (left) and Top-10% (right) uplink compression.

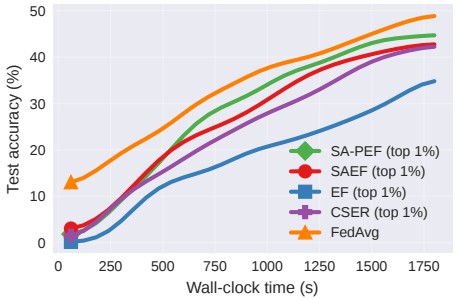 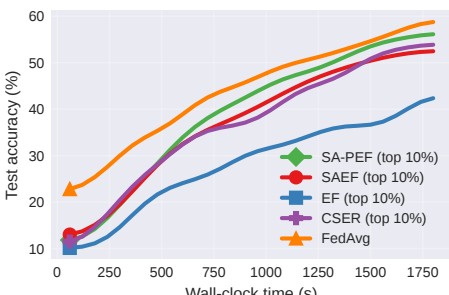

Figure 11: Test accuracy vs. wall-clock time (s) on CIFAR-10 with ResNet-9 under Top-1% (left) and Top-10% (right) uplink compression.

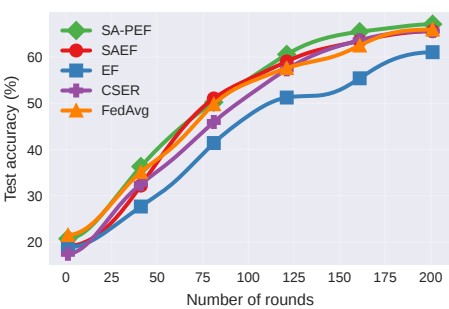 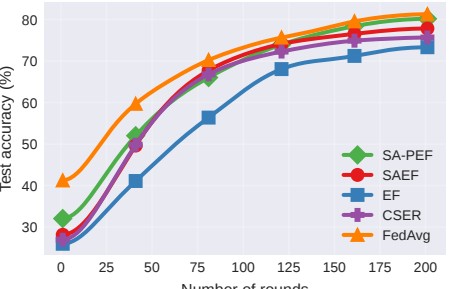

Figure 12: Test accuracy vs. number of rounds on CIFAR-10 with ResNet-9 for Dirichlet-$\gamma$ partitions: $\gamma = 0.1$ (left) and $\gamma = 0.5$ (right).

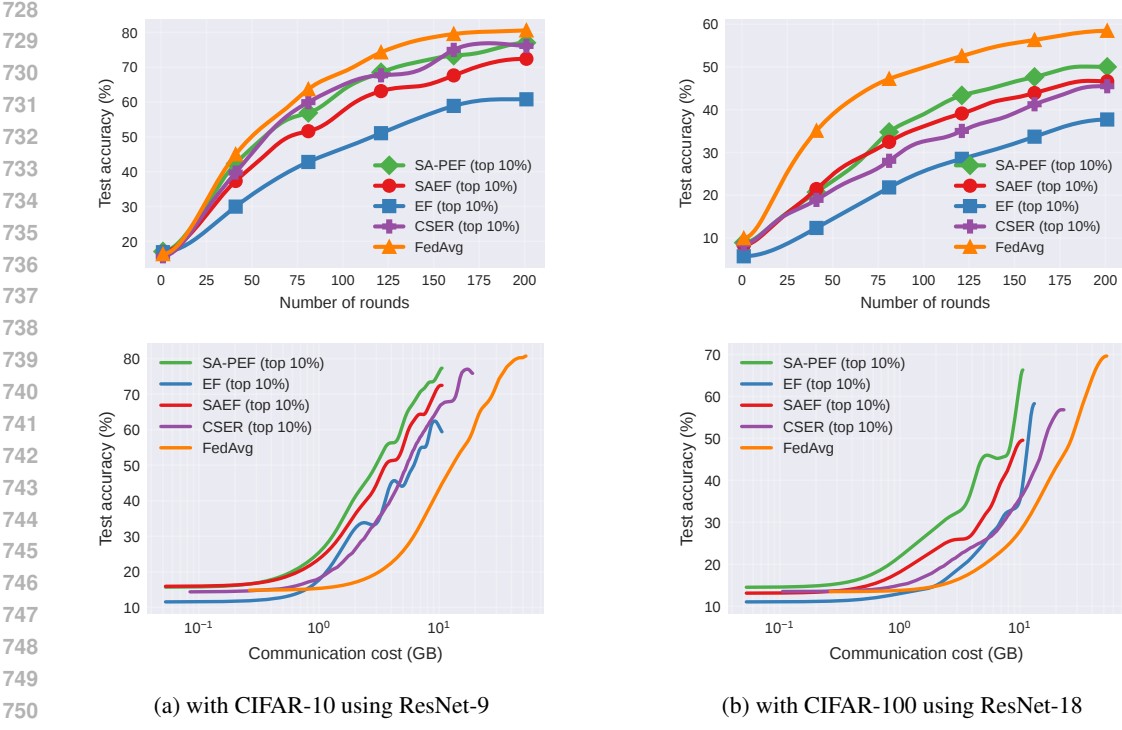

(a) with CIFAR-10 using ResNet-9          (b) with CIFAR-100 using ResNet-18

Figure 13: Test accuracy vs number of rounds (row 1) and communicated GB (row 2) using $p$=0.1, Top-10%, and $\gamma$=0.5.

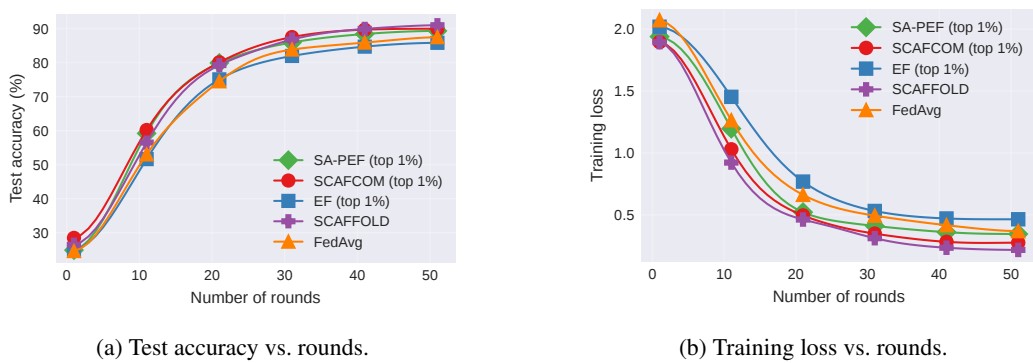

(a) Test accuracy vs. rounds.          (b) Training loss vs. rounds.

Figure 14: Comparison with SCAFCOM on MNIST under Top-1% compression, partial participation $p$=0.1, and 10 local steps.

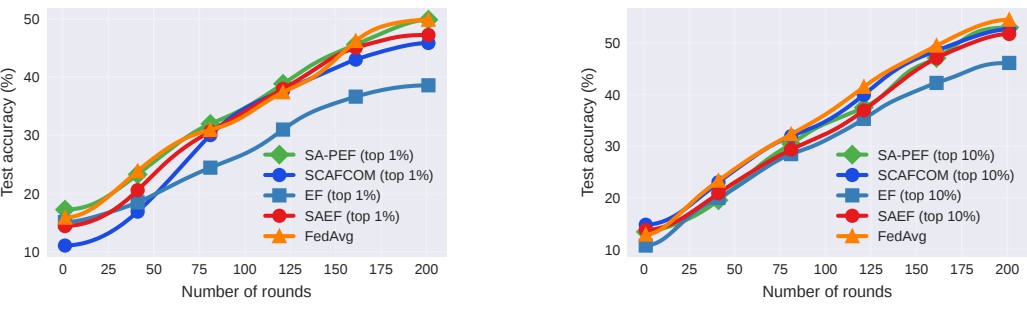

Figure 15: Test accuracy vs. number of rounds on CIFAR-10 with ResNet-9, partial participation $p$=0.1, and 10 local steps under Top-1% (left) and Top-10% (right) uplink compression.

