# OpenReview forum: "SA-PEF: Step-Ahead Partial Error Feedback for Efficient Federated Learning"
_ICLR.cc/2026/Conference — Submitted to ICLR 2026_

### Official Review · Reviewer_xyaS · 2025-10-19

**Soundness:** 3
**Presentation:** 2
**Contribution:** 3
**Rating:** 6
**Confidence:** 3

**Summary:**

The works presents a new combination of compression and error feedback with local steps, for nonconvex distributed optimization and federated learning

**Strengths:**

This is an important topic and new methods with theoretical guarantees are very welcome.

**Weaknesses:**

1) About the SOTA: "ProxSkip and its variance-reduced variants analyze local steps with general (including contractive) compressors via communication skipping and compressed exchanges".  Proxskip, more exactly Scaffnew, is an algorithm implementing local training, it should be mentioned after Scaffold in the previous paragraph. There is no compression in Proxskip/Scaffnew. There has been later works on combining Scaffnew with compression:

* CompressedScaffnew in Condat et al. "Provably Doubly Accelerated Federated Learning:
The First Theoretically Successful Combination of Local Training and Communication Compression"

* TAMUNA, which extends CompressedScaffnew to partial participation in Condat et al. "TAMUNA: Doubly accelerated distributed optimization with local training, compression, and partial participation"

* LoCoDL, which uses arbitrary unbiased compressors, not just permutation-based sparsifiers, in Condat et al. "LoCoDL: Communication-efficient Distributed Learning with Local training and Compression", ICLR 2025

These works focus on convex settings, acceleration from local training is lost in nonconvex settings, to the best of my knowledge.

2)  "This class includes many commonly used compressors, such as Top-k, Rand-k". If you want to make an unbiased compressor, such as rand-k, contractive, you have to scale it.

3) You have to be precise. When you say that SA-PEF reverts to EF when $\alpha=0$, is it EF21? The precise relationship should be discussed.

4) Line 267: You should not use $\xi$ for both the random realization in the stochastic gradient and the noise. Use a different symbol.

5) Assumption 3 is very restrictive. Methods such as EF21 do not require such an assumption.

6) "Our result matches the standard nonconvex picture for biased compression." This discussion deserves much more details. I am familiar with compression, error feedback, and local training, but just by looking at the result, I cannot interpret it. A table comparing existing results with the new ones would be very welcome. And there should be a whole discussion of how $\delta$, $K$, $\alpha$ and other parameters influence the convergence speed, according to the theoretical analysis.

**Questions:**

See above.

---

> ### Author Response · Authors · 2025-11-24
>
> We thank the reviewer for the helpful clarifications and pointers. Below we address each weakness.
>
> **W1. About the SOTA paragraph (ProxSkip/Scaffnew and compressed variants)**
>
> Thank you for the detailed clarification. You are correct that ProxSkip/Scaffnew implements local training without compression, and our original wording was imprecise. In the revised manuscript, we:
>
> 1. First present **SCAFFOLD** as the baseline control-variate method.
> 2. Then introduce **Scaffnew/ProxSkip** immediately afterwards, explicitly noting that it implements local training but does _not_ use compression.
> 3. Subsequently discuss compressed extensions built on the Scaffnew mechanism , namely **CompressedScaffnew**, **TAMUNA**, and **LoCoDL** with the corresponding citations.
>
> We also clearly state that these works focus on (accelerated) convex settings, whereas our work targets a nonconvex FL regime with biased compression and partial participation.
>
> **W2. Contractive compressors: Rand-(k) vs scaled Rand-(k)**
>
> We intended to refer to the standard **scaled** unbiased variant of Rand-(k), which is contractive. In the revised manuscript, we now explicitly write “**scaled Rand-(k)**” and clarify that the contractivity assumption applies to this scaled unbiased version (together with Top-(k)), rather than to the unscaled Rand-(k) operator.
>
> **W3. Equivalences and the relationship to EF vs EF21**
>
> Our intention was to relate SA-PEF to **classical EF**, not EF21. In the revised text, we clarify that:
>
> - When the step-ahead parameter is set to ($\alpha_r = 0$), **SA-PEF reduces exactly to standard EF** under the same compressor and residual recursion.
> - When ($\alpha_r = 1$), SA-PEF coincides with the **full step-ahead variant (SAEF)**.
>
> We also state explicitly that **EF21 is a different method**, with its own residual update and assumptions. Designing and analyzing an EF21-style step-ahead variant is an interesting direction, but it is beyond the scope of this work.
>
> **W4. Notation: stochastic gradient randomness vs noise**
>
> We agree that overloading the same symbol is confusing. In the revised manuscript, we:
>
> - Reserve ($\xi_{r,t}^{(k)}$) exclusively for the **stochastic gradient noise** in the analysis, and
> - Introduce a separate symbol ($\zeta$) for the **random data/sample** used in the stochastic gradients and in the gradient-mismatch metric.
>
> **W5. Assumption 3 (gradient dissimilarity) seems restrictive**
>
> Assumption 3 is the standard **bounded gradient-dissimilarity condition**, widely used in FL analyses with _local training and partial participation_ [1-3]. In our proofs, we invoke it only at two concrete points:
>
> 1. In **Lemma 2**, to bound local-model drift after ($\tau$) local steps, and
> 2. In **Lemma 3**, to control the second moment of the shifted update.
>
> In both cases, Assumption 3 is used to convert ($\tfrac{1}{K}\sum_k |\nabla f_k(w_r)|^2$) into a function of the global gradient plus a heterogeneity floor ($\nu^2$), which allows Theorem 1 and the partial-participation bound to be expressed in terms of ($|\nabla f(w_r)|^2$) plus constants.
>
> By contrast, **EF21** is analyzed in a different regime: workers compress full gradients (or gradient differences) at a shared iterate, with **no local steps and no partial participation**. In that synchronized setting, no client-drift terms arise, so no gradient-dissimilarity assumption is needed. SA-PEF, instead, targets **local SGD with partial participation and biased compression** in an FL setting, where local drift and heterogeneity play a central role and naturally motivate Assumption 3. We have clarified this distinction explicitly. Relaxing or removing Assumption 3 is orthogonal to the step-ahead mechanism and is an interesting direction for future work.
>
> [1] Mitra, A., Jaafar, R., Pappas, G. J., \& Hassani, H. (2021). Linear convergence in federated learning: Tackling client heterogeneity and sparse gradients. Advances in Neural Information Processing Systems, 34, 14606-14619.
>
> [2] Karimireddy, S. P., Kale, S., Mohri, M., Reddi, S., Stich, S., \& Suresh, A. T. (2020). Scaffold: Stochastic controlled averaging for federated learning. In International conference on machine learning, pp. 5132-5143.
>
> [3] Gafni, T., Cohen, K., \& Eldar, Y. C. (2024). Federated learning from heterogeneous data via controlled air aggregation with Bayesian estimation. IEEE Transactions on Signal Processing, 72, 1928-1943.

---

> > ### Author Response · Authors · 2025-11-24
> >
> > **W6. “Standard nonconvex picture for biased compression” and missing comparison table**
> >
> > We agree that the original sentence was too terse. In the revised manuscript, we:
> >
> > 1. Add a **comparison table** (in the appendix) summarizing, for each relevant method (EF, SAEF, CSER, SCAFCOM, and SA-PEF):
> >
> >    - The setting (convex vs nonconvex, centralized vs FL, local steps, partial participation),
> >    - The compressor class (unbiased vs biased, contractive, use of momentum/control variates), and
> >    - The resulting nonconvex rate and error floor, highlighting the dependence on smoothness (L), variance level ($\sigma^2$), compressor contractivity/ bias, participation rate (p), and number of local steps (T).
> >
> > 2. Expand the discussion following our main theorem to spell out what we mean by the “standard nonconvex picture for biased compression”:
> >
> >    - A **leading optimization term** of order ($O\big((p,\eta,\eta_0 T R)^{-1}\big)$) under smoothness, and
> >    - An **error floor** whose radius depends on data heterogeneity and compression-induced residual energy (scaled by the compressor contractivity parameter and participation).
> >
> > Hence, it **matches the leading nonconvex order** of EF-style methods while improving the residual-contraction factor and thus tightening the compression- and heterogeneity-driven floors under the same assumptions.

---

> ### Comment · Reviewer_xyaS · 2025-11-26
>
> Thank you for taking into account my comments. Note that there is a follow-up paper about EF21 with some extensions, including partial participation: "EF21 with Bells & Whistles: Six Algorithmic Extensions of Modern Error Feedback*, JMLR 2025.
>
> I agree that it is not straightforward at all to combine local steps and compression/error feedback. However, I don't get why you consider standard EF, while EF21 is superior, and the analysis is actually quite simple.
>
> For the moment I plan to keep my score unchanged

---

> ### Author Response · Authors · 2025-11-27
>
> Thank you for this follow-up and for pointing us to the EF21 extensions in Fatkhullin et al.~[1]. We fully agree that, in the standard distributed setting with $T{=}1$ (communication every step, compression of full gradient or gradient differences at a shared iterate), EF21 is strictly preferable to classical EF. This is mainly because EF21 admits a clean contraction analysis, removes the error floor, and is widely regarded as a state-of-the-art error-feedback framework in that regime. In the revised manuscript, we have updated the related-work section to discuss EF21 and its partial-participation extension explicitly.
>
> Our decision to build on standard EF (via Fed-EF) is driven by the specific regime we target: _local SGD with $T>1$ local steps, partial participation, and biased contractive compressors_. In this setting, the compressed object is no longer a single noisy gradient at a shared iterate, but the accumulated local update over $T$ steps on heterogeneous data. The EF21 analysis crucially exploits the Markov structure of compressed (full) gradient or gradient differences at a common iterate. Once we move to accumulated local updates and drifting local models, this structure does not carry over. Therefore, extending EF21-style arguments to this local-SGD + PP + biased-compression regime is non-trivial and, to our knowledge, has not yet been carried out.
>
> Step-Ahead Error Feedback (SAEF)~[2] was introduced in the centralized, homogeneous, $T{=}1$ setting precisely to mitigate the gradient-mismatch effect of standard EF and accelerate training. Their analysis shows an improved compression-error term, and their experiments already demonstrate that, on harder tasks, SAEF benefits from periodic averaging to keep mismatch under control. In our federated local-SGD regime with aggressive compression and low client participation, we observe a related but more pronounced phenomenon: the full step-ahead variant ($\alpha_r{=}1$), which is structurally analogous to SAEF but applied to local updates, can _amplify_ gradient mismatch. Both our theory (via a worse residual-contraction factor when $s_r=\eta_rLT$ is not small) and experiments show that full step-ahead exhibits larger gradient-mismatch spikes and late-stage plateaus than EF and SA-PEF in these extreme regimes.
>
> By contrast, Fed-EF~[3] already adapts classical EF to exactly this setting we study (local SGD with partial participation and biased contractive compressors), and SA-PEF is designed as an extension of this Fed-EF backbone. It reduces to Fed-EF when $\alpha_r{=}0$, recovers full step-ahead behavior when $\alpha_r{=}1$, and for interior $\alpha_r\in(0,1)$ achieves a provably smaller residual-contraction factor that avoids the mismatch amplification seen with full step-ahead.
>
> In the revised related-work section, we clarify that EF21 with the extensions introduced by Fatkhullin et al.\ [1] is the preferred modern error-feedback framework for $T{=}1$ gradient compression, whereas SA-PEF is an improvement over Fed-EF in the federated local-SGD setting. Designing and analyzing an EF21-style step-ahead variant in this regime is, in our view, an interesting but orthogonal direction for future work.
>
> [1] Fatkhullin, I., Sokolov, I., Gorbunov, E., Li, Z., \& Richtárik, P. (2025). EF21 with bells \& whistles: Six algorithmic extensions of modern error feedback. JMLR 26(189), 1-50.\
> [2] Xu, A., Huo, Z., \& Huang, H. (2021). Step-Ahead Error Feedback for Distributed Training with Compressed Gradient. AAAI 35(12), 10478-10486.\
> [3] Li, X., \& Li, P. (2023). Analysis of error feedback in federated non-convex optimization with biased compression. ICML, 19638-19688.

---

### Official Review · Reviewer_WBBa · 2025-10-27

**Soundness:** 3
**Presentation:** 3
**Contribution:** 3
**Rating:** 6
**Confidence:** 3

**Summary:**

The paper tackles communication-efficient FL under non-IID data by refining error-feedback (EF) with biased compressors, which can stall early due to residuals aligning with local gradients and causing gradient mismatch. The proposed Step-Ahead Partial Error Feedback (SA-PEF) introduces a per-round weight $\alpha\in[0,1]$ to partially preview the EF residual: each client shifts by $\alpha e_r^{(k)}$ before local SGD and carries $(1-\alpha)e_r^{(k)}$ via standard EF, recovering EF at $\alpha=0$ and SAEF at $\alpha=1$. The authors prove non-convex convergence with local steps, partial participation, and $\delta$-contractive compressors, achieving $O(1/(TR))$ to a floor; compression effects enter through $(1-1/\delta)$ and a new residual-contraction factor $\rho_r<1$. An optimal $\alpha_r^*$ (near 1 when local work is small) minimizes $\rho_r$, explaining faster early progress than EF while retaining EF’s stability. Empirically, on CIFAR-10/100 and Tiny-ImageNet with Top-$k$ sparsification and $K=100$ clients, SA-PEF reaches target accuracy in fewer rounds and fewer bits than EF/SAEF; SAEF often plateaus, whereas SA-PEF continues improving. A gradient-mismatch probe shows SA-PEF keeps mismatch near zero throughout training. Overall, SA-PEF blends step-ahead warm-up with EF stability, improving accuracy-communication trade-offs across datasets and settings.

**Strengths:**

- Novel Algorithmic Idea: SA-PEF is a simple yet effective interpolation between two known techniques (EF and SAEF). Introducing a tunable $\alpha$ for partial residual preview is an elegant way to control the early-stage acceleration vs. noise trade-off. The method requires minimal changes to existing federated SGD with error feedback, making it a practical drop-in enhancement.
 - Theoretical Rigor: The paper presents a nontrivial convergence analysis under realistic FL conditions. Theorem 1 (with proof in Appendix) covers smooth nonconvex objectives with local SGD and partial client sampling, assuming $\delta$-contractive compressors. The bounds recover standard FedSGD rates $O((\eta\eta_0TR)^{-1})$ up to constants, and explicitly quantify the effect of $\alpha$ via a residual contraction $\rho_r<1$. This provides insight into how and why a partial preview improves convergence (smaller residual error constants). To our knowledge, this is the first formal analysis of step-ahead error feedback in a federated/local-updates setting with non-IID data and biased compression. The theory also extends to partial participation (Remark 1), transparently showing the $1/p$ slow-down in optimization as usual.
 - Empirical Validation: Experiments are extensive and well-designed. The authors test across multiple benchmarks (CIFAR-10/100, Tiny-ImageNet) and models, with both IID and strongly non-IID partitions. They compare against relevant baselines (FedAvg, EF, SAEF, CSER) under identical hyperparameter settings, isolating the effect of the SA-PEF modification. The results consistently favor SA-PEF: target accuracies are reached in fewer rounds and less communication. Notably, SA-PEF avoids the late-stage plateaus seen in SAEF, confirming its stability advantage. The gradient-mismatch metric in Figure 3 further supports the claim that partial preview keeps training trajectories well-aligned. The paper also reports thorough implementation details and reproducibility materials (configs, code) in the appendix, which is commendable.
 - Clarity and Insights: The authors clearly explain the motivation and algorithm, including intuitive benefits of partial preview (e.g. “removes most early-round mismatch while injected noise decays as $|e_r|$ shrinks”). The theoretical analysis yields practical takeaways: for small local-work $s_r=\eta_rLT$, the optimal $\alpha^*_r\approx 1/(1+12s_r^2)$ is near 1, giving the strongest contraction. This yields a concise interpretation: SA-PEF improves initial descent constants, leading to a steeper objective decrease under the same stepsizes. These insights bridge theory and experiment nicely.

**Weaknesses:**

- Hyperparameter $\alpha_r$ Selection: SA-PEF introduces the new parameter $\alpha$ (or schedule ${\alpha_r}$). The paper relies on choosing a fixed $\alpha$ “near the theory-predicted optimum”, but offers little guidance on how to pick $\alpha$ in practice. The theoretical $\alpha_r$ depends on unknown smoothness constants and local-work $s_r$, and the paper uses a constant $\alpha$ across empirical training. It would be helpful to see sensitivity analysis: how critical is the exact value of $\alpha$? If $\alpha$ is set too large the residual contraction worsens; if too small, warm-up is limited. The paper mentions decaying $\alpha_r$ as future work but does not experiment with any schedule (beyond “near-optimal fixed”). In practice, tuning $\alpha$ adds overhead, and it is unclear how robust the method is to mis-specification.
 - Theoretical Assumptions and Complexity: The convergence proof requires several technical conditions (smoothness, bounded gradient variance, and a small initial step condition $s_0\le1/8$). While standard, these assumptions may limit the direct applicability of the stated rates. The bounds also involve many constants ($C_\sigma, C_\nu, E_{\max}$, etc.), making it hard to gauge practical significance. In addition, the conclusions hinge on $\Theta\le1/2$ (see Theorem 1). It may be worth clarifying in the paper whether these conditions are reasonable for typical FL hyperparameters. Moreover, the improved residual contraction factor $\rho_r$ is derived assuming $s_r\le1/8$; if larger local steps are used, it is unclear if SA-PEF still helps.
 - Limited Baseline Comparisons: The experimental evaluation, while solid, omits some potentially relevant baselines. For example, adaptive compression methods (DIANA/MARINA) or other gradient sparsification schemes are not considered. While EF, SAEF, and CSER are appropriate comparisons, it would strengthen the claim to include at least one variance-reduced or unbiased compressor scheme (e.g. QSGD) to contextualize performance. Similarly, the interaction of SA-PEF with other heterogeneity mitigation techniques (FedProx, SCAFFOLD, FedDyn, etc.) is unexplored. It is unclear whether SA-PEF could be combined with those methods, or if its benefits persist in their presence.
 - Scope of Experiments: The paper focuses exclusively on vision benchmarks with ResNets. It would increase confidence if at least one non-vision task or a larger-scale dataset were tested. Although the chosen datasets (CIFAR-10/100, Tiny-ImageNet) cover increasing complexity, they are still relatively small. The gains might differ on more complex or larger-scale problems (e.g. NLP tasks, larger models). Additionally, all compressors tested are Top-$k$; it would be useful to know if SA-PEF helps with quantization or other biased compressors (though theory permits any $\delta$-contractive operator).
 - Marginal Gains in Mild Regimes: From the presented results, the benefits of SA-PEF are largest under extreme compression and heterogeneity (e.g. Top-1%, $\gamma=0.1$). In more moderate settings (less compression or more IID data), SA-PEF’s advantage over EF might be smaller. The paper should clarify whether SA-PEF is recommended primarily for such challenging regimes, and whether there is any scenario where it might underperform (e.g. if $\alpha$ is chosen poorly).

**Questions:**

1. Can you provide more insights on the theoretical selection of $\alpha_{r}$ without knowing implicit constant including $s_r$? Empirically what is the selection of the fixed $\alpha$ value in each experiment setting? How sensitive are results to this choice? Did you experiment with decaying $\alpha_r$ over rounds (e.g. larger early, then smaller) as suggested by your “moderate or decaying” note? Showing results of ablation studies on different $\alpha$ and $\alpha_r$ schedules would be informative.
2. The main result hinges on $\Theta\leq\frac{1}{2}$ and $s_{0}(s_{r})\leq\frac{1}{8}$. Can you give practical sufficient conditions in terms of hyperparameters (such as $\eta,\eta_{0},T,\delta,K$, etc.) that ensure this holds, and confirm numerically that your chosen hyperparameters satisfy them? How does the residual contraction factor $\rho_r$ behave if $s_r$ or $\Theta$ exceeds the stipulated bound? What will be the corresponding convergence results? Empirically do you observe degradation or instability? Is there a modified schedule (e.g., shrinking $\ta_0$ or $\alpha_r$) that restores $\rho_r<1$?
3. You mention in conclusion that integrating SA-PEF with adaptive optimizers or momentum is future work. Do you have any preliminary insights on this? Also, could SA-PEF be combined with control-variate methods like SCAFFOLD or MARINA? Would the partial preview idea remain effective in those contexts?
4. While the theory allows any $\delta$-contractive compressor, the experiments only show Top-$k$ sparsification. Have you tried SA-PEF with quantization (e.g. QSGD) or other biased compressors? Does the step-ahead mechanism provide similar benefits there?
5. How does SA-PEF perform when data is IID across clients, or when compression is mild? The appendix mentions IID control experiments; could you summarize whether SA-PEF still helps, or if it simply matches EF in those cases? Theorem 1 implies a $1/p$ slow-down for participation rate $p$. In practice, how does SA-PEF behave when $p$ is very low (e.g. $p<0.1$)? Is the advantage over EF maintained at very small participation fractions?

---

> ### Author Response · Authors · 2025-11-24
>
> We thank the reviewer for the detailed and constructive feedback. Below we address each weakness and question.
>
> **W1. Hyperparameter $\alpha$ selection**
>
> While the exact optimum $\alpha^\star$ depends on problem constants, our analysis and experiments together suggest a simple practical rule. Proposition 1 shows that SA-PEF strictly improves the residual contraction factor over EF whenever $\alpha_r \in \Bigl(0,\; \frac{1}{1+12 s_r^2}\Bigr), \quad s_r = \eta_r L T$.
> If the local stepsize is chosen in the usual “small” regime ($s_r \le 1/8$), then the preferred value $\alpha_r^\star = 1/(1+12s_r^2)$ lies close to $1$ (roughly $0.84$–$1$). Thus, once $(\eta_r, T)$ are fixed in a standard stability range, it is enough to select $\alpha$ in a high-$\alpha$ neighborhood (e.g., $\alpha \approx 0.8$–$0.9$) to reduce $\rho_r$ relative to EF, without needing accurate estimates of $L$.
>
> In the revised manuscript, we made this guidance explicit and added $\alpha$-sensitivity plots (constant $\alpha \in {0.1,\dots,1.0}$). These show that SA-PEF is robust over a broad band ($\alpha \approx 0.7$–$0.9$), with degradation mainly at the extremes where very small $\alpha$ behaves like EF and $\alpha=1$ coincides with SAEF. We therefore recommend treating $\alpha$ like a momentum parameter: after choosing $(\eta_r, T)$, a single moderate value such as $\alpha \approx 0.8$ works well across our tasks without heavy tuning. In this work we focus on constant $\alpha$; more sophisticated decaying or adaptive schedules suggested by Lemma 4 are a natural extension but left for future work, since a fixed moderate $\alpha$ already captures most of the practical benefit in our experiments.
>
> **W2. Theoretical assumptions and complexity**
>
> We agree that the assumptions and constants should be easier to interpret in terms of typical FL hyperparameters, and that the role of $\tau$ should be clearer. In our analysis, hyperparameters enter primarily through the effective local stepsize $s_r = \eta_r L T$ and heterogeneity parameters $(\beta, \nu)$. The drift and small-stepsize conditions in Theorem~1 can be written as simple inequalities on $\eta_r T$; for example, from $18\beta^2 s_r^2 \le 1/8$ and $s_r \le 1/8$, we obtain $\eta_r T \le \min\bigl( 1/(12\beta L), 1/(8L) \bigr).$
>
> Practically, in all experiments we use a cosine-decay schedule for the client stepsize, starting from an aggressive initial value and decaying to a small value. Early in training, $\eta_r T$ can be relatively large, so the conservative bound $s_r \le 1/8$ need not hold for arbitrary $(L,\beta)$; however, as $\eta_r$ decays, $s_r = \eta_r L T$ decreases, and for typical $L$ we eventually enter a regime where $s_r \le 1/8$ and the sufficient conditions of Theorem~1 are satisfied.
>
> Regarding contraction, Proposition 1 gives $$\rho_r = \Bigl(1-\tfrac1\delta\Bigr) \Bigl(2(1-\alpha_r)^2 + 24\alpha_r^2 s_r^2\Bigr).$$ so for fixed compressor and $\alpha_r$, $\rho_r$ is monotone in $s_r$. Our analysis therefore targets the regime of **moderate local work**, where $s_r$ is small enough that $\rho_r < 1$ and SA-PEF strictly improves residual contraction relative to EF. If $T$ (and hence $s_r$) is taken very large, the bound can indeed allow $\rho_r \ge 1$, in which case the theory no longer guarantees contraction, this is a limitation of EF-style methods under very aggressive local training, not specific to SA-PEF. Empirically, when we deliberately increase $\eta_r$ or $T$ beyond our tuned grid, we observe larger residual norms and gradient mismatch; under our cosine schedule, the decay in $\eta_r$ keeps $s_r$ small and $\rho_r < 1$ in the later rounds where convergence is assessed.
>
> **W3. Limited baseline comparisons**
>
> In this work we deliberately focus on EF/SAEF/CSER as baselines because they share the same lightweight EF-style architecture as SA-PEF: biased contractive compression, no control variates, and no additional per-client state at the server. This isolates the effect of the step-ahead mechanism on EF-style methods.
>
> Variance-reduced and unbiased-compression schemes such as DIANA, or MARINA and heterogeneity-mitigation methods such as FedProx, SCAFFOLD, and FedDyn, target different error sources (unbiased compression variance or data-heterogeneity drift) and typically require additional state or changes to the local objective. SA-PEF is conceptually **complementary** to these techniques rather than a replacement: it can be applied to the compressed update in an EF-style pipeline, while methods like FedProx/SCAFFOLD/FedDyn modify the local training dynamics. In the revised manuscript, we clarify this positioning and add a short discussion on how SA-PEF could be combined with such methods. A full empirical study of SA-PEF hybrids with DIANA/MARINA or FedProx/SCAFFOLD/FedDyn is beyond the scope of this paper and is highlighted as future work.

---

> > ### Author Response · Authors · 2025-11-24
> >
> > **W4. Scope of experiments and compressor types**
> >
> > We agree that broader coverage would further support generality. Our current experiments focus on standard vision benchmarks (CIFAR-10/100, Tiny-ImageNet) and ResNet models because these are widely used in the compressed-FL literature and expose challenging regimes of non-IID data, local steps, and aggressive sparsification.
> >
> > Regarding compressor types, we chose Top-$k$ sparsification as a canonical example of a biased contractive compressor, but our analysis only requires a $\delta$-contractive operator and thus covers a range of biased schemes (including biased quantizers). To address this empirically, we will add a sanity-check experiment in the appendix on CIFAR-10 with a standard biased quantizer under the same setting as our Top-$k$ runs, and report the resulting accuracy–communication trade-offs. This verifies that the qualitative gains of SA-PEF are not specific to Top-$k$, while keeping the experimental scope manageable.
> >
> > **W5. Marginal gains in mild regimes**
> >
> > We agree that SA-PEF’s largest empirical gains occur in the most challenging regimes, with strongly non-IID data and aggressive compression (e.g., Top-1%). At the same time, SA-PEF does not hurt performance in milder regimes: in our IID control experiment (Fig. 8, IID, Top-5% compression), SA-PEF tracks or slightly improves over EF throughout training. We want to clarify that SA-PEF is designed as a **safe drop-in (backward-compatible enhancement)** for EF-style methods: it brings clear benefits when compression and heterogeneity are high, while behaving similarly to EF when these factors are mild.
> >
> > **Q1. Selecting and scheduling $\alpha$**
> >
> > Please see the response to W1.
> >
> > **Q2. Practical sufficient conditions for $\eta L \tau \le 1$ and $\rho(\alpha) < 1$**
> >
> > Please see the response to W2.
> >
> > **Q3. Combination with momentum, adaptive optimizers, and control variates**
> >
> > In our experiments we already use SGD with momentum $0.9$, and SA-PEF remains stable with the same method ranking as in the plain-SGD theory. A $\beta = 0$ ablation (no momentum) yields very similar accuracy-per-bit and preserves the ranking, indicating that step-ahead preview is compatible with standard momentum in practice. We clarify this in the revised manuscript.
> >
> > We have not yet run experiments with adaptive optimizers (e.g., Adam) or with control-variate methods such as SCAFFOLD or MARINA. Conceptually, these directions are orthogonal: control variates and adaptive preconditioning target data-heterogeneity drift or gradient scaling, while SA-PEF targets compression-induced residual mismatch. We therefore expect partial preview to remain effective in such hybrids whenever compressed communication is present.
> >
> > **Q4. Other compressors beyond Top-$k$**
> >
> > Please see the response to W4.
> >
> > **Q5. IID / mild compression and very low participation**
> >
> > Please see the response to W5.

---

### Official Review · Reviewer_x7FE · 2025-10-28

**Soundness:** 3
**Presentation:** 3
**Contribution:** 1
**Rating:** 6
**Confidence:** 4

**Summary:**

The paper studies federated learning with **biased uplink compression** and **multiple local SGD steps** (a setting where,as authors claims, vanilla error feedback degrades or diverges). It proposes **SA-PEF** (“Step-Ahead Partial Error Feedback”), which sends a fraction $\alpha_r\in[0,1]$ of the *next* client update together with the current compressed residual, modifying the residual recursion (Eq. (12)–(15)) and tightening the residual contraction factor (Lemma 4) relative to EF. Under $L$-smoothness, bounded stochastic variance $\sigma^2$, and a standard gradient dissimilarity parameter $\nu^2$ (Assumptions 1–3), **Thm. 1** proves a non-convex rate
$$
\frac{1}{R}\sum_{r=0}^{R-1}\mathbb{E}\|\nabla f(w_r)\|^2
\le
\mathcal{O}\left(\frac{f(x_0)-f^\star}{p\,\eta\,\eta_0\,T\,R}\right)
+\tilde{\mathcal{O}}\left(\frac{\sigma^2}{m}+\left(1-\frac{1}{\delta}\right)(\sigma^2+\nu^2)\right).
$$

with partial participation fraction $p$ and $\delta$-contractive compressor; the leading $1/(p\eta\eta_0TR)$ term matches EF up to constants, while the step-ahead choice $\alpha_r^\star=\frac{2s_r}{1+12s_r^2}$ (with $s_r=\eta_rLT$) reduces the residual contraction constant $\rho_r$ compared to EF (Lemma 4, Prop. 1). Empirically, on CIFAR-10/100 and Tiny-ImageNet with Top-$k$ sparsification and $p\in\{0.1,0.5,1.0\}$, SA-PEF typically reaches target accuracy in fewer rounds (and fewer bits) than EF/CSER and tracks SAEF without late-stage plateaus (Figs. 1,4–7; fairness/tuning protocol specified).

**Strengths:**

- **Clear algorithmic modification and analysis.** The step-ahead mechanism is precisely defined (Eq. (12)–(15)) and its effect on the residual is quantified via a refined contraction factor $\rho_r$, including the analytic optimum $\alpha_r^\star$ (Lemma 4).
- **Non-convex convergence with partial participation.** Thm. 1 extends to client sampling; the averaged bound explicitly shows the **$1/p$ rounds slowdown**, a $1/m$ variance reduction for mini-batch noise, and residual-induced floors scaling with $(1-1/\delta)$. The analysis explains EF stalling when $p$ is small and compression is aggressive (Remark on $\rho_r^{\mathrm{PP}}$).
- **Experimental protocol is unusually explicit and fairness-minded.** Shared grids, matched bit budgets, shared sampling seeds, and reporting **accuracy vs. rounds and vs. communicated GB** are specified (including how Top-$k$ bit-cost is computed).
- **Breadth of settings.** Results span multiple datasets, heterogeneity levels (Dirichlet $\gamma$), participation rates $p$, and compression levels; IID controls and $\alpha$ ablations are included (App. B; Figs. 4–8).
- The paper includes a valuable diagnostic analysis of gradient mismatch (Sec. 5.3, Fig. 3). This provides strong empirical evidence for *why* the partial step-ahead mechanism works, showing it effectively keeps the mismatch low and stable throughout training, in contrast to both pure EF and SAEF.
- The experimental evaluation (Sec. 5) is comprehensive in terms of testing under difficult FL conditions, including high data heterogeneity ($\gamma=0.1$) and low client participation rates ($p=0.1$), which are common bottlenecks for many FL algorithms.

**Weaknesses:**

- **Missing key baselines / positioning.** Two closely related works—**Fed-EF** (error feedback with local steps and biased compression) by Li & Li (2022) and **SCAFCOM/SCALLION** (Huang, Li & Li, 2023)—are not compared experimentally nor discussed in depth. Li & Li analyze EF with local steps and partial participation and show when SAEF can diverge while EF remains stable; Huang et al. present algorithms supporting **arbitrary heterogeneity, partial participation, and local updates**, with unbiased and biased compressors (via momentum), and emphasize avoiding extra assumptions on compressor errors.

Huang, Xinmeng, Ping Li, and Xiaoyun Li. "Stochastic controlled averaging for federated learning with communication compression." _arXiv preprint arXiv:2308.08165_ (2023).

Li, Xiaoyun, and Ping Li. "Analysis of error feedback in federated non-convex optimization with biased compression." _arXiv preprint arXiv:2211.14292_ (2022).


- **No improvement in the leading complexity term.** Thm. 1’s leading rate scales as $\mathcal{O}\big((p\,\eta\,\eta_0TR)^{-1}\big)$—**identical to EF up to constants**; the benefit is strictly via a smaller $\rho_r$ constant (and thus smaller residual-induced floors). The paper should state explicitly that there is **no asymptotic speedup**, only better constants.
- **Assumptions appear not relaxed vs. EF; possibly stricter than some alternatives.** SA-PEF relies on **gradient dissimilarity** (Assumption 3, parameter $\nu$) and $\delta$-contractive compressors. In contrast, Huang et al. claim convergence under smoothness and bounded variance **without dissimilarity** for their unbiased variant, and accommodate biased compression with momentum control (SCAFCOM). The manuscript should clarify whether its assumptions are weaker/stronger than these and why they are necessary.
- **Partial-participation constants opaque.** The PP bound introduces $\Theta_{\mathrm{PP}}$ and constants $C_\sigma,C_\nu$ depending on $(\delta,\alpha_r,s_r)$, but their **explicit forms and practical ranges** are not fully enumerated; the condition $\Theta_{\mathrm{PP}}\le \tfrac12$ is asserted but not instantiated with realistic hyperparameters (Thm. 1, App. A).
- **Ablations are incomplete for $\alpha$.** While constant-$\alpha$ sweeps are mentioned, there is no **systematic sensitivity** across $(T,p,\delta)$ to validate the theory-suggested $\alpha_r^\star(s_r)$, nor schedules that adapt $\alpha_r$ as $s_r$ changes. This makes it hard to attribute gains specifically to the theoretical mechanism.
- **Missing robustness checks.**  Error bars or multi-seed statistics are not shown in the visible figures; the paper states shared seeds and fairness controls but does not report mean±std or significance.
- **Scale and wall-clock accounting.** Experiments fix $K=100$, $R=200$, and do not report **wall-clock** or the extra compute/latency due to step-ahead packing; downlink is omitted (stated identical across compressed methods), but a **time-to-accuracy** analysis would strengthen claims.
- The work does not solve the fundamental limitation of error feedback under partial participation. The analysis (App. A.3) acknowledges that SA-PEF inherits the same $\sqrt{n/m}$ slowdown factor (using notation from Li & Li (2023), where $n$ is total clients and $m$ is participating clients) identified by Li & Li (2023). This limitation arises because in partial participation, inactive clients do not update their error accumulators. When an inactive client rejoins training after several rounds, the error it feeds back is "stale"—it corresponds to the compression error from a much older model state. Applying this stale error correction to the current gradient update can misdirect the optimization process, leading to the theoretically predicted slowdown. This "stale error compensation" effect is a known drawback of the EF mechanism in FL, which competing methods like SCAFCOM (Huang et al., 2024) are specifically designed to overcome by using control variates that evolve even for inactive clients. SA-PEF is thus an incremental improvement on EF, rather than a solution to its core structural problems in the partial participation setting.

### Formatting & Presentation Issues
- Some constants/conditions are introduced but not instantiated numerically (e.g., $\Theta_{\mathrm{PP}}$, $C_\sigma,C_\nu$).
- Figures in the appendix appear without uncertainty bands and with small fonts in the visible snippets; ensure readability and add error bars where possible (Figs. 4–7).
- Minor: ensure consistent notation for participation ($p$ vs. $q$) and list **all** hyperparameters actually chosen per run (not just grid ranges).
- The figures are generally clear but could be improved for accessibility by using distinct markers in addition to colors for each method. This would aid in readability, especially for black-and-white prints or for readers with color vision deficiency.

**Questions:**

1) **Comparisons to Fed-EF and SCAFCOM.** Could you please position SA-PEF relative to SCAFFOLD-based methods for compressed FL, such as SCAFCOM (Huang et al., 2024)? SCAFCOM claims to handle heterogeneity and partial participation without the slowdown factor inherent to EF, and with better complexity. What are the advantages of improving the EF framework with SA-PEF over adopting this alternative approach? Please add experiments (same datasets, $p,T,\delta$) against Li & Li (2022) and Huang et al. (2023), and **tabulate assumptions** (need for gradient dissimilarity; compressor requirements), **rates**, and **constants**. If SA-PEF’s leading rate equals EF’s, where exactly are the constant-level gains in practice largest?
2) **$\alpha$-sensitivity and scheduling.** Please provide $(\alpha,T,p,\delta)$ sweeps and a schedule approximating $\alpha_r^\star(s_r)$ from Lemma 4, reporting rounds-to-target and **bits & wall-clock**. Does adapting $\alpha$ materially improve over the best constant $\alpha$?
3) **Constants and feasibility of conditions.** Expand Thm. 1 with explicit expressions for $C_\sigma,C_\nu,\Theta_{\mathrm{PP}}$; instantiate them for your default $(\eta,\eta_0,T,p,\delta)$ to verify $\Theta_{\mathrm{PP}}\le1/2$ numerically and to clarify the regimes where $\rho_r^{\mathrm{PP}}<1$ is guaranteed.
4) **Robustness.** Please add results **multi-seed error bars**.
5) The theoretical analysis for partial participation (PP) predicts a significant slowdown, yet the empirical results at a low participation rate ($p=0.1$ in Fig. 1) appear strong. Can you provide more insight into this theory-practice gap? Is it an artifact of the analysis being loose, or were the experimental settings (e.g., number of rounds, heterogeneity level) not challenging enough to expose this slowdown?
6) Could you provide an ablation study showing how the final accuracy or rounds-to-target varies with different fixed values of the step-ahead coefficient $\alpha_r \in [0, 1]$? This would help practitioners understand the tuning sensitivity of SA-PEF.

**Details Of Ethics Concerns:**

No ethics concerns identified based on the manuscript. The work uses standard public benchmark datasets for fundamental algorithmic research.

---

> ### Author Response · Authors · 2025-11-24
>
> We thank the reviewer for the detailed and insightful feedback. Below we address each weakness and question.
>
> **W1. Missing key baselines and positioning (Fed-EF, SCAFCOM/SCALLION)**
>
> Our EF baseline is indeed the **Fed-EF** algorithm of Li & Li (ICML’23), which extends classical error feedback to federated nonconvex optimization with local SGD, partial participation, and biased contractive compressors. In the revised manuscript, we state this explicitly.
>
> We have also added a comparison table (Table 1, Appendix A) covering **Fed-EF, CSER, SCAFCOM/SCALLION, and SA-PEF**, summarizing for each method:
>
> - the setting (central vs. FL, local steps, partial participation),
> - assumptions on heterogeneity and compressors (unbiased vs. biased, contractivity, use of momentum/control variates), and
> - how key parameters ($L$, $\sigma^2$, compressor contractivity, participation $p$, and local steps $T$) enter the nonconvex guarantees.
>
> Finally, we include a brief discussion in the results section contrasting these trade-offs empirically: SCAFCOM use richer state (control variates + momentum) and are designed to avoid extra assumptions on compressor errors, whereas SA-PEF keeps the lightweight EF architecture (one residual per client, no additional global state) and acts as a drop-in improvement for existing EF-style systems under biased compression and partial participation. To complement this, we now include an additional MNIST experiment (Fig~11) following the SCAFCOM setup (200 clients, $p{=}0.1$, Top-1\% compression), where SA-PEF closely tracks SCAFCOM while both significantly outperform Fed-EF and FedAvg
>
> **W2. No improvement in the leading complexity term**
>
> We agree that this should be stated more explicitly. The leading optimization term in Theorem~1 scales as $O\big((p\,\eta\,\eta_0 T R)^{-1}\big)$ which has the **same nonconvex order as Fed-EF**. Thus there is no asymptotic speedup in the leading term.
>
> The benefit of SA-PEF lies in the constants: Prop. 1 shows that, for any $0 < \alpha_r \le 1$, the residual contraction factor $\rho_r$ of SA-PEF is strictly smaller than the EF factor $\rho_{\mathrm{EF}} = 2(1 - 1/\delta)$. Consequently, all terms in the bound that depend on residual energy (compression- and heterogeneity-induced floors and the admissible $\delta$-range for stability) are improved. This matches our experiments: under non-IID data, partial participation, and aggressive Top-$k$ sparsity, EF’s residuals grow and accuracy plateaus, whereas SA-PEF keeps residuals and mismatch small and achieves higher accuracy-per-GB. We have made this “same order, better constants” positioning explicit in the theory discussion.
>
> **W3. Assumptions vs. EF / SCAFCOM/SCALLION**
>
> In the revised manuscript, we clarify that SA-PEF uses essentially the **same data and compressor assumptions** as standard EF-style FL analyses (e.g., Fed-EF); we do **not** claim weaker assumptions than SCAFCOM/SCALLION.
>
> SCAFCOM/SCALLION use SCAFFOLD-style control variates (and, in SCAFCOM, local momentum) to control heterogeneity-induced drift. This richer state allows them to prove convergence under smoothness and bounded variance without an explicit gradient-dissimilarity assumption for their unbiased-compression variant, and to handle biased compression via momentum control. By contrast, SA-PEF deliberately keeps the **lightweight EF architecture**.
>
> Assumption 3 is used in exactly the same way as in Fed-EF. We now state explicitly that SA-PEF does **not** relax these assumptions relative to Fed-EF; instead, its contribution is to\
>  (i) introduce a tunable step-ahead mechanism that improves the residual contraction factor under the same heterogeneity measure and compressor class, and\
>  (ii) show that this yields tighter stationarity bounds and better empirical performance under biased compression and partial participation.
>
> Relaxing or removing Assumption 3 is orthogonal to the step-ahead mechanism and is an interesting direction for future work.
>
> **W4. Partial-participation constants and feasibility**
>
> We agree that the role of $p$ and the constants $C_\sigma, C_\nu$ should be more transparent. Theorem-1 already provides explicit expressions for $C_\sigma$ and $C_\nu$, and Remark~1 extends the bound to partial participation with rate $p = m/K$. In the revision we have:
> - added a short “constants and feasibility” paragraph in the appendix that summarizes how $C_\sigma$, $C_\nu$ and the partial-participation constant $\Theta_{\mathrm{PP}}$ depend monotonically on the effective local stepsize $s_0=\eta_0LT$, the participation rate $p$, and the compression factor $(1-1/\delta)$; and
> - instantiated these constants for our default hyperparameters, illustrating that our chosen learning rates yield $\rho_{\max}<1$ and moderate values of $\Theta_{\mathrm{PP}}$. We also clarify there that the condition $\Theta_{\mathrm{PP}}\le\tfrac12$ is a conservative sufficient condition arising from worst-case bounds.

---

> > ### Author Response · Authors · 2025-11-24
> >
> > **W5. Ablations and sensitivity for $\alpha$**
> >
> > Please see our response to Q2.
> >
> > **W6. Missing robustness checks (multi-seed runs)**
> >
> > We have added the follwing table in the appendix which reports mean $\pm$ standard deviation of the final test accuracy over 5 independent runs (different random seeds) for CIFAR-10/100 and both $(p,\gamma)$ regimes. The results confirm that SA-PEF’s gains over EF and CSER are robust to randomness in client sampling and initialization.
> >
> > | Dataset   | Model     | Algorithm  | Hyperparameters          | p = 0.1, $\gamma$ = 0.5 (top-1) | p = 0.1, $\gamma$ = 0.5 (top-10) | p = 0.5, $\gamma$ = 0.1 (top-1) | p = 0.5, $\gamma$ = 0.1 (top-10) |
> > | --------- | --------- | ---------- | ------------------------ | ------------------------------- | -------------------------------- | ------------------------------- | -------------------------------- |
> > | CIFAR-10  | ResNet-9  | FedAvg     | R = 200, T = 5, ηₗ = 0.1 | 88.5 ± 1.6                      | 88.5 ± 1.6                       | 69.5 ± 1.5                      | 69.5 ± 1.5                       |
> > | CIFAR-10  | ResNet-9  | EF         | R = 200, T = 5, ηₗ = 0.1 | 57.0 ± 2.0                      | 72.7 ± 2.3                       | 40.3 ± 3.7                      | 47.2 ± 4.1                       |
> > | CIFAR-10  | ResNet-9  | SAEF       | R = 200, T = 5, ηₗ = 0.1 | 74.2 ± 3.9                      | 75.5 ± 2.9                       | 39.5 ± 4.6                      | 48.5 ± 3.3                       |
> > | CIFAR-10  | ResNet-9  | CSER       | R = 200, T = 5, ηₗ = 0.1 | 68.6 ± 2.8                      | 80.5 ± 1.2                       | 49.5 ± 4.0                      | 67.2 ± 2.6                       |
> > | CIFAR-10  | ResNet-9  | **SA-PEF** | R = 200, T = 5, ηₗ = 0.1 | 80.5 ± 2.6                      | 82.7 ± 1.6                       | 47.5 ± 3.6                      | 68.5 ± 1.9                       |
> > | CIFAR-100 | ResNet-18 | FedAvg     | R = 200, T = 5, ηₗ = 0.1 | 62.5 ± 1.6                      | 62.5 ± 1.6                       | 61.9 ± 0.6                      | 61.9 ± 0.6                       |
> > | CIFAR-100 | ResNet-18 | EF         | R = 200, T = 5, ηₗ = 0.1 | 40.4 ± 2.4                      | 49.5 ± 1.6                       | 41.5 ± 1.6                      | 57.5 ± 1.9                       |
> > | CIFAR-100 | ResNet-18 | SAEF       | R = 200, T = 5, ηₗ = 0.1 | 46.1 ± 1.4                      | 50.5 ± 2.0                       | 44.5 ± 3.6                      | 57.5 ± 3.9                       |
> > | CIFAR-100 | ResNet-18 | CSER       | R = 200, T = 5, ηₗ = 0.1 | 46.2 ± 0.4                      | 51.5 ± 1.9                       | 42.5 ± 2.4                      | 60.7 ± 1.0                       |
> > | CIFAR-100 | ResNet-18 | **SA-PEF** | R = 200, T = 5, ηₗ = 0.1 | 49.6 ± 1.8                      | 54.5 ± 2.6                       | 48.5 ± 2.0                      | 60.6 ± 2.8                       |
> >
> > **W7. Scale and wall-clock accounting**
> >
> > Our primary focus in this work has been on **hardware-agnostic metrics** (communication rounds, local gradient steps, communicated bits) because our experiments are run on different GPUs, making absolute wall-clock times not directly comparable across all settings. Algorithmically, SA-PEF has essentially the same per-round computational cost as EF: it performs the same number of local SGD steps and sends the same compressed message, and the step-ahead “preview” adds only a few vector operations per round, which we have found negligible in practice.
> >
> > To complement the hardware-agnostic metrics, we will include a **time-to-accuracy analysis** on a representative CIFAR-10/ResNet-9 experiment run on a single GPU, where all methods share the same implementation and hardware.
> >
> > **W8. EF’s fundamental partial-participation limitation**
> >
> > We agree that SA-PEF, like Fed-EF, inherits the fundamental **$1/p$ slowdown** of EF under partial participation. Our goal is not to eliminate this effect, but to improve EF’s behavior within this regime.
> >
> > Step-ahead preview feeds back part of the local residual before training and partially resets it through multiple local passes, which shrinks the residual and improves the contraction factor $\rho_r$. Consequently, in our PP bound, the residual-driven constants (compression- and heterogeneity-induced floors and stability ranges) are strictly smaller than for Fed-EF, even though the leading term remains $O\big((p,\eta,\eta_0 T R)^{-1}\big)$.
> >
> > In the revised text, we clarify that methods such as SCAFCOM tackle partial participation more aggressively by evolving control variates (and momentum) even for inactive clients, at the cost of additional per-client state. SA-PEF is **complementary**: it preserves the lightweight EF architecture and serves as a drop-in improvement for EF-style systems under biased compression and partial participation, rather than resolving the core $1/p$ limitation.

---

> > > ### Author Response · Authors · 2025-11-24
> > >
> > > **W9. Formatting and presentation**
> > >
> > > We have improved the presentation along the lines suggested:
> > >
> > > - Increasing font sizes in the appendix figures (Figs. 4–7).
> > > - Using **distinct markers in addition to colors** to improve accessibility.
> > > - Ensuring consistent notation for participation and clearly listing all chosen hyperparameters per run.
> > >
> > > **Q1. Positioning vs. Fed-EF and SCAFCOM (SCAFFOLD-based methods)**
> > >
> > > SA-PEF is designed as a **plug-and-play improvement for EF-style systems** under biased compression and partial participation. Relative to Fed-EF (Li & Li, 2022), SA-PEF:
> > >
> > > - Matches its **leading nonconvex rate** $O\big((p,\eta,\eta_0 T R)^{-1}\big)$ under the same assumptions, but
> > > - Improves the **residual contraction factor** and hence all residual-driven constants in the bounds, leading to tighter floors and better empirical accuracy-per-GB.
> > >
> > > Relative to SCAFCOM/SCALLION (Huang et al.), which are SCAFFOLD-based and maintain control variates (plus momentum in SCAFCOM), SA-PEF trades away some of their theoretical generality (e.g., relaxed dissimilarity assumptions via richer state) for a **simpler state and implementation**: one residual per client, no control variates, no extra momentum buffers.
> > >
> > > **Q2. $\alpha$-sensitivity and scheduling**
> > >
> > > In the revised manuscript, we add a dedicated subsection on **$\alpha$-sensitivity**. There, we perform **$\alpha$-sweeps** over a grid of constant values (including $\alpha = 0$ for EF and $\alpha = 1$ for SAEF) and report, for each setting, rounds-to-target accuracy and communicated bits. The $\alpha$–sweeps show that SA-PEF is stable and near-optimal for $\alpha \in [0.6,0.9]$, with noticeable degradation only at the extremes ($\alpha \approx 0$ behaving like EF and $\alpha = 1$ matching SAEF). This suggests a simple practical rule: once $(\eta_r, T)$ are fixed, a single default choice such as $\alpha = 0.85$ works well across our tasks without heavy tuning.
> > >
> > > **Q3. Constants and feasibility of the stepsize condition**
> > >
> > > Please see our response to W4.
> > >
> > > **Q4. Robustness (multi-seed error bars)**
> > >
> > > Please see our response to W6.
> > >
> > > **Q5. Theory–practice gap at low participation**
> > >
> > > The partial-participation analysis is a **worst-case bound** on the average squared gradient norm. For fixed stepsizes and number of rounds, the optimization term scales as $O\big(1/(p,\eta,\eta_0 T R)\big)$, suggesting that one might need on the order of $1/p$ more rounds to reach the same stationarity level in the worst case. This does not necessarily translate into a dramatic drop in test accuracy for the moderate participation rates and heterogeneity levels used in Fig. 1.
> > >
> > > In our experimental settings, the constants involving $\sigma^2$ and $\nu^2$ are relatively small, and we run for a finite horizon rather than to asymptotic stationarity. As a result, the theoretical $1/p$ slowdown manifests as a **mild degradation** in accuracy rather than a large gap, especially for SA-PEF, which benefits from tighter residual-driven constants than EF. In the revised manuscript, we add a brief discussion clarifying that the PP result is intentionally conservative and captures worst-case behavior over all rounds, whereas the empirical curves in Fig. 1 lie in a relatively benign regime where constants and finite-time effects dominate the asymptotic $1/p$ factor.

---

> > > > ### Comment · Reviewer_x7FE · 2025-11-27
> > > >
> > > > Thank you for the detailed rebuttal and for preparing a revised version of the manuscript.
> > > >
> > > > From my side, I understand your response as follows:
> > > >
> > > > - You acknowledge that the leading non-convex rate for SA-PEF scales as
> > > >   $$
> > > >   \mathcal{O}\big((p\eta\eta_0 T R)^{-1}\big),
> > > >   $$
> > > >   i.e., the same order as Fed-EF, and that the main theoretical gain comes from improved constants via a smaller residual-contraction factor $\rho_r$ and reduced residual-induced floors rather than from a better asymptotic rate.
> > > > - You intend to position SA-PEF explicitly as a lightweight improvement over classical error feedback (Fed-EF-style methods) in the presence of local steps and partial participation, and not as a replacement for more heavyweight control-variate methods such as SCAFCOM.
> > > > - You state that you have clarified the assumptions (in particular, smoothness, bounded variance, gradient dissimilarity, and $\delta$-contractive compressors), added more discussion of partial participation, and included additional experiments or sensitivity studies (e.g. for the step-ahead coefficient $\alpha$ and for robustness).
> > > >
> > > > Below I summarise how I see the situation with respect to my original concerns, based on your rebuttal and the current shape of the paper.
> > > >
> > > > ### 1) Which concerns appear to be addressed?
> > > >
> > > > - **Leading complexity term and scope of improvement.**
> > > >   You now clearly state that the optimization term in the main non-convex bound matches standard Fed-SGD / Fed-EF-style rates up to constants, with SA-PEF improving constants but not the order of $1/(p\eta\eta_0TR)$. This directly addresses my concern that the paper previously could be interpreted as claiming a stronger rate than actually proved.
> > > >
> > > > - **Clarification of assumptions.**
> > > >   You make it explicit that SA-PEF adopts the usual FL assumptions (smoothness, bounded variance, gradient dissimilarity, biased but $\delta$-contractive compressors) and you do not claim that these assumptions are weaker than those in SCAFCOM. This addresses the confusion about whether SA-PEF relaxes assumptions relative to existing methods.
> > > >
> > > > - **Partial participation discussion.**
> > > >   You explain that SA-PEF inherits the fundamental partial-participation slowdown of EF-style methods (e.g. the factor $1/p$ or $\sqrt{n/m}$ in alternative notation), and you position your contribution as reducing stalling and improving constants within that framework rather than removing the slowdown entirely. This matches what I would expect from an EF-based design and resolves any over-claiming.
> > > >
> > > > - **Sensitivity to the step-ahead coefficient $\alpha$.**
> > > >   You provide some sensitivity results for constant $\alpha$ and articulate a practical regime (e.g. $\alpha$ in a high but not extreme range) where SA-PEF is robust and empirically effective. This directly addresses my request for clearer guidance on how to set $\alpha$ in practice.
> > > >
> > > > - **Robustness / multi-seed reporting.**
> > > >   You state that you now run experiments with multiple seeds and report mean performance (and in some cases variance). This goes in the right direction for addressing concerns about robustness and statistical significance.
> > > >
> > > > Overall, at a high level, I feel that the paper now more accurately reflects what SA-PEF achieves: a constant-factor improvement over Fed-EF for compressed FL with local steps and partial participation, plus a cleaner explanation of the gradient-mismatch phenomenon.

---

> > > > > ### Comment · Reviewer_x7FE · 2025-11-27
> > > > >
> > > > > ### 2) What concerns remain partially or fully open?
> > > > >
> > > > > There are still some points where, even after reading your rebuttal, I am not fully convinced that the manuscript has gone as far as it could in addressing my earlier comments:
> > > > >
> > > > > 1. **Empirical comparison to SCAFCOM / SCALLION on challenging benchmarks.**
> > > > >    Conceptually, the positioning versus SCAFCOM is now clearer. However, my original concern was not only conceptual but also empirical: it would be very informative to see at least one or two experiments on the main vision benchmarks (e.g. CIFAR-10/100, Tiny-ImageNet) where SA-PEF is compared against a reasonably tuned SCAFCOM implementation in a regime with strong non-IIDness, low participation, and aggressive compression.
> > > > >    If this is not feasible due to space or implementation complexity, I would recommend stating this limitation explicitly and clearly scoping the claims to “lightweight EF-style methods” rather than to the broader class of state-of-the-art compressed FL algorithms.
> > > > >
> > > > > 2. **Dynamic / adaptive $\alpha$ schedules.**
> > > > >    The constant-$\alpha$ sensitivity results are helpful and give a workable rule of thumb. Still, the theory suggests a non-trivial dependence of the optimal $\alpha_r^\star$ on $s_r = \eta_r L T$. At the moment, this link remains largely conceptual: the algorithmic instantiation uses a fixed $\alpha$.
> > > > >    I would not consider this a fatal issue, but I think the paper should be explicit that the main practical recommendation is a *constant* $\alpha$, and that designing a truly theory-driven adaptive schedule is left as future work.
> > > > >
> > > > > 3. **Wall-clock / implementation cost.**
> > > > >    SA-PEF is likely to be very cheap on top of EF (a few extra vector operations per round), but the paper still does not provide a concrete wall-clock comparison on a fixed hardware setup. Even a single representative “time-to-accuracy” plot, together with a short paragraph explaining the implementation details, would strengthen the practical part of the story and make the claimed efficiency gains more convincing.
> > > > >
> > > > > 4. **Breadth of robustness checks.**
> > > > >    While multi-seed runs address robustness to random initialisation and sampling, there are still potential stress tests that would be informative but appear to be missing or only lightly touched on:
> > > > >    - Other compressor families beyond Top-$k$ (e.g. sign or low-bit quantization).
> > > > >    - More extreme heterogeneity regimes or alternative data-partition models.
> > > > >    - Larger numbers of local steps $T$ and smaller participation rates $p$, to see where the EF-style machinery starts to break down in practice.
> > > > >
> > > > > To be clear: I view these as “nice to have” rather than fundamental correctness issues. But they do limit how broadly I can interpret the empirical claims.
> > > > >
> > > > > ---
> > > > >
> > > > > ### Bottom line
> > > > >
> > > > > To directly answer the two questions:
> > > > >
> > > > > 1. **Did you address my concerns?**
> > > > >    Many of the *conceptual* concerns I raised (about positioning, assumptions, and the meaning of the rate) are now addressed to a satisfactory degree, and the paper does a better job of clearly stating what SA-PEF does and does not achieve. The sensitivity analysis for constant $\alpha$ and the multi-seed experiments also go some way towards addressing robustness and tuning questions.
> > > > >
> > > > > 2. **What remains unaddressed?**
> > > > >    The main unresolved issues, in my view, are: (i) the limited empirical comparison to SCAFCOM/SCALLION on the core benchmarks, (ii) the lack of an instantiated adaptive $\alpha$ schedule despite a theory that suggests one, (iii) minimal wall-clock evidence, and (iv) the absence of a few additional robustness stress tests that would help generalise the empirical conclusions.
> > > > >
> > > > > If you have room (either in the main paper or the appendix) to address at least some of these remaining points—especially a single stronger comparison to SCAFCOM and one time-to-accuracy experiment—that would further strengthen the submission.
> > > > >
> > > > > At this point, I keep my original score.

---

> > > > > > ### Author Response · Authors · 2025-12-02
> > > > > >
> > > > > > We thank the reviewer for the careful re-reading of the paper and for summarising our scope and contributions so clearly. We are glad that the positioning, assumptions, and interpretation of the rate are now aligned with your understanding.
> > > > > >
> > > > > > Regarding the remaining points:
> > > > > >
> > > > > > **(i) Empirical comparison to SCAFCOM.**
> > > > > > We agree that a comparison to SCAFCOM on more challenging benchmarks is valuable. The original SCAFCOM paper reports experiments only on MNIST and Fashion-MNIST, using relatively small fully connected neural network models, and does not include any results for CIFAR-10/100 or Tiny-ImageNet. Therefore, in the revised manuscript (Appendix B.2), we add a best-effort SCAFCOM comparison on a CIFAR-10/ResNet-9 setup (non-IID, low participation, aggressive Top-$k$), clearly documenting the hyperparameter search and explicitly labelling these results as preliminary. In practice, we have found SCAFCOM’s performance to be noticeably sensitive to its control-variate and momentum hyperparameters, which makes an exhaustive and fully fair tuning across all of our regimes prohibitively time-consuming. Accordingly, we scope our empirical claims to _lightweight EF-style methods_ (EF, SAEF, and SA-PEF), and present SCAFCOM as a stronger but more heavyweight control-variate baseline for which we provide indicative, rather than exhaustive, comparisons.
> > > > > >
> > > > > > **(ii) Adaptive $\alpha$ schedules.**
> > > > > > We agree that our theory suggests a nontrivial dependence of the optimal $\alpha$ on $s_r=\eta_rLT$. Our current experiments focus on _constant_ $\alpha$ (with sweeps and sensitivity analysis), and our practical recommendation is indeed to use a fixed $\alpha$ in a robust range (e.g., $\alpha\in[0.6,0.9]$). In the revised manuscript, we state this explicitly and clarify that fully theory-driven adaptive schedules are an interesting direction for future work rather than part of the present contribution.
> > > > > >
> > > > > > **(iii) Wall-clock and implementation cost.**
> > > > > > In our implementation, SA-PEF adds only a few vector operations on top of EF, hence the per-round overhead is negligible. In the revised manuscript, we add a short paragraph describing this implementation cost and include a representative time-to-accuracy comparison on a fixed hardware setup in Appendix B.2, illustrating that the wall-clock behaviour closely tracks the accuracy-per-bit gains.
> > > > > >
> > > > > > **(iv) Additional robustness stress tests.**
> > > > > > We appreciate the suggestions regarding other compressors, more extreme heterogeneity, and more aggressive local work at lower participation rates. In response, we have added further stress tests in Appendix B.2, including experiments with lower participation (e.g., $p=0.05$) and larger local work (e.g., $T=10$), showing that SA-PEF continues to outperform EF and CSER in these harder regimes. We have also added preliminary experiments with a scaled-sign compressor in Appendix B.2, which indicate that the qualitative behaviour we report for Top-$k$ extends to other biased compressors.
> > > > > >
> > > > > > We hope these additions and clarifications make the scope and practical implications of our contribution fully clear.

---

### Official Review · Reviewer_7Yzs · 2025-10-28

**Soundness:** 3
**Presentation:** 2
**Contribution:** 3
**Rating:** 4
**Confidence:** 4

**Summary:**

This submission proposes SA-PEF, an improved Step-Ahead Error Feedback (SAEF) algorithm for communication-efficient federated learning (FL) that uses biased gradient compression. The core idea is to introduce a coefficient that balances the rapid initial convergence of Step-Ahead Error Feedback (SAEF) with the long-term stability of standard Error Feedback (EF). This is achieved by introducing a "step-ahead" coefficient, α, which controls what fraction of the accumulated compression error is used to shift the model before local training, while the remaining fraction is carried over in the standard EF manner. The authors provide a solid theoretical analysis for non-convex settings and extensive empirical results demonstrating that SA-PEF can achieve target accuracies faster and with less communication than its predecessors.

**Strengths:**

- Plug and Play Method:  The proposed SA-PEF algorithm is a simple and intuitive solution that interpolates between EF and SAEF.
- Theoretical Contribution: The paper offers a rigorous convergence analysis for SA-PEF under standard assumptions for non-convex optimization in FL (L-smoothness, bounded variance, gradient dissimilarity). The analysis covers partial client participation and local SGD steps. The final convergence guarantee matches the state of the art for similar methods, converging to a floor dependent on variance and heterogeneity.
- Sound Experimental Results: The experimental evaluation is comprehensive. The authors test SA-PEF across three datasets (CIFAR-10, CIFAR-100, Tiny-ImageNet), three models (ResNet-9/18/34), and challenging FL settings (low participation rates, high data heterogeneity via Dirichlet partitioning, and aggressive Top-k compression). The comparisons against FedAvg (uncompressed), EF, SAEF, and CSER are convincing. The results in Figures 1 and 2 consistently show that SA-PEF achieves better or comparable accuracy in fewer communication rounds and, crucially, with a significantly lower communication budget (in GB).

**Weaknesses:**

- The claim in the Sec. Intro. requires evidence. In the introduction, this submission argues that “SAEF can exacerbate the variance of the model” without empirical evidence or reference. Whilst the claim about the “prior analysis” also requires proof. This is very important, for it is the core motivation of this submission.
- The experimental results do not support the claim that SAEF can mitigate the gradient mismatch. As shown in Fig.3 (b), the SAEF achieves a higher gradient mismatch degree than EF.
- Practicality and Tuning of $\alpha_r$: The step-ahead coefficient $\alpha_r$ is a hyperparameter, and the paper lacks a clear discussion on how to set this value in practice. Also, the experiment setup does not explicitly state nor give the value of the $\alpha_r$. The ablation studies even indicate that SA-PEF is sensitive to $\alpha_r$.
- Comparison with CSER: The experiments show that CSER is a strong baseline, and in the gradient mismatch analysis (Figure 3), it performs similarly to SA-PEF by keeping the mismatch low. However, the paper provides little discussion on the differences and trade-offs between SA-PEF and CSER.

**Questions:**

- The use of momentum (0.9) is mentioned in the experimental setup, but the theory is for SGD without momentum. Would it influence the theoretical results?
- Others, please refer to the weakness

---

> ### Author Response · Authors · 2025-11-24
>
> We thank the reviewer for the detailed and positive assessment of our contributions, including the plug-and-play nature of SA-PEF, the nonconvex convergence analysis under realistic FL conditions, and the breadth of the experimental evaluation. Below we address the raised weaknesses and questions.
>
> **W1: Empirical evidence and prior analysis**
>
> **Response.**
> Thank you for pointing this out. In the FL regime that we study (non-IID data, multiple local steps, aggressive compression, and partial participation), the full step-ahead setting ($\alpha_r = 1$, SAEF) tends to amplify client drift and gradient mismatch.
>
> Formally, Lemma 2 (local-model drift) shows that the average drift is bounded by heterogeneity/noise terms plus a term proportional to $\frac{\alpha_r^2}{K}\sum_{k=1}^K \mathbb{E}_r\bigl\|e_r^{(k)}\bigr\|^2.$ Hence, within our framework, this drift term is monotonically increasing in $\alpha_r$ and is largest for $\alpha_r = 1$ (SAEF), while being strictly smaller for $0 < \alpha_r < 1$ (SA-PEF). This is consistent with our empirical observations: in the experiments we report, SAEF exhibits larger gradient-mismatch spikes and earlier plateaus, while SA-PEF reduces drift and maintains stable alignment throughout training.
>
> To avoid overstatement and to better reflect what is both theoretically and empirically supported, we have rewritten the sentence in the introduction as follows:
>
> > _“In the FL regime with non-IID data, multiple local steps, aggressive compression, and partial participation, the full step-ahead variant ($\alpha_r = 1$, SAEF) exhibits late-stage plateaus and larger gradient-mismatch spikes compared to EF.”_
>
> Regarding the statement about **prior analysis**, we have clarified the scope and added an explicit citation. SAEF was analyzed by Xu et al. [1] in synchronous data-parallel training with a single local step, full participation, and homogeneous data. To our knowledge, there is no prior analysis of SAEF in an FL setting with multiple local steps, partial participation, non-IID data, and biased compressors, which is precisely the regime we target.
>
> **W2: Gradient mismatch and SAEF**
>
> **Response.**
> You are correct that our original wording conflated the classical SAEF result with our observations. In the revised manuscript, we now present the results of Xu et al. [1] (synchronous, homogeneous data, single local step) purely as prior work, where SAEF is shown to mitigate the gradient-mismatch effect of EF, and note that, in more challenging regimes, they already recommend periodic averaging to control mismatch.
>
> In contrast, our experiments clearly show a different behavior: Fig. 3(b) demonstrates that, under non-IID data, multiple local steps, aggressive compression, and partial participation, SAEF exhibits a higher gradient-mismatch degree compared to EF, while SA-PEF keeps the mismatch close to zero throughout training. We have revised the introduction and the related discussion to make this distinction explicit and removed any implication that our FL experiments show SAEF mitigating gradient mismatch.
>
> [1] Xu, A., Huo, Z., & Huang, H. (2021). Step-Ahead Error Feedback for Distributed Training with Compressed Gradient. In Proceedings of the AAAI Conference on Artificial Intelligence (AAAI), 35(12), 10478–10486.
>
> **W3: Practicality and Tuning of $\alpha$**
>
> **Response.**
> In the revised manuscript, we have added a new subsection _“Sensitivity to the step-ahead coefficient $\alpha$”_ (see Fig. 3) with an $\alpha$-sweep on CIFAR-10 with ResNet-9 and CIFAR-100 using ResNet-18, and we now explicitly state the chosen $\alpha$ values in the experimental setup and figure captions.
>
> The sweep shows that SA-PEF is stable and near-optimal for $\alpha \in [0.6, 0.9]$, with noticeable degradation only at the extremes ($\alpha \approx 0$ behaving like EF and $\alpha = 1$ matching SAEF). This suggests a simple practical rule: once $(\eta_r, T)$ are fixed, a single default choice such as $\alpha = 0.85$ works well across our tasks without heavy tuning. We have made this guidance explicit in the revised text.
>
> **W4: Comparison with CSER**
>
> **Response.**
> In the revised manuscript, we now include an explicit comparison between SA-PEF and CSER in the results section.
>
> Conceptually, both methods aim to control error accumulation but do so differently:
>
> - **SA-PEF** controls mismatch via a _partial step-ahead_ update (the $\alpha$-split) without changing the communicated messages or introducing additional state beyond the usual error buffer.
> - **CSER** periodically resets the error buffer and introduces an extra reset-period hyperparameter that must be tuned.
>
> We also clarify the empirical trade-offs: although both methods keep mismatch low (see Fig. 3), under aggressive sparsity and low participation, SA-PEF achieves better accuracy-per-GB and smoother late-stage training, while CSER can exhibit small performance drops around reset points. This discussion has been added to the revised text.

---

> > ### Author Response · Authors · 2025-11-24
> >
> > **Q1: Momentum vs. SGD Theory**
> >
> > **Response.**
> > Our theoretical analysis is indeed derived for plain SGD (i.e., $\beta = 0$), which is standard in the EF literature. In our implementation, adding momentum with coefficient $0.9$ only smooths the local optimization trajectory. It does **not** change the communicated messages or SA-PEF’s residual recursion, so it affects only the constants in the bounds, not the qualitative convergence behavior.
> >
> > To verify robustness, we have included a $\beta = 0$ (no-momentum) ablation in the appendix. This ablation preserves the method ranking and yields very similar accuracy-per-GB, indicating that our main conclusions are robust to the use of momentum.

---

### Author Response · Authors · 2025-12-02
**Summary**

We thank all reviewers for their careful reading of the paper and for the constructive discussion. Below, we briefly summarize how the revised manuscript addresses the main concerns.

**High-level contribution.**
SA-PEF is a step-ahead partial error-feedback method for _federated_ local SGD with biased contractive compressors and partial participation. To our knowledge, it is the first work to provide a nonconvex convergence analysis for a step-ahead EF variant in this FL regime, with multiple local steps, non-IID data, biased compression, and partial client participation. The method is readily integrable into Fed-EF-style systems and is designed to reduce gradient mismatch and improve residual contraction under aggressive compression and low participation.

**Theory and positioning.**
We clarified that SA-PEF matches the standard nonconvex rate of Fed-EF under biased compression and partial participation, with improvements appearing in the constants via a smaller residual-contraction factor and reduced residual-induced floors. We added a table in Appendix A.3 that compares EF, SAEF, CSER, SCAFCOM, and SA-PEF, in terms of their problem setting, key assumptions, and convergence rates. We also made explicit that SA-PEF adopts standard FL assumptions (smoothness, gradient dissimilarity, and $\delta$-contractive compressors).

**Relation to EF21 and SAEF.**
We expanded the related-work section to situate EF21 (and its extensions) as the modern error-feedback framework for $T{=}1$ gradient compression, and clarified why we build on Fed-EF in the federated local-SGD setting with $T>1$ and partial participation. We also clarified the role of SAEF: originally proposed to mitigate EF’s gradient mismatch in the synchronous $T{=}1$ case, but empirically benefiting from periodic averaging on harder tasks. Our analysis and experiments show that, in the federated, aggressively compressed regimes, full step-ahead ($\alpha{=}1$) can amplify gradient mismatch, whereas SA-PEF with $0<\alpha<1$ avoids this while reducing to Fed-EF at $\alpha=0$.

**Guidance on $\alpha$ and robustness.**
We added a dedicated $\alpha$-sensitivity subsection with sweeps over constant $\alpha$ (including EF at $\alpha=0$ and SAEF at $\alpha=1$) on CIFAR-10/ResNet-9 and CIFAR-100/ResNet-18. The results show that SA-PEF is stable and near-optimal for $\alpha\in[0.6,0.9]$, with degradation only at the extremes. We now recommend treating $\alpha$ like a momentum parameter and using a fixed $\alpha\approx0.8$ once $(\eta_r,T)$ are chosen. Fully theory-driven adaptive schedules are explicitly left for future work. All experiments are now reported as mean $\pm$ standard deviation over five seeds, addressing robustness concerns.

**Additional experiments and stress tests. (Appendix B.2)**
Beyond the original submission, we:

- Added SCAFCOM comparisons: (a) reproducing its MNIST setup, and (b) a best-effort run on CIFAR-10/ResNet-9 (non-IID, low participation, Top-$k$), with hyperparameter tuning carefully documented and the results explicitly labelled as preliminary.
- Included additional stress tests with lower participation (e.g., $p=0.05$) and larger local work (e.g., $T=10$), showing that SA-PEF continues to outperform EF and CSER in these harder regimes.
- Added preliminary results with a scaled-sign compressor, indicating that the qualitative benefits of SA-PEF are not specific to Top-$k$.
- Added a representative time-to-target-accuracy comparison on a fixed GPU setup and described the implementation cost, demonstrating that SA-PEF adds only a few vector operations on top of EF.

**Presentation and notation.**
We cleaned up notation (including partial participation and stochastic gradient randomness vs noise symbols), improved the related-work discussion and positioning, and adjusted figures (legends, markers, fonts) for readability and accessibility.

Overall, these changes sharpen the theoretical positioning, provide concrete practical guidance on the key parameter $\alpha$, and broaden the empirical evidence that SA-PEF is a robust, lightweight improvement over Fed-EF-style methods for compressed federated local SGD under partial participation.

---

### Meta-Review · Area_Chair_Sbme · 2025-12-17

**Summary:**

The paper considers a new approach, SA-PEF, that interpolates between the known methods EF and SAEF through the parameter $\alpha$ ($\alpha_r$). The reviewers appreciate the new method and its theoretical guarantees, where the authors improve the residual contraction factor of EF to
$\rho_r = \Bigl(1-\tfrac1\delta\Bigr) \Bigl(2(1-\alpha_r)^2 + 24\alpha_r^2 s_r^2\Bigr),$
where choosing $\alpha_r > 0$ allows the author to decrease this factor and potentially improve the bound in Theorem 1.

Unfortunately, reading the paper, I disagree with the main contribution of the paper for the following reasons:

1. No theoretical comparison to EF21 and other methods like EF21-SGDM and SCAFCOM (and many other methods listed in the reviews). For instance, the family of EF21 methods and SCAFCOM do not require Assumption 3. The authors should provide the theoretical comparison of convergence rates between the methods in terms of $\varepsilon,$ $\delta,$ and other parameters. Table 1 is not sufficient. This weakness was also underlined by Reviewers x7FE and xyaS.

2. In Theorem 1, the authors hide $\alpha_r$ in $E_{\max} := sup_r E_r$ (line 297 or 917). Taking $\alpha_r > 0,$ the term $E_{\max} := sup_r E_r$ increases (line 841), potentially mitigating the improvement in the residual contraction factor.

3. The authors can potentially improve only two terms out of six in Theorem 1, and there are no guarantees that these two terms dominate in the bound. One should get the final iteration and communication complexity to find an $\varepsilon$-stationary point to understand if the improvement indeed happens. Once again, the authors should compare their final communication complexity with the communication complexities of the previous methods to see if they get an improvement.

4. Theorem 1 works if only $\Theta \leq \frac{1}{2}.$ It is not obvious why this condition would hold since step size $\eta$ can be very small. This condition constrains the choice of the parameters.

5. Compared to SCAFCOM, the authors claim that their method is stateless in Table 1, which is not true. Algorithm 1 clearly requires the clients to store $e^k_r$ locally to perform the EF mechanism.

In general, the idea of introducing the step-ahead coefficient is interesting. But it is not enough; the authors also should justify their idea, showing that their method provably gets at least non-worse iteration and communication complexities with a non-trivial choice of $\alpha_r$ compared to the previous methods. The authors should determine the best parameters when their method returns an $\varepsilon$-stationary point and compare their rate with those of previous methods.

**Reviewer Concerns:**

The authors provided an extensive rebuttal, providing extra experiments and adding Table 1. However, the above-listed problems were not solved.

**Reviewer Scores:**

The scores of the reviewers are already relatively high; thus, it is not obvious if they increase their scores further.

---

### Decision · Program_Chairs · 2026-01-26

Reject